# CELLECT: contrastive embedding learning for large-scale efficient cell tracking

Hongyu Zhou [1,2,3,4,12], Seonghoon Kim [1,2,5,12], Zhifeng Zhao[1,2], Jiaqi Fan [1,2], Wen Huang[1,2], Xinghua Sui[6], Lizhi Shao [7], Haoran An[8,9], Jing-Ren Zhang[10], Jiamin Wu [1,2,3,4,11] ✉ & Qionghai Dai [1,2,3,4,11] ✉

Quantitative analysis of large-scale cellular behaviors plays an increasingly crucial role in understanding mechanisms of diverse physiopathological processes, but achieving cell tracking with both high performance and efficiency in practical applications remains a challenge. Here we introduce CELLECT, a contrastive embedding learning method for large-scale efficient cell tracking, and demonstrate it on the *Caenorhabditis elegans* dataset in the Cell Tracking Challenge. By contrastive learning of latent embeddings of diverse cellular structures, a CELLECT model pretrained on a single public dataset can be effectively applied across different imaging modalities and species with broad generalization. Using advanced two-photon imaging, CELLECT enables real-time 3D tracking of large-scale B cells with frequent divisions during germinal center formation in a mouse lymph node, quantitative identification of cell–bacterium interactions in the mouse spleen and high-fidelity extraction of neural signals during strong nonrigid motions. We believe that these results demonstrate broad applications of CELLECT in immunology, pathology and neuroscience.

Quantitative analysis of cellular behaviors is crucial for understanding biological systems and has widespread applications in immunology, pathology, development, neuroscience and pharmacology[1–5]. Recent advancements in optical imaging technologies and fluorescence indicators have ushered in a new era of high spatiotemporal resolution with a mesoscale field of view and long-term in vivo observations of biological tissues[6–17]. These developments allow intercellular activities to be visualized in native three-dimensional (3D) environments[18]. However, longer imaging durations and wider fields of view produce massive datasets that exceed manual processing capacity. The task of extracting high-dimensional valuable information from the immense pool of terabyte-scale or petabyte-scale imaging data has raised an urgent

need for downstream analysis algorithms with both high performance and high efficiency[19].

Deep learning-based cell-tracking algorithms have recently achieved remarkable success in tracking accuracy, closely matching human tracking results in various cellular images[2,20–28]. However, these methods typically require extensive manual annotations for model training, including details on cell position, shape and trajectory, limiting their applications in complex datasets such as time-lapse 3D imaging data of diverse cell structures in different species[24,29,30]. To alleviate this dependency on manual annotations, a tracking algorithm based on sparse annotations (linajea) has been proposed recently[31]. By using sparse annotations of cell positions within a sequence for

[1]Department of Automation, Tsinghua University, Beijing, China. [2]Institute for Brain and Cognitive Sciences, Tsinghua University, Beijing, China. [3]Beijing Key Laboratory of Cognitive Intelligence, Tsinghua University, Beijing, China. [4]IDG/McGovern Institute for Brain Research, Tsinghua University, Beijing, China. [5]Hangzhou Zhuoxi Institute of Brain and Intelligence, Hangzhou, China. [6]School of Pharmaceutical Sciences (Shenzhen), Shenzhen Campus of Sun Yat-sen University, Shenzhen, China. [7]School of Internet, Anhui University, Hefei, China. [8]Institute of Medical Technology, Peking University Health Science Center, Beijing, China. [9]Department of Microbiology and Infectious Disease Center, Peking University Health Science Center, Beijing, China. [10]Center for Infectious Biology, School of Basic Medical Sciences, Tsinghua University, Beijing, China. [11]Beijing National Research Center for Information Science and Technology, Tsinghua University, Beijing, China. [12]These authors contributed equally: Hongyu Zhou, Seonghoon Kim. ✉e-mail: wujiamin@tsinghua.edu.cn; qhdai@tsinghua.edu.cn

training, linajea substantially reduces the annotation burden and achieves accurate tracking of cells even during cell division events. However, its reliance on fixed-length temporal windows for global trajectory optimization introduces high computational costs and strict requirements on temporal sampling, limiting real-time performance. Therefore, there is still a missing gap in cell-tracking algorithms to achieve accurate tracking of large-scale cells in complex and dynamic 3D environments while maintaining high efficiency with sparse annotations.

Here, we propose a contrastive embedding learning method for large-scale efficient cell tracking, named CELLECT, to achieve real-time 3D tracking of large-scale cells in a complicated environment with state-of-the-art tracking accuracy on the *C. elegans* dataset in the Cell Tracking Challenge and only sparse annotations. Instead of using explicit intensity distributions in the 3D spatial domain directly for segmentation and localization in traditional methods, we developed a contrastive learning framework to first learn the latent embeddings of diverse cellular structures in public datasets, in which the differences among sparsely annotated cells can be maximized for better identification. Next, the segmentation and tracking process can be conducted in the latent embedding space with both high fidelity and low computational costs without further tuning. With quantitative validations in an open-source dataset and experimentally captured data, our CELLECT model pretrained solely on a dataset shows better performance and much faster processing speed than previous deep learning-based methods in diverse imaging modalities including confocal microscopy, light sheet microscopy, two-photon microscopy and light-field microscopy across different species.

The strong generalization capability of CELLECT facilitates diverse biological applications in an easy-to-use manner, which are challenging or time consuming for previous methods. In developmental biology, CELLECT enables better accuracy in lineage tracing during large-scale cell divisions with processing speed over 50-fold faster than that of linajea. In immunology, combined with two-photon synthetic aperture microscopy (2pSAM), CELLECT achieves real-time 3D tracking of large-scale B cells with frequent cell divisions during the complete formation process of a germinal center (GC) in the mouse lymph node. In pathology, quantitative identification of cell–bacterium interactions in the mouse spleen can be obtained with accurate 3D tracking and segmentation of heterogeneous sample structures. In neuroscience, CELLECT facilitates high-fidelity extraction of 3D neural signals during strong nonrigid motions. We believe that all these experiments and demonstrations have shown the broad applications of CELLECT with both high performance and high efficiency, opening up a new horizon for quantitative biology study in diverse fields.

## Results

### Overview of CELLECT

Conventional cell-tracking algorithms usually conduct feature extraction for each cell individually with region-of-interest (ROI) detection directly based on the intensity distributions in the two-dimensional or 3D spatial domain. While the intensity features are explicit, they exhibit large variances across different cell types and labeling methods (for example, cells with membrane labeling or nuclear labeling), often necessitating retraining the neural network to adapt to diverse cellular sizes and morphological characteristics. In addition, unifying a single pretrained neural network with strong generalization for large sample diversity is challenging. Although methods such as linajea[31] use distance-based global path optimization with time series data to filter out unreasonable candidate centers, enabling high-accuracy tracking with only sparsely annotated cell centers, they usually require substantial computational costs, reducing the scalability for high-throughput imaging data.

To address this problem, we developed a unified learning framework, CELLECT, aiming to map 3D intensity distributions into the 3D confidence map indicating the probability of being the cell center. The sparse annotations of cell positions are used to generate a ground truth multilevel confidence map based on the assumption that the image voxel with a distance closer to the labeled cell center has a higher probability of being the cell center (Fig. 1a). Next, we employed contrastive learning to learn the latent embeddings representing diverse cellular structures from a large public dataset in the Cell Tracking Challenge[24,32] by maximizing the difference between sparsely annotated cells and minimizing the feature distance within the same cell (Fig. 1b; detailed network structures are illustrated in Supplementary Fig. 1). In this case, diverse sample structures and sizes are compatible in this framework, leading to strong generalization of a pretrained model in different species and imaging modalities even in previously unseen datasets with nuclear or membrane labeling (Fig. 1c, Supplementary Fig. 2 and Supplementary Video 1). In addition, this latent embedding helps to distinguish dividing cells from nondividing cells as shown by the clustering results of the embedding vectors (Supplementary Fig. 3). Convolutional approximations of feature gradients reveal local variations across cellular regions, demonstrating CELLECT's generalization capability in segmentation tasks without the requirement of explicit segmentation annotations (Supplementary Fig. 4).

We employed a 3D U-Net architecture, using two adjacent frames as input (Supplementary Fig. 1). By leveraging the spatiotemporal differences in cell regions between the two frames, the model generates three maps: a confidence map for the first frame indicating the probability of being the cell center, a map of 64-channel features in the latent embedding for each voxel and a probability map predicting whether a cell undergoes division (Fig. 1a). Next, we can directly obtain a coarse estimation of cell size (radius) and the segmentation of each cell based on the confidence map. A lightweight 3D U-Net branch, referred to as the center enhancement network (CEN), is used to generate a 3D mask for cell centers by progressively increasing probability values closer to the cell center in the confidence map. We can then apply this 3D mask during inference to extract corresponding features for each cell for the tracking process. Given the relative sparsity of cells compared to the

---

**Fig. 1 | Schematic of CELLECT. a**, We trained the model using confidence maps generated from two adjacent frames with sparse annotations of center points. For each voxel in the first frame, the model outputs confidence map predictions, embedding vectors and division prediction probabilities. Using the confidence map, the model further derives size predictions, cell segmentation and an enhanced confidence map for center point regions. Peaks in the enhanced confidence map are selected as masks for cell center points. The features are extracted by masking the feature map with the center point mask to reduce the computational costs. AU, arbitrary units. **b**, Schematic of the training procedure. The model is designed to extract center point coordinates and a feature map. By combining the embedding vectors of different cell center points with contrastive learning, the model minimizes feature distances for the same cell while maximizing feature distances between different cells. **c**, Schematic of the inference procedure for cell tracking. The model extracts center points together and corresponding feature maps from the input data. Within each frame, redundant detections are merged by identifying the five nearest candidates per target cell and evaluating their feature distances with an intraframe MLP to determine whether they represent the same cell. Across frames, the model links by locating the five nearest candidates in the adjacent two frames and using an interframe MLP to classify each as the same cell, a division event or a different cell. **d**, Examples of center point and segmentation obtained via CELLECT and Imaris on nuclear-labeled cells (left) and membrane-labeled cells (right). Panels **a**–**c** use sample data from the Cell Tracking Challenge Fluo-N3DH-CE dataset. Panel **d** uses nuclear-labeled data from the mskcc-confocal dataset and membrane-labeled neutrophil data from mouse imaging (same as in Fig. 4). The model used in all examples was trained on sequence 3 of the mskcc-confocal dataset. Scale bars, 1 μm for CLSM (confocal laser scanning microscopy) images and 5 μm for sLFM (scanning light-field microscopy) images (**c**).

number of voxels in imaging data, this approach dramatically reduces computational costs. Finally, we can remove redundant cell centers and achieve cell linkage sequentially for each cell during the inference procedure of CELLECT (Fig. 1d). Within this process, a feature distance matrix is calculated in the latent embedding space, including five nearest candidate cell centers in terms of the spatial distance within the same frame and the next frame for each targeted cell. First, an intraframe multilayer perceptron (MLP) model is used to identify and remove redundant cell centers within the same frame (Supplementary Fig. 5).

Later, an interframe MLP model determines whether these candidates correspond to the same cell, a dividing cell or neither, enabling accurate linkage of cells between adjacent frames (Fig. 1d). The separation of the two lightweight MLPs from the core feature extraction pipeline reduces computational overhead and allows for scalable processing across spatial patches. Due to the optimization of the latent embedding in contrastive learning, lightweight network structures are sufficient for both MLPs to achieve efficient and accurate 3D cell tracking.

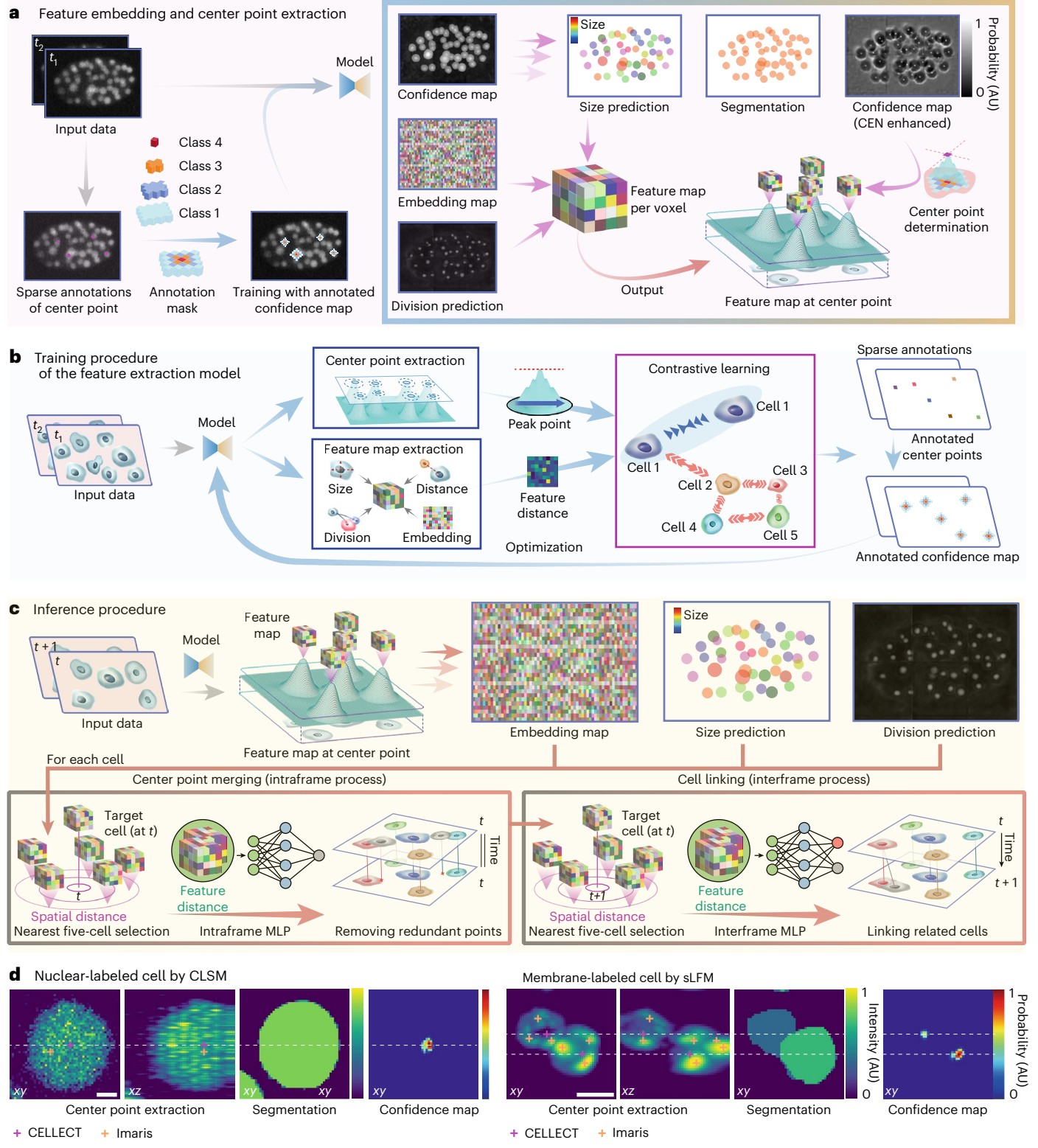

**a** Feature embedding and center point extraction

Input data | Sparse annotations of center point | Annotation mask | Training with annotated confidence map | Model

Class 4 | Class 3 | Class 2 | Class 1

Confidence map | Size prediction | Segmentation | Confidence map (CEN enhanced) | Probability (AU)

Embedding map | Feature map per voxel | Center point determination

Division prediction | Output | Feature map at center point

**b** Training procedure of the feature extraction model

Input data | Model | Center point extraction | Peak point | Contrastive learning | Sparse annotations

Feature map extraction: Size | Distance | Division | Embedding | Feature distance | Optimization

Cell 1 | Cell 1 | Cell 2 | Cell 3 | Cell 4 | Cell 5

Annotated center points | Annotated confidence map

**c** Inference procedure

Input data | Model | Feature map | Feature map at center point | Embedding map | Size prediction | Division prediction

For each cell

Center point merging (intraframe process) | Cell linking (interframe process)

Target cell (at t) | Spatial distance | Nearest five-cell selection | Feature distance | Intraframe MLP | Removing redundant points | Time

Target cell (at t) | Spatial distance | Nearest five-cell selection | Feature distance | Interframe MLP | Linking related cells | Time | t + 1

**d** Nuclear-labeled cell by CLSM

xy | xz | xy | xy
Center point extraction | Segmentation | Confidence map
+ CELLECT + Imaris

Membrane-labeled cell by sLFM

xy | xz | xy | xy
Center point extraction | Segmentation | Confidence map
+ CELLECT + Imaris

Intensity (AU) | Probability (AU)

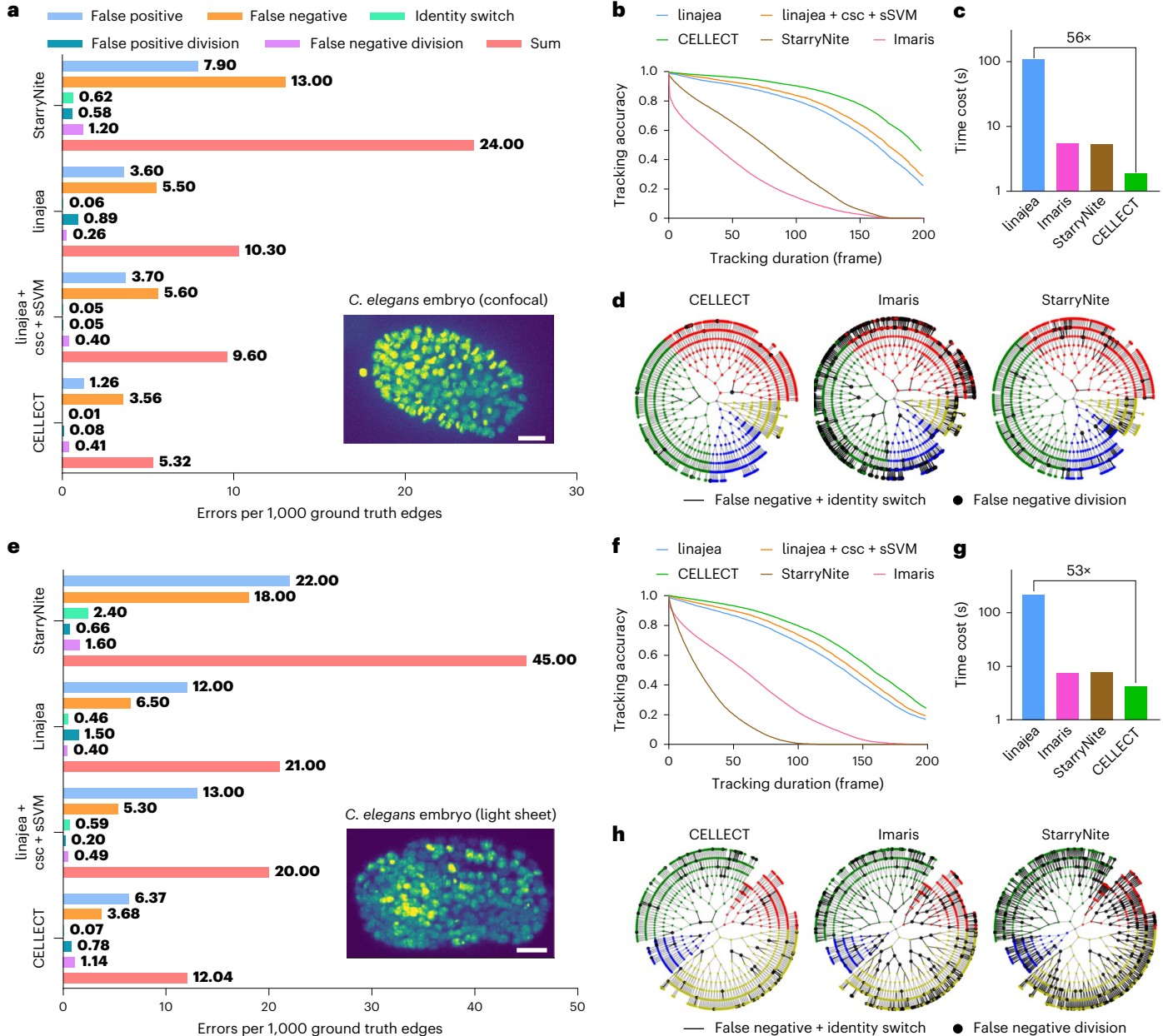

**Fig. 2 | Benchmarking of CELLECT with state-of-the-art algorithms in the Cell Tracking Challenge. a**–**d**, Performance evaluation on the mskcc-confocal dataset. **e**–**h**, Performance evaluation on the nih-light sheet dataset. **a**,**e**, The average number of errors per 1,000 ground truth edges for each error type. Errors were categorized into five conditions: false positive edges, false negative edges, identity switches, false positive divisions and false negative divisions. **b**,**f**, The proportion of error-free tracks (marked as tracking accuracy) versus tracking duration in terms of frames. **c**,**g**, Comparisons of average computational time cost for each frame among different algorithms. **d**,**h**, Hierarchical circular diagram of lineage tracing, with gray lines indicating cell division. Bold black lines and circles indicate false negative identity switch and division, respectively.

The scores for the three models other than CELLECT and the accuracy curves of linajea are derived from a previous report[35]. 'linajea + csc + sSVM' refers to an enhanced version of linajea that incorporates a csc to model biological states (for example, division, continuation, polar body) and an sSVM to learn tracking cost functions from data. All evaluations were conducted using the first 270 frames of each of the three subvolumes for both datasets (mskcc-confocal and nih-light sheet), following a cross-validation protocol. The CELLECT models used in each comparison were trained on one of the two other subvolumes. Further details on the training and evaluation splits are provided in Supplementary Tables 1 and 2. Scale bars, 10 μm.

## Quantitative benchmarking of CELLECT on public datasets

To demonstrate the advantages of CELLECT, we first conducted quantitative benchmarking with state-of-the-art algorithms in the widely accepted Cell Tracking Challenge[32]. We used two publicly available datasets of *C. elegans* developing embryos (nih-light sheet[33] and mskcc-confocal[34]), which are 3D time-lapse data captured by confocal microscopy and light sheet microscopy with only sparse annotations[35]. Using the same evaluation protocol as linajea[31], cross-validation

comparisons were conducted on the first 270 frames of these datasets (Fig. 2). Evaluation metrics included the fraction of completely reconstructed lineages over time and the number of errors per ground truth edge. The errors were categorized into five conditions: false positive edges, false negative edges, identity switches, false positive divisions and false negative divisions (Supplementary Fig. 6).

Compared with state-of-the-art tracking algorithms including linajea, Imaris and StarryNite[34,36,37], CELLECT demonstrated substantially

lower error rates across both datasets (Fig. 2a,e and Supplementary Tables 1 and 2) and achieved the top ranking in both tracking and segmentation in the Cell Tracking Challenge[32] on the Fluo-N3DH-CE dataset, an independently evaluated 3D benchmark, with comprehensive annotations that were not used in our study's experiments (team 'THU-CN (3)' on Fluo-N3DH-CE). To further quantitatively assess long-term tracking consistency with comparisons of state-of-the-art algorithms, we adopted the 'tracking accuracy' metric proposed by Malin-Mayor et al.[31], which measures the proportion of fully reconstructed cell tracks over varying temporal lengths. We applied this metric to the mskcc-confocal dataset to evaluate the fraction of correctly reconstructed tracks across trajectory lengths from 1 to 200 consecutive frames (corresponding to an imaging duration of 250 min). CELLECT achieved a tracking accuracy of 46%, more than double that of linajea (22%) and also surpassing that of the upgraded linajea + csc + sSVM[35] (30%, Fig. 2b), which is an improved version incorporating a cell state classifier (csc) to refine division modeling and a structured support vector machine (sSVM) to learn tracking weights via structured output loss. In the meantime, the computational costs of CELLECT are much lower than those of linajea and its variant. In a test using a single NVIDIA GTX 3090 graphics processing unit, CELLECT reached an average processing speed of 2 s per frame (for a dataset with a volume size of 512 × 512 × 41 voxels), 56 times faster than that of linajea (111.3 s per frame; Fig. 2c). Although Imaris and StarryNite also have low computational costs (still much higher than those of CELLECT due to the ROI selection process), they almost failed in this challenging case with large amounts of cell division (0% accuracy). With both high accuracy and high efficiency, CELLECT enables real-time 3D tracking of cells (Supplementary Video 2) and accurate continuous lineage tracing, which is critical for quantitative analysis in developmental biology to understand complicated organizations of large-scale cells (Fig. 2d,h). Furthermore, we have shown that the same pretrained model of CELLECT was compatible for different imaging modalities, achieving similar improvement over previous methods in the dataset captured by light sheet microscopy (Fig. 2e–h). This broad generalization facilitates ease of use in diverse fields.

To further characterize the functions of each module in CELLECT and parameter configurations, we conducted ablation studies and reproducibility tests on the mskcc-confocal dataset (Supplementary Figs. 7–14 and Supplementary Tables 3 and 4). The CEN module is used to refine the confidence map by concentrating values around cell centers and suppressing peripheral noise, ensuring precise predictions in noisy environments or complex tissue structures (Supplementary Fig. 7). This refinement improves cell center extraction accuracy and dramatically reduces the candidates of cell centers by using the prior that different cells will not overlap in 3D physical space. We used four levels to label the ground truth confidence map during training. If we further reduce the level number, there are many false negatives of the

tracking and the network cannot converge well to estimate the cell center (Supplementary Fig. 8). The performance gradually converges when we increase the level number to four. Building on the evaluations of the core modules, we also investigated the impact of key training configurations on tracking performance, including patch size, search mask shape and temporal resolution of the dataset. We observed that patch size had a minimal influence on overall performance, while smaller search masks facilitated more precise center localization without increasing computational costs (Supplementary Figs. 9 and 10). For temporal resolution of the input data, we designed cross-frame pairing experiments to systematically evaluate how candidate selection strategies, candidate pool size and the temporal span of training samples influence tracking robustness (Supplementary Figs. 11 and 12). The results demonstrate that selecting candidates based on spatial distance yields more stable performance and incorporating samples with longer temporal intervals and a larger candidate pool during training further improves CELLECT's generalizability across different temporal resolutions. Finally, we conducted reproducibility tests across five independent runs, demonstrating CELLECT's consistent performance under identical parameter configurations (Supplementary Figs. 13 and 14).

## CELLECT enables accurate 3D tracking of dense immune cells in GCs

Cell tracking is crucial for immunology, as it reveals how immune cells migrate, interact and function dynamically during processes such as immune surveillance, inflammatory responses and tissue-specific immunity. Uncovering normal and dysfunctional migration patterns advances our understanding of immune behaviors in diseases, aiding in therapeutic development and predictive modeling.

To rigorously validate CELLECT's advantage in challenging and dynamic cellular contexts during the immune response, we used it with our recently developed 2pSAM method[17] to track rapid dynamics of dense T cells and B cells in mouse GCs over the long term. These terabyte-scale datasets enabled a thorough assessment of CELLECT's efficient ability to handle highly dynamic, densely packed cellular populations and allowed for direct comparison with the widely used Imaris software.

To obtain the ground truth during the tracking of dense T cells, we used transgenic mice with DsRed-labeled and green fluorescent protein (GFP)-labeled OT-II T cells at a 1:5 ratio (Fig. 3a). The low phototoxicity of 2pSAM allows us to image the mouse lymph node in 3D at high speed continuously for over 5 h. Because selectively DsRed-labeled cells are relatively much easier to track in 3D by both CELLECT and Imaris, we used them as the ground truth for the dense cell-tracking task. CELLECT, pretrained on the mskcc-confocal dataset, was applied directly to this dataset without retraining and compared with the results obtained with Imaris (Fig. 3b–d). Tracking accuracy was quantitatively evaluated by comparing the trace obtained in the dense

**Fig. 3 | Three-dimensional tracking of dense immune cells in mouse lymph nodes. a**, Example orthogonal maximum intensity projections (MIP) of dual-color labeled T cells imaged by 2pSAM in mouse lymph nodes. T cells were labeled with both DsRed and GFP at a 1:5 ratio. Sparsely labeled cells are generally colabeled with GFP for use as ground truth during evaluation. **b**, Tracking result of the dense GFP channel obtained with CELLECT. **c**, Tracking result of the sparse DsRed channel obtained with Imaris and used as the ground truth. **d**, Tracking result of the dense GFP channel obtained with Imaris. **e**, Accurately tracking traces in the dense channel obtained with CELLECT (about 91.5% of all traces). We define the overlap ratio of the total tracking length compared with the ground truth trace in the sparse channel larger than 0.8 as the accurately tracked cell. **f**, Accurately tracking traces in the dense channel obtained by Imaris (about 70.7% of all traces). **g**, Comparison of the traces of the same cell in dense datasets obtained with CELLECT and Imaris. While Imaris cannot track the same cell consistently, CELLECT can track in a continuous manner. **h**, Bar chart showing comparisons of tracking accuracy under different overlap thresholds used to define accurately tracked cells. **i**, Example orthogonal MIPs of B cells in mouse

lymph nodes during the GC formation process. The white box in the GC region highlights a representative dividing demonstrated in panel (**l**) for continuous cell tracking performance of CELLECT. **j**, Three-dimensional tracking traces obtained with CELLECT with different colors corresponding to different time stamps. **k**, Spatiotemporal distribution of the events of cell division (corresponding to each dot) identified with CELLECT, indicating that a large amount of cell division happens during GC formation. **l**, Comparison of the traces of the same cell obtained with CELLECT and Imaris, indicating a cell division event. **m**, Comparison of cell proliferation and accumulated (Acc.) cell division events over time in both GC and non-GC regions. Note that the term 'sparse' in this figure refers to biological sparsity due to selective DsRed expression in a subset of cells, not to sparsity in manual annotations used during model training. All data were acquired from live imaging of mouse lymph nodes using 2pSAM. Panels **a**–**h** use dual-labeled T cell datasets, while panels **i**–**m** use B cell datasets during GC formation. All results were obtained using the CELLECT model trained on sequence 3 of the mskcc-confocal dataset. Scale bars, 100 μm.

cell channel with the trace of the same cell obtained in the sparse cell channel. We define the overlap ratio of the total tracking length larger than 0.8 as the accurately tracked cell (Fig. 3e,f). While continuous cell movement is expected in live tissue, Imaris exhibited fragmented movement trajectories, reflecting its limitations in densely packed regions (Supplementary Fig. 15). CELLECT, however, maintained consistent and

continuous tracking performance, with a considerably higher accuracy (91.5%) than Imaris (70.7%). CELLECT enables long-term stable tracking of T cells for over 30 min with smoother and longer trajectories. Representative trajectory visualizations (Fig. 3g) and quantitative comparisons (Fig. 3h) further demonstrated CELLECT's superiority in densely distributed cellular regions.

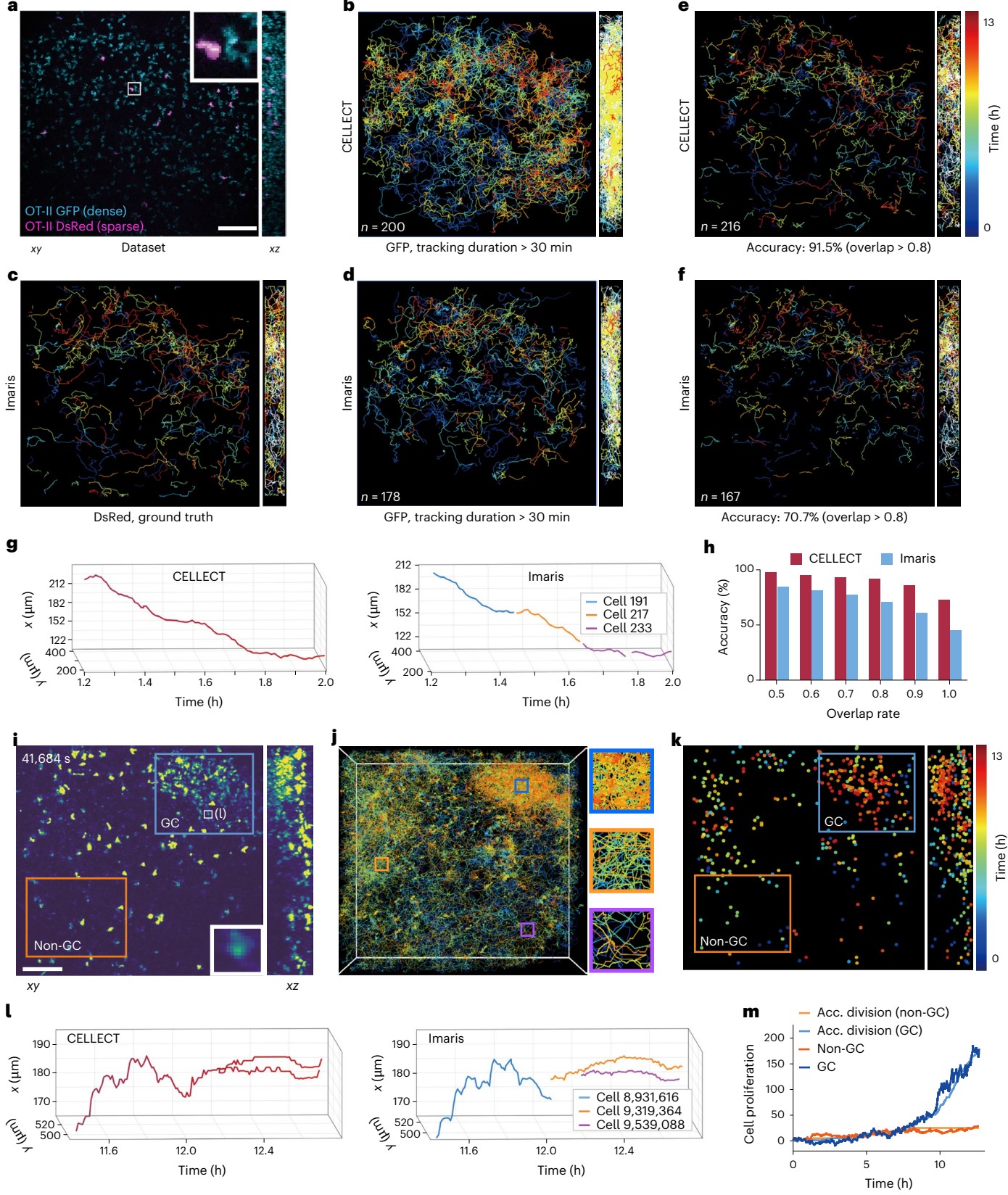

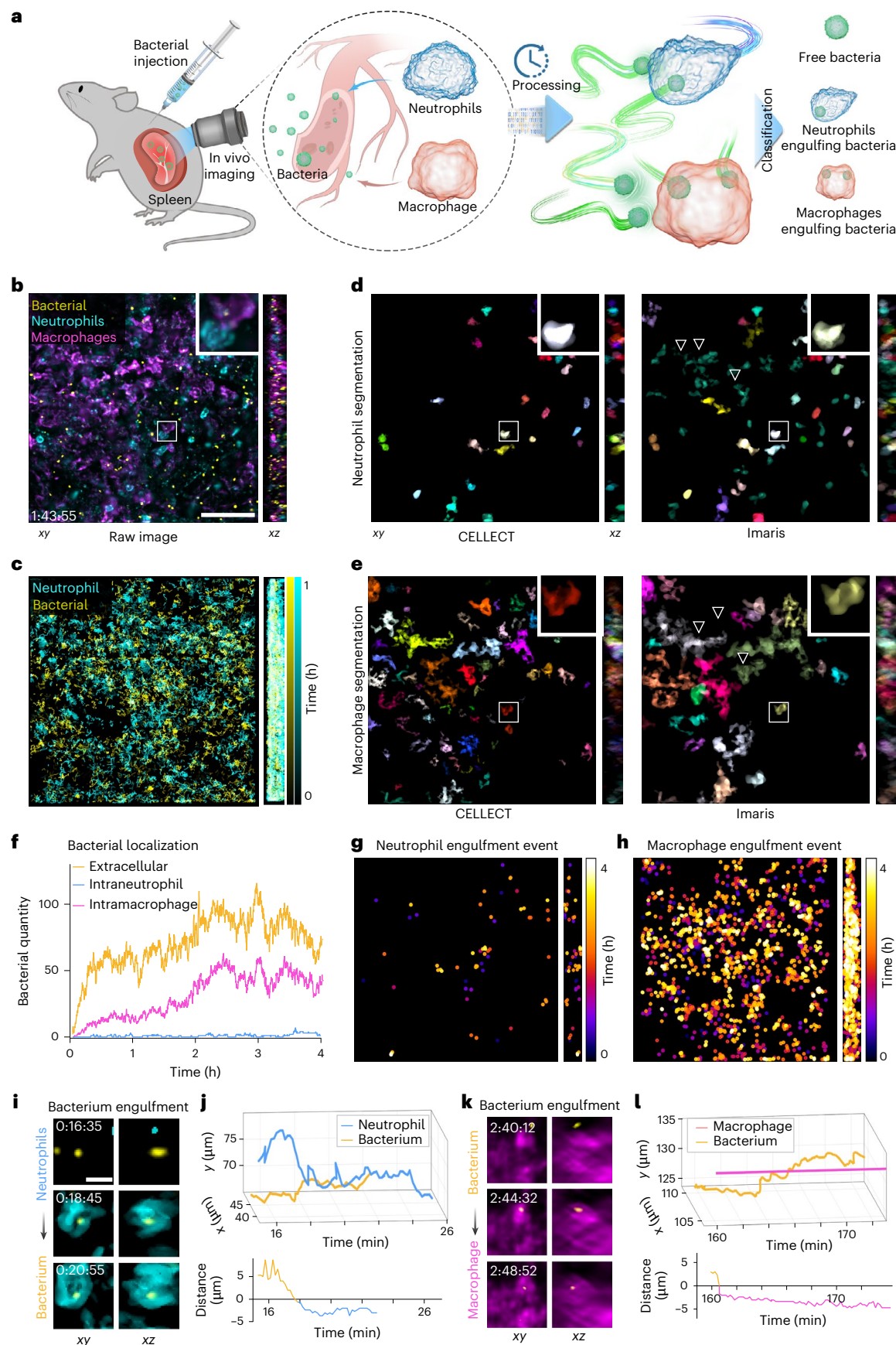

**Fig. 4 | Simultaneous tracking and segmentation of CELLECT facilitate accurate identification of immune interactions among bacteria, neutrophils and macrophages. a**, Experimental and processing schematics: bacteria were injected into the spleen of a mouse, and activities of bacteria, neutrophils and macrophages were monitored. Bacterial phagocytosis events were identified by tracking and segmentation. **b**, Example of orthogonal three-color MIP imaged in the mouse spleen. **c**, Visualization of bacterial and neutrophil trajectories extracted with CELLECT. **d,e**, Segmentation results of neutrophils and macrophages obtained with CELLECT and Imaris, respectively. For Imaris results, we adjusted intensity thresholds manually. The inverted triangles indicate distinct structures that were inaccurately segmented as neutrophils but resembled macrophages. **f**, Quantification of bacteria in different states. **g,h**, Spatiotemporal distribution of neutrophil phagocytosis and macrophage phagocytosis events. **i,k**, MIPs of bacteria engulfed by neutrophils and macrophages at different time points, respectively. **j,l**, Corresponding traces of **i,k** indicating the dynamic interaction process with the relative distance changes over time shown below. All data were acquired by live 2pSAM imaging of surgically exposed mouse spleens at 30 volumes per second over a 4-h time span. CELLECT results were obtained using the model trained on sequence 3 of the mskcc-confocal dataset. Scale bars, 50 μm (**b**–**h**) and 5 μm (**i**–**k**).

Next, we apply CELLECT to long-term tracking of large-scale B cells during the complete GC formation process. GCs are key sites of B cell proliferation and migration during immune responses, regulated by chemotactic signals[38–40]. These densely populated environments pose substantial challenges for both manual and automated tracking methods, especially due to large amounts of cell divisions during this process. Imaris, for instance, often fails to accurately capture dynamic processes such as mitosis, as its reliance on signal intensity is susceptible to errors during morphological changes, resulting in fragmented tracking paths. Using CELLECT pretrained on the mskcc-confocal dataset, we directly applied the model to a GC dataset captured by 2pSAM (Fig. 3i), comprising 1,352 frames (12.5 h) of 512 × 512 × 231 voxel data (~260 GB), without retraining. CELLECT successfully tracked over 7,000 cells persisting for more than 5 min within the imaging area (Fig. 3j). In addition to tracking, CELLECT identified cell division events and visualized their spatiotemporal distribution (Fig. 3k), revealing abundant divisions within GC regions, which cannot be detected by Imaris. Notably, CELLECT consistently reconstructed continuous cell trajectories during mitosis, whereas Imaris identified them as two new cells (Fig. 3l, Supplementary Fig. 16 and Supplementary Video 3). Validation of division event detection showed strong agreement between CELLECT-detected events and the cell proliferation curve (increased cell count minus inflow), confirming CELLECT's accuracy in both GC and non-GC regions (Fig. 3m). Moreover, CELLECT outperformed Imaris in tracking highly migratory cells (persisting >1 h and moving >50 μm), producing more extensive and continuous trajectories (Supplementary Fig. 17). In terms of computational efficiency, CELLECT demonstrated remarkable scalability. Processing this 260-GB dataset required only 157 min using CELLECT, compared to 480 min for Imaris, including 180 min for data conversion. CELLECT's lightweight architecture allowed it to run on a standard laptop with only 4 GB of RAM, whereas Imaris required nearly 100 GB of memory. Further scalability tests on four B cell datasets (total size of 4.3 TB, 1,024 × 1,024 × 301 voxels) revealed that CELLECT reduced per-frame processing time from 47 s (Imaris) to 15 s on a single NVIDIA GTX 3090 GPU with better accuracy (Supplementary Fig. 18). It may take linajea[31] about 3 months to process such a large dataset with the same computer, while CELLECT finished within 1 d.

With the dense T cell dynamics and B cell GC formation experiments, we validated CELLECT's strong generalization, robustness, accuracy and computational efficiency, highlighting its capacity to track dynamic and densely populated cellular environments with greater precision and scalability than existing methods.

## CELLECT facilitates quantitative analysis of intercellular interactions in vivo

Various biological processes involve diverse, specific and dynamic interactions among different types of cells[41]. From embryonic tissues to immune synapses, cell tracking provides the temporal continuity that links segmented snapshots into coherent narratives of cell behavior. While segmentation defines 'who' and 'where' a cell is at a given moment, tracking reveals 'which one' over time and 'when' or 'for how long' interactions occur. By capturing trajectories, divisions and contacts, cell tracking enables measurement of directed migration, observation of fate transitions and identification of signaling-mediated targeting or repeated engagements that define dynamic cell-to-cell interactions in vivo. To effectively study dynamic cell-to-cell interactions, it is essential to visualize cell boundaries and track their movements[42–45]. Fluorescent labeling of cell membranes enables clear delineation of boundaries, facilitating the detection of interaction sites and morphological changes. However, the fluorescence signal at the membrane is often weak due to its limited volume, and, in small, fast-moving cells such as bacteria, tracking becomes even more challenging. Therefore, analysis tools capable of simultaneous segmentation and tracking with high performance are crucial for capturing these complex, real-time cellular interactions within living tissues. CELLECT fills in this niche.

To demonstrate CELLECT's capability in quantitatively analyzing intercellular interactions, we performed multichannel in vivo imaging of bacteria, neutrophils and macrophages in the mouse spleen. Fluorescently labeled bacteria were intravenously injected into mice, and three-channel volumetric imaging of the spleen was continuously conducted using 2pSAM[17] for over 4 h (Fig. 4a,b). The CELLECT model pretrained on the mskcc-confocal dataset was directly applied for tracking bacteria and immune cells without retraining. During imaging, neutrophils exhibited active motility, whereas macrophages remained largely stationary.

CELLECT enabled simultaneous extraction of both bacterial and immune cell trajectories as well as accurate cell segmentation. In Fig. 4c, we visualize not only the bacterial trajectories but also the motion paths of neutrophils and their spatial overlap, offering a clear representation of dynamic cell–pathogen interactions. The segmentation results of neutrophils and macrophages generated with CELLECT are shown in Fig. 4d,e, with comparisons to Imaris. Notably, Imaris results were highly sensitive to threshold tuning, while CELLECT did not require manual parameter adjustment (Supplementary Fig. 19).

**Fig. 5 | Tracking of neural activities at single-cell resolution during strong tissue deformation. a**, MIP of soma-targeted GCaMP6s-expressing neurons in the *Drosophila* brain. Scale bar, 50 μm. **b**, Temporal-coded MIP across all frames (14,800 frames, corresponding to 6,413 s), showing large deformations of the brain during imaging. **c**, Heatmap of neural activity traces for neurons detected in the first frame obtained by different tracking methods (top, CELLECT; middle, Imaris; bottom, TrackMate). A representative trace for ROI 266 is depicted at the bottom. **d**–**f**, Trajectories and neuronal activity traces of three representative neurons obtained with CELLECT, with and without ROI tracking, within the selected time range from 433 to 3,464 s. **g**–**i**, Distribution of fully tracked (yellow) and partially tracked (purple) neurons obtained by three methods: CELLECT (**g**), Imaris (**h**) and TrackMate-Weka (**i**). The number of neurons tracked in the first frame of the video is indicated as *n*, excluding those that appeared in later frames. Note that we manually connected broken tracks obtained with Imaris and TrackMate-Weka with nearby new tracks to improve their tracking continuity and accuracy. **j**, Number of tracked neurons over time frames obtained by different methods. **k,l**, Comparison of movement tracking (**k**) and calcium activity tracing (**l**) for the same neuron by different methods. All data were acquired from live 2pSAM imaging of GCaMP6s-expressing *Drosophila* brains during spontaneous activity. CELLECT results were obtained using the model trained on sequence 3 of the mskcc-confocal dataset. Scale bars, 50 μm (**a**,**b**,**g**–**i**) and 5 μm (**d**–**f**,**k**).

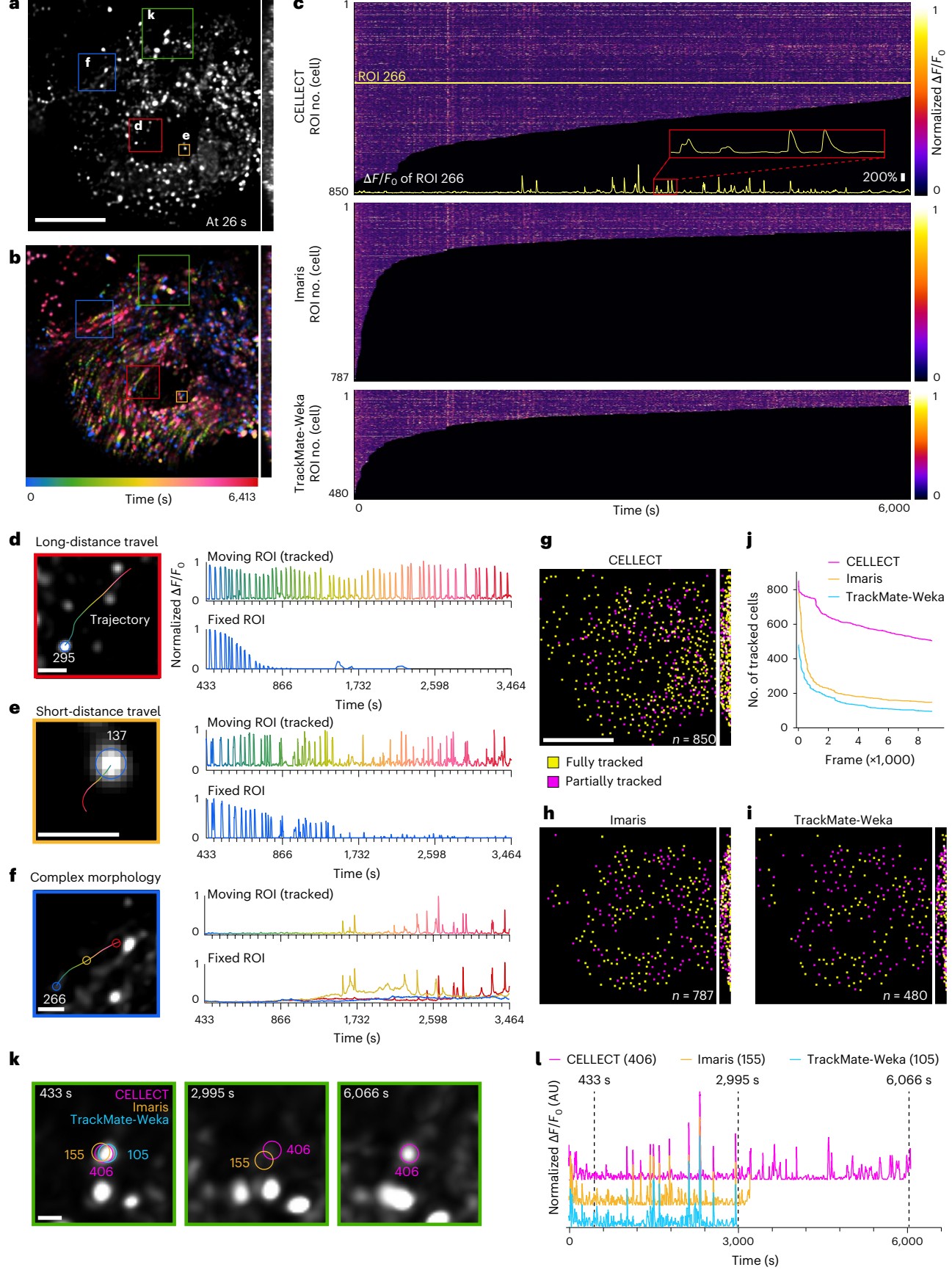

With reliable segmentation and tracking of both cells and bacteria, CELLECT supports automated classification of bacterial phagocytosis events based on 3D spatial colocalization between cells and bacteria (Fig. 4f–h). In Supplementary Fig. 20, we present a representative example of simultaneous segmentation and tracking, showing a single neutrophil chasing and engulfing multiple bacteria while undergoing dynamic membrane deformation. We further analyzed representative cases of phagocytosis identified automatically by tracking and segmentation results: one involving active migration and engulfment by neutrophils and the other involving macrophages capturing bacteria drifting in the extracellular environment (Fig. 4i–l). Spatiotemporal trajectories and relative distances were quantitatively extracted to characterize the two interaction patterns. The results reveal distinct immune strategies: neutrophils tend to pursue motile bacteria, whereas macrophages capture passively transported bacteria. Overall, macrophages engulfed more bacteria in the spleen, which may be attributed to their higher density or broader capture radius (Fig. 4f).

### Robust 3D calcium extraction during strong tissue deformation

Neuronal activity signals provide insight into how an organism processes different stimuli, making them a crucial source of information for the study of brain network studies[46]. Typical observation of neuronal activities is performed by recording and analyzing fluorescence signal intensity changes of neurons expressing calcium indicators in fixed positions. In the case of rigid tissue, most motion artifacts from the animal under study, which induce baseline fluctuation, can be corrected by rigid registration algorithms. However, when the neurons migrate or deform nonuniformly during data recording, conventional intensity-based tracking methodologies become inaccurate due to the fluctuation of calcium signals. CELLECT can solve this problem by identifying the same cell in the latent embedding.

We tested our method by observing actively firing neurons during the deformation process of *Drosophila* brain tissue as a case study, evaluating its efficacy compared to conventional tracking software such as Imaris or TrackMate (Fig. 5 and Supplementary Video 4). We imaged soma-targeted neural responses across a large volume in the *Drosophila* brain with high spatiotemporal resolution of 2pSAM with low phototoxicity for approximately 100 min (Fig. 5a and Supplementary Video 4). We could observe dynamic and heterogeneous deformation of brain tissue occurring continuously over the long term (Fig. 5b). Using CELLECT, we tracked and obtained reliable calcium activity traces of 451 cells of the 851 cells detected in the first frame of the dataset over 14,800 frames (Fig. 5c–f). Notably, a representative comparison of manual and automated tracking results for calcium activity of neurons demonstrated the importance of CELLECT in capturing dynamic cellular movement (Fig. 5d–f). Additionally, CELLECT had tracking accuracy more than threefold higher than that of Imaris or TrackMate (Fig. 5g–j). Both Imaris and TrackMate (with gap frames set to exceed 2,000 frames) performed poorly in tracking intermittent and moving neuronal activity signals. To improve the tracking continuity in these two methods, we manually connected broken tracks to the most relevant neighboring tracks. We further compared the activity traces of the same neuron obtained by the three methods (Fig. 5k,l). Notably, CELLECT showed extended tracking duration and consistency in baseline fluctuation compared to the other two methods, even when the neuron was inactive. These advantages in both performance and efficiency enable robust calcium extraction of neural activities even in highly dynamic environments.

### Discussion

In this study, we proposed a contrastive embedding learning framework, named CELLECT, to achieve accurate, efficient and quantitative cell tracking and segmentation across various biological species and applications. Compared to previous methods, CELLECT achieved state-of-the-art accuracy and a processing speed over 50 times faster for in vivo 3D time-lapse datasets with broad generalization across varying cell shapes, signal intensities and resolutions.

The integration of contrastive learning in the CELLECT network algorithm facilitates robust differentiation and quantification of cellular morphology and behavior. Consequently, a CELLECT model pretrained on the public dataset with sparse annotations can be widely applied to various biological study scenarios without the requirement of retraining. Building on this, CELLECT also supports human-in-the-loop correction by providing additional reliability scores for cell linkage and identifying unstable detections, such as cells that vanish prematurely or appear abruptly. Even for challenging or unseen datasets, manual refinement by assigning higher reliability scores can make CELLECT focus more on the new dataset, further improving usability and adaptability. Self-supervised learning strategies can also be exploited to fully avoid annotations. In this case, the framework can be further scaled up as a foundation model for cell tracking and segmentation by including more datasets in the future. Currently, CELLECT uses only two temporally adjacent frames as input to maximize efficiency, which limits its capacity to capture long-term temporal relationships. While this input strategy offers a degree of tolerance to low temporal resolution of the imaging data, tracking accuracy may still degrade when critical events such as intercellular motion, collisions or cell division occur between frames. We therefore recommend that the temporal resolution of the dataset should match or exceed the timescale of the cellular behavior under investigation. To address this limitation, future extensions may incorporate motion priors, global optimization or trajectory modeling to improve temporal coherence across frames. Another limitation lies in the patch-based processing of CELLECT: when the target cells are considerably larger than those seen during training and occupy most of the patch, the model may produce more redundant detections and increase the risk of linking errors. In these cases, we can downsample the input images to maintain model performance. CELLECT includes a built-in size estimation module that can support adaptive scaling strategies to help address this issue. Remaining tracking errors of CELLECT are mainly concentrated in challenging conditions, such as low signal-to-noise regions, degraded axial resolution and densely packed cell clusters in late-stage development. These scenarios are also difficult for manual annotations as ground truth, and improving CELLECT's discriminative power under such conditions will be an important direction for future work.

Nevertheless, with the ongoing rapid advancements in imaging technology, we can now achieve high-speed 3D intravital observations of large-scale cells across the whole organ and extended durations[16,47]. While these instruments provide a direct visualization of intercellular and intracellular interactions during different pathological and physiological states, CELLECT fills in the gap to analyze such high-throughput datasets with high accuracy, which is critical for quantitative understanding of the basic physics in the living system across multiple scales. By efficiently extracting various aspects of quantitative cellular dynamics, including division events, lineage identity, intercellular interactions, motility, proliferation kinetics and responses to environmental changes, we believe that CELLECT will prompt broad discoveries in diverse fields such as developmental biology, pathology, immunology, neuroscience and pharmacology.

### Online content

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

## Methods

### Mouse experiments

All mouse experiments were performed on C57BL/6J mice (*Mus musculus*, 6–8 weeks old), without consideration of sex, according to the animal protocols approved by the Institutional Animal Care and Use Committee of Tsinghua University. All procedures complied with the standards of the Association for Assessment and Accreditation of Laboratory Animal Care, including animal breeding and experimental manipulations. Mice were housed in standard cages with a maximum of five mice per cage, under a 12-h reverse dark–light cycle and at an ambient temperature of 72 °F and a relative humidity of ~30%. Food and water were provided ad libitum.

### Model architecture

We employed a 3D U-Net architecture[48] as the backbone, in which the downsampling operations use stride-2 convolutions to preserve feature granularity and spatial consistency (Supplementary Fig. 1). The primary model architecture is designed to process two consecutive imaging frames as input. The network has two output branches. The first branch outputs a confidence map for the first frame indicating the probability of being a cell center, a 64-channel latent embedding vector for each voxel and a probability map predicting whether a cell undergoes division. These outputs leverage spatiotemporal differences between consecutive frames to extract cell center-specific features and provide coarse estimates of cell size (radius), without relying on boundary-based or shape-dependent information. The second branch incorporates a lightweight 3D U-Net referred to as the CEN. The CEN sharpens the confidence maps by concentrating probability values near the cell centers, reducing the influence of intensity variations and ensuring accurate localization of cell center points. The outputs from both branches are merged to achieve precise cell center localization, segmentation and feature extraction.

To refine the tracking process, the architecture integrates two MLPs: an intraframe MLP and an interframe MLP (Supplementary Fig. 5). Each MLP takes the embedding of a target cell along with comparative features computed against five candidate cells with the nearest spatial distances as input. The obtained features, including embedding vector offsets, spatial distances, estimated size differences and changes in predicted division probabilities, were subsequently input into the intraframe and interframe MLP networks for further classification.

The intraframe MLP outputs six similarity scores: five corresponding to the candidate cells and one representing 'none', indicating that the target cell is not matched with any of the candidates. This 'none' score handles cases in which the cell disappears or moves out of the field of view. The interframe MLP not only outputs six similarity scores (five corresponding to the candidate linking cells and one representing 'none') but also outputs a prediction of whether the target cell is undergoing division. The detailed input–output structure of the two MLPs is shown in Supplementary Fig. 5 and Supplementary Table 3. This design enables the model to flexibly handle passive drifting of dying cells, transient cell fragments and field-of-view transitions through learned similarity scores and dynamic track assignment.

The difference between the two MLPs lies in their respective roles and operating contexts. The intraframe MLP functions within a single frame to suppress redundant or overlapping cell center predictions. It treats any two objects in the same frame as distinct cells, as long as they appear at the same time. And the MLP will attempt to remove these redundant identifications of the same cell. The interframe MLP is used to link cells between adjacent frames. It uses the division prediction to decide whether a cell at time *t* should be linked to one or two cells at time *t* + 1. In this case, daughter cells resulting from division are treated as part of the same lineage and grouped together during linking.

These two MLPs are intentionally implemented as separate modules because they rely on different input structures and perform distinct tasks. Separating them from the core feature extractor

also enables efficient patch-wise inference, which is important for large-scale 3D datasets. If the MLPs were integrated into the main backbone, handling linkages across patch boundaries would become much more difficult. Compared to sequence-based models that require fixed-length temporal inputs and long sequences, this modular and lightweight design provides better generalization across varying frame intervals and temporal resolutions with reduced computational costs.

To further address the limitations of training with only sparse annotations, we adopt a spatial distance-based candidate selection strategy, leveraging previous knowledge that cells in closer proximity to the target cell are more likely to be correct linkage candidates. As shown in Supplementary Fig. 11, this spatial proximity-based approach exhibits better robustness and stability, particularly in high-density conditions or late-stage embryonic datasets with frequent mitotic events than feature distance-based selection. While feature distance-based candidate selection performs well in low-density regions, it becomes less reliable under challenging scenarios due to the sparsity of supervision in the embedding space, which often leads to incorrect associations.

Moreover, the parameter for the 'number of candidate cells' does not need to be fixed at five, the default parameter in CELLECT, and can be flexibly adjusted based on the frame rate and cell density of the dataset. The default configuration of selecting five candidate cells provides a good trade-off between computational efficiency and adequate coverage of potential division events. We demonstrate that five candidates are sufficient for datasets with a common frame rate (Supplementary Fig. 12). However, in data that have a low imaging frame rate and in which cells undergo larger displacements between frames, expanding the candidate pool to ten can improve matching coverage and reduce false negatives in densely populated regions.

### Center enhancement network

The CEN employs a simplified and effective method for center point extraction, which directly identifies local maxima within a designated search mask zone, bypassing the complexities of traditional nonmaximum suppression[49,50].

For training the center point extraction module, a 7×7×7 region surrounding the labeled center point is selected. The voxel at the exact center point is labeled as level 4, with adjacent voxels progressively categorized into levels 3, 2, 1 and 0 based on increasing distance. This creates five distinct levels. During training, the model learns to distinguish these levels, accumulating scores across them using the following loss function:

$$\mathrm{Loss_{center}} = \sum_{i=1}^{4}\sum_{j=i}^{4} \mathrm{CrossEntropy}\left(m(a,j), n(b,i)\right)$$

$$m(a,j) = \mathrm{concat}\left(\max\left(a_0, a_1, \cdots, a_{j-1}\right), a_j\right)$$

$$n(b,i) = \begin{cases} 1 & b \geq i \\ 0 & b = 0, \end{cases}$$

where a represents the predicted scores corresponding to the four levels from 1 to 4 and b denotes the label function, which represents the manual division into four levels. The function $m(a,j)$ takes its maximum value in the range $(0, j-1)$ when calculating the score for each level; subsequently, this value is compared with $a_j$ at the *j*th level. The function $n(b,i)$ is designed for selecting the computational area. Here, areas in *b* with a value greater than or equal to *i* are identified as positive samples and those with a value equal to 0 are treated as negative samples. Values not meeting either of these criteria are excluded from the loss calculation.

The choice of search mask size considerably impacts performance, as confirmed by ablation studies (Supplementary Fig. 10). Larger masks increase false negatives due to difficulty isolating individual center

points, while smaller masks raise false positives by capturing multiple peaks within a cell. This balance is particularly sensitive to axial resolution differences, which influence mask effectiveness. The $5\times5\times3$ search mask achieves the best trade-off, accommodating anisotropic resolutions while maintaining high accuracy.

The level configuration also plays a crucial role in CEN performance (Supplementary Fig. 8). Reducing the number of levels, especially higher levels such as 3 and 4, weakens the confidence gradient around the center point, resulting in multiple redundant peaks within the same cell. This increases the difficulty of linking cells across frames and leads to reduced tracking accuracy. Conversely, including all levels from 1 to 4 maintains robust confidence distributions and ensures precise center localization.

Compared to Gaussian mask and nonmaximum suppression approaches, such as those used in linajea, the CEN directly prioritizes accurate center localization without relying heavily on temporal integration. By selecting the most prominent peaks and avoiding redundant predictions, the CEN enables more efficient feature extraction and contrastive learning. This approach ensures robust performance in datasets with dense cell populations and anisotropic resolutions, making it highly effective for real-world biological imaging scenarios.

We compared structural variations of the framework: CELLECT with CEN, and CELLECT without CEN (Supplementary Fig. 7). Results indicate that incorporating the CEN improves tracking performance under the current output settings. The CELLECT-with-CEN configuration had lower false negative rates and demonstrated more stable overall performance. We further conducted a quantitative ablation study (Supplementary Table 4) demonstrating that CELLECT's core modules, such as CEN, intraframe MLP and interframe MLP, are highly interdependent. Thus, disabling any of the modules substantially degrades tracking accuracy and stability, confirming their essential roles in reliable cell tracking.

## Contrastive learning

Contrastive learning is a training approach that focuses on quantifying similarities among objects. Unlike conventional methods that require the explicit specification of several levels, this method permits category determination based on distances between feature vectors of objects. This approach has already found extensive application in tasks such as face recognition and reidentification[51], which require adaptation to unknown categories.

The core principle of contrastive learning involves minimizing the discrepancies between an object's feature vector and those of other objects belonging to the same level while maximizing differences from the feature vectors of objects belonging to different levels. In this project, we extracted a 64-dimensional embedding vector for each cell center point. By applying the triplet loss optimization methodology, we ensured that each cellular feature vector was uniquely distinguishable with respect to the Euclidean distance.

$$\text{Loss}_{\text{triplet}} = \sum_{i}^{N} \left( \|f(\mathbf{x}_i) - f(\mathbf{x}_i^{\text{p}})\|_2^2 - \|f(\mathbf{x}_i) - f(\mathbf{x}_i^{\text{n}})\|_2^2 + \alpha \right)_+ ,$$

where $f(\cdot)$ refers to the model function; $\mathbf{x}_i$ denotes the $i$th center point; $\mathbf{x}_i^{\text{p}}$ represents a positive sample, namely, a sample from the same cell as the $i$th center point; and $\mathbf{x}_i^{\text{n}}$ signifies a negative sample, that is, a sample from a different cell than the $i$th center point. The '+' symbol on the lower right indicates that only values greater than zero in the equation are summed. $\alpha$ is a constant pertaining to the distance boundary, which was set to 0.3. In our study, we aimed to refine the feature distances both for the same cell between consecutive frames (one versus one) and for divided cells (one versus two) relative to other cells. The obtained features were subsequently input into an MLP network for further classification.

## Training and inference

The training process of CELLECT is designed to refine feature representations, optimize confidence maps and classify cell relationships accurately. This modular pipeline integrates multiple objectives, ensuring robust tracking and segmentation across diverse datasets.

To achieve effective feature separation, the model employs a contrastive learning framework. The optimization process minimizes feature distances within the same cell while maximizing distances between different cells. This is formalized as:

$$E_{\mathbf{x}\approx p(\mathbf{x})} \left( \max_{\mathbf{y}^+\approx p(\mathbf{y}|\mathbf{x})} \varphi(\mathbf{x}, \mathbf{y}^+) - \min_{\mathbf{y}^-\approx q(\mathbf{y}|\mathbf{x})} \varphi(\mathbf{x}, \mathbf{y}^-) \right)_+ .$$

Here, $E$ represents the expectation, averaging the loss across all labeled sample points $\mathbf{x}$, while $\varphi(\mathbf{x}, \mathbf{y})$ measures the similarity between the feature vectors of $\mathbf{x}$ and $\mathbf{y}$, typically computed as the Euclidean distance. Positive samples $\mathbf{y}^+ \approx p(\mathbf{y}|\mathbf{x})$ are defined as points originating from the same cell or dividing cells, while negative samples $\mathbf{y}^- \approx q(\mathbf{y},|,\mathbf{x})$ represent points from distinct labeled cells. The $\max(\cdot)$ operator in this context selects the most distant positive sample for a given input $\mathbf{x}$, ensuring that the training process focuses on maximizing the separation of the hardest-to-cluster positive pairs within the same cell. Conversely, the $\min(\cdot)$ operator selects the nearest negative sample for $\mathbf{x}$, which corresponds to the most challenging negative pair near the decision boundary. By optimizing for these specific positive and negative sample pairs, the model achieves tighter clustering within the same cell while simultaneously pushing apart distinct cells in the feature space.

For training, cross-entropy loss was employed to optimize segmentation, division probability prediction and MLP classification tasks, while the mean absolute error loss was used for cell size assessment. This combination of loss functions allowed the model to address diverse objectives simultaneously, ensuring consistent performance across tasks.

The model was trained on the mskcc-confocal dataset for 149 epochs, using the first 270 frames for training and reserving 50 frames for testing. Each epoch consisted of 400 steps, with random patches sampled from the training frames and 130 samples randomly selected from the testing frames. Input data were normalized based on the specific dataset. For datasets with fixed intensity ranges, such as nih-ls (0–255), no additional preprocessing was required. For datasets such as mskcc-confocal, which lack a fixed upper intensity bound, a logarithmic transformation was applied as input = log (input + 1).

To ensure stable optimization, MLP modules were selectively trained during specific epochs (2–7, 15–20 and 35–40), with their parameters frozen during the remaining epochs. This selective training approach minimized interference among tasks, enabling focused optimization. The Adam optimizer with a learning rate of 0.0001 was used for training, and no additional data augmentation was applied.

During inference, the model processes large images using a block-wise strategy. Images are divided into partially overlapping patches, which are processed in parallel. Using the confidence maps and search masks, cell center points are identified, retaining only the feature information corresponding to these points for downstream classification. This distributed processing strategy enables scalability and efficiency for large-scale datasets. The reproducibility of the approach was validated in five independent training runs, as shown in Supplementary Figs. 13 and 14, with consistent results demonstrating reliable, low-error tracking on the mskcc-confocal dataset.

By combining modular training with a robust optimization framework, CELLECT ensures scalable, accurate tracking and segmentation under diverse imaging conditions. The integration of contrastive learning enhances feature separation, while selective training of MLP modules and efficient inference strategies improve the overall robustness and adaptability of the model.

## Temporal linking with larger intervals

CELLECT is not limited to linking cells between strictly adjacent frames. As shown in Supplementary Fig. 14, even when trained only on datasets with the original frame rate, CELLECT maintains stable tracking performance across longer intervals, such as linking from frame $t$ to $t + 2$ or $t + 3$, in rapidly developing sequences such as *C. elegans* embryogenesis. While the majority of trajectories remain stable, a moderate decrease in performance is observed in late-stage sequences characterized by high cell density and frequent mitotic events. To further investigate this effect, we evaluated the impact of varying temporal downsampling during training and different settings for selecting the number of candidate cells (Supplementary Figs. 11 and 12). The results indicate that CELLECT performs reliably under common frame rate conditions using five spatial distance-based candidate cells. For lower imaging frame rate conditions, incorporating multitemporal training samples and increasing the number of spatial candidates improve robustness by reducing false negatives in crowded, actively dividing regions.

In addition to extended cross-frame inference, CELLECT supports postprocessing optimization strategies that operate on previously inferred trajectories. These include global trajectory planning, reconnection of fragmented tracks and pruning of redundant branches. For example, in the *Drosophila* neuron-tracking task (Fig. 5), in which signal dropout is common, we applied a ±200-frame temporal window to reconnect fragmented tracks based on feature similarity and spatial continuity. This strategy improves trajectory completeness without large computational costs. For consistency, Imaris and TrackMate were also configured with a maximum 2,000-frame temporal linking window, following the recommended settings.

These two complementary strategies, extended temporal matching during inference and postprocessing optimization based on trajectory structure, enhance CELLECT's adaptability to a wide range of temporal resolutions and imaging conditions. It is important to note that these procedures were applied exclusively to the *Drosophila* neuron dataset (Fig. 5). All other results presented in this work were obtained using CELLECT's default frame-by-frame inference without any additional temporal or postprocessing mechanisms.

## Segmentation and radius estimation

While CELLECT does not rely on dense segmentation masks, it provides coarse segmentation and radius estimation based solely on sparse center point annotations. During training, approximate cell sizes were derived from radius values associated with each annotated center (for example, in the mskcc-confocal dataset) or from silver-standard masks in datasets such as Fluo-N3DH-CE. These estimates were used to define local neighborhoods for contrastive learning, where voxels within a specified radius were treated as belonging to the same cell.

The model learns to encode instance-specific spatial embeddings by minimizing intra-cell distances and maximizing inter-cell distances within local neighborhoods. This results in soft but biologically coherent segmentation maps around center points, even in the absence of boundary-level supervision. Although CELLECT does not explicitly optimize precise boundary delineation, the resulting coarse segmentation remains highly informative for downstream tasks such as cell–cell interaction analysis, as demonstrated in Fig. 4.

Supplementary Fig. 3 further illustrates the smooth clustering of embedding vectors around the center points, and Supplementary Fig. 4 shows how spatial convolution of the embedding map helps delineate cellular boundaries in an unsupervised fashion. We note that the grid-patterned artifacts observed in Supplementary Fig. 4 result from the independent inference performed on each patch. However, these artifacts do not impact performance, as only the embedding vectors at the cell centers are used.

## Comparison with embedding-based and contrastive learning methods

CELLECT shares conceptual similarities with previous instance segmentation frameworks that use learned embeddings to distinguish object instances, such as that of Neven et al.[52,53]. Their model predicts pixel-wise spatial embeddings and applies clustering as a postprocessing step to derive object masks. However, this approach requires dense, per-pixel supervision and is designed for natural images with full segmentation annotations. By contrast, CELLECT is tailored for sparse annotation settings typical of 3D microscopy, in which only center points are available. Our architecture directly learns object-level representations from these sparse annotations without relying on full-resolution masks or clustering. In short, Neven's model learns centers from full images; CELLECT learns object structure from sparse centers.

We also compare CELLECT with the recent contrastive learning approach of Zyss et al.[28], which focuses on lineage reconstruction from fully segmented cells. Their framework is optimized for low-imaging-frame rate datasets lacking trajectory labels and infers division events from appearance-based cell similarity. CELLECT, by comparison, is designed for high-resolution 3D time-lapse microscopy, for which accurate cell identity tracking is required in dynamic environments with frequent mitosis. While both methods leverage contrastive learning, Zyss et al. aim to recover lineage trees, whereas CELLECT emphasizes short-term identity association and frame-to-frame correspondence.

In sum, these methods differ from CELLECT in annotation regime (dense masks versus sparse centers), learning objective (lineage inference or instance segmentation versus tracking) and data modality (two-dimensional natural or static bioimages versus dynamic 3D microscopy). We view these approaches as complementary: each addressing different challenges in the broader problem of cell analysis.

## Evaluation with the Cell Tracking Challenge benchmark

To evaluate CELLECT's generalizability on a standardized third-party test set, we submitted our method to the Cell Tracking Challenge under the team named 'THU-CN (3)'. Please note that the number (3) in the name THU-CN means that there are two earlier submissions under the same institution that were actually submitted several years ago by other research teams at Tsinghua University and not our group. On the Fluo-N3DH-CE dataset, CELLECT achieved overall performance scores of 0.853 in the segmentation benchmark and 0.850 in the tracking benchmark, indicating the highest performance in both tasks.

We selected the Fluo-N3DH-CE dataset because it is currently the most suitable 3D time-lapse dataset in the Cell Tracking Challenge that provides complete training annotations, including cell center positions, trajectories and mitosis events. Its annotation structure and biological context align well with our training design, and it stands out as a rare 3D dataset within the Cell Tracking Challenge that supports comprehensive and integrated evaluation across segmentation, mitotic event detection and tracking metrics.

While the official training subset of Fluo-N3DH-CE was used to prepare the Cell Tracking Challenge submission, this dataset was not involved in any training or evaluation experiments described in the main text. All results were independently evaluated by Cell Tracking Challenge organizers using hidden ground truth annotations. The full results are available at https://celltrackingchallenge. net/latest-csb-results/ and https://celltrackingchallenge.net/latest-ctb-results/.

## Comparative methods and parameter settings

To ensure fair and reproducible benchmarking, we evaluated CELLECT against representative and widely used tracking methods. For Imaris (Bitplane, version 9.0.1), we used the 'Spots' detection and 'Tracking' modules. Estimated spot diameters were manually adjusted based on voxel size for each dataset. Gap closing was enabled with a maximum

of two to three frames for standard datasets and extended to 2,000 frames for neuronal sequences to accommodate intensity flickering. For TrackMate[54] (via ImageJ, version 7.10.1), we used the provided pre-trained Weka[55] model detector applied in a slice-by-slice manner. The gap-closing interval was set to 2,000 frames for neuronal recordings, consistent with the Imaris configuration. For StarryNite[34,36,37] (2020 release), we adopted the default parameter configuration distributed with the 2020 version of StarryNite and modified the cell radius range settings based on the ground truth annotation files.

### C. elegans embryo public dataset

The data used in the quantitative benchmarking of CELLECT were obtained from two public imaging datasets of *C. elegans* embryo development, acquired using a confocal microscope (mskcc-confocal, https://doi.org/10.5281/zenodo.6460303) and a light sheet microscope (nih-ls, https://doi.org/10.5281/zenodo.6460375), respectively. The confocal dataset contains three anisotropic 3D time-lapse series, while the light sheet dataset contains three isotropic 3D time-lapse series, both featuring sparse cell annotations for all cells. These datasets were employed to evaluate the performance metrics of CELLECT based on the provided ground truth. Each dataset contains three sequences, and each sequence comprises a single embryo imaged over 400 frames with sparse center point annotations.

We adopted a threefold cross-validation protocol following the linajea framework. Specifically, for each dataset, the first 270 frames of each sequence were used. In each round, two sequences were used for training and validation and the remaining one was reserved for testing. This setup ensures strict separation between training and testing data and allows fair comparison with previous work. All other experiments in the study also used only the first 270 frames from each sequence. The results presented in Fig. 2 were obtained using this cross-validation protocol. Detailed evaluation results for each fold are reported in Supplementary Tables 1 and 2.

### Dual-channel T cell imaging in lymph nodes

To achieve simultaneous sparse and dense labeling of the same T cells across different fluorescence imaging channels, we used transgenic mice coexpressing OT-II DsRed and OT-II GFP at a ratio of 1:5. The sparsely labeled DsRed channel, tracked using Imaris, was used as the gold standard to evaluate tracking performance of the same cells in the densely labeled GFP channel. We used a two-photon microscope to acquire dual-channel imaging datasets at 30 Hz per plane across 50 planes. For an improved signal-to-noise ratio, results from every 20 frames were averaged, resulting in a final frame interval of 33.33 s and a 2.5-h 3D time-lapse series dataset. The imaging area covered 688 μm × 688 μm × 100 μm.

### B cell imaging during germinal center formation

To observe the active behaviors of B cell during GC formation, a surgical procedure for intravital imaging of the inguinal lymph node was performed. Mice were anesthetized with a mixture of isoflurane and oxygen, the surgical area was shaved, and residual hair was removed with depilatory cream. Following established procedures[56,57], a stable imaging window of the lymph node was prepared. The mouse was then securely fixed under the stage of a 2pSAM instrument for imaging[17]. Intravital imaging was performed over 12 h with a frame interval of 33.90 s per two planes, with each plane covering a thickness of 100 μm. The image area covered 688 μm × 688 μm × 200 μm.

### Imaging of tripartite interactions among bacteria, neutrophils and macrophages

To investigate tripartite interactions among bacteria, neutrophils and macrophages, we used *Streptococcus pneumoniae* (TH870, serotype 6A) expressing GFP, neutrophils labeled with PE-Cy5-conjugated anti-Ly6G antibody and splenic red pulp macrophages labeled with PE-conjugated

anti-F4/80 antibody. For intravital imaging, the spleen was exposed, stabilized using an adsorption pump and observed through an 8-mm coverglass window under anesthesia using 2pAM. Thirty minutes after tail vein injection of $10^7$ CFU of GFP-expressing bacteria, 3D imaging was performed at 30 volumes per second, covering the volume of 229 μm × 229 μm × 25 μm.

### *Drosophila* brain imaging under tissue deformation

Flies expressing pan-neuronal GCaMP6s (nSyb-Gal4, UAS-nls-GCaMP6s) were raised on standard cornmeal medium with a 12-h light and 12-h dark cycle at 23 °C and 60% humidity and housed in mixed male–female vials. Female flies (3–8 d old) were selected for brain imaging. UAS-nls-GCaMP6s flies were provided by D.J. Anderson (California Institute of Technology). To prepare for imaging, flies were anesthetized on ice and mounted in a 3D-printed plastic disk. The posterior head cuticle was opened using sharp forceps (5SF, Dumont) at room temperature in fresh saline (103 mM NaCl, 3 mM KCl, 5 mM TES, 1.5 mM CaCl₂, 4 mM MgCl₂, 26 mM NaHCO₃, 1 mM NaH₂PO₄, 8 mM trehalose and 10 mM glucose (pH 7.2), bubbled with 95% O₂ and 5% CO₂). After this, the fat body and air sac were also removed carefully. The position and angle of the flies were adjusted to keep the posterior of the head horizontal, and the window was made large and clean, for the convenience of observing multiple brain regions. Brain movement was minimized by adding UV glue around the proboscis[58]. Due to motion and latent osmotic pressure mismatch between the imaging saline and the body fluid of the fly, the brain showed deformation and nucleus drift during the imaging period.

We used a 2pSAM (mid-NA) system implemented with a water-immersion objective lens (×25, 1.05 NA, Olympus). Intravital volume imaging was acquired at a frame rate of 30 Hz with 13-angle scanning. The lateral field of view, 229 μm × 229 μm, covered the entire lateral range of the left central brain and some parts of the optic lobe. For continuous imaging over about 2 h, a wavelength of 920 nm at 25–35 mW of power was used for excitation. For preprocessing and reconstruction, we applied DeepCAD-RT to the images of each angle to enhance the signal-to-noise ratio[59,60]. Customized denoising models were trained for each channel of each fly. Next, we reconstructed the volumes using the algorithm of 2pSAM[17]. After the reconstruction, the lateral and axial resolutions were 0.45 μm and 0.75 μm, respectively.

### Data analysis

All data processing and analysis were performed using Python (version 3.9). The temporal traces of neurons in Fig. 5 were computed using the formula $\Delta F/F_0 = (F - F_0)/F_0$, where $F$ represents the intensity at the center points of neurons and $F_0$ denotes the mean value of $F$. Imaris (version 9.0.1) and TrackMate (version 7.10.1 in ImageJ) were used for cell tracking and segmentation analysis.

### Statistics and reproducibility

No statistical method was used to predetermine sample size. No data were excluded from the analyses. The experiments were not randomized. The investigators were not blinded to allocation during experiments and outcome assessment. Figure 2 is based on public datasets with cross-validation and does not involve independent experimental replication. Figures 3–5 are based on existing microscopy datasets, and the experiments were performed once without independent replication.

### Reporting summary

Further information on research design is available in the Nature Portfolio Reporting Summary linked to this article.

### Data availability

The *C. elegans* embryo datasets used in this study were taken from two public datasets: mskcc-confocal (https://doi.org/10.5281/zenodo.6460303)[61] and nih-ls (https://doi.org/10.5281/zenodo.6460375)[62]. The additional experimental imaging data generated in this work,

including large-scale 3D datasets of immune cells, cell–bacterium interactions and *Drosophila* neural activity, are not deposited in a public repository because of their terabyte-scale size and complex format. These datasets are available from the corresponding authors upon request. All materials that support the findings of this study are available from the corresponding authors upon request. Source data are provided with this paper.

## Code availability

All relevant codes of CELLECT are readily accessible and available at https://github.com/zzz333za/CELLECT. Code and example data for comparisons among different algorithms are also provided in the GitHub repository.

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

## Acknowledgements

This work was supported by the Beijing Natural Science Foundation (Z240011 to J.W.), the Natural Science Foundation of China (62088102 to Q.D., 62222508 to J.W. and 82302255 to H.Z.) and the Postdoctoral Science Foundation of China (2023M731985 to H.Z.).

## Author contributions

Q.D. and J.W. supervised the project. J.W. and H.Z. designed and initiated the project. H.Z. and S.K. designed the detailed implementations and performed visualizations. Z.Z. provided critical support for data collection beyond public datasets, including data acquisition for B cells, T cells, bacteria and neutrophils. J.F. contributed experimental data on *Drosophila* neurons. W.H. proposed improvements to the mathematical formulations in the paper. X.S. and L.S. provided valuable input on the article's structure and validated the simulation. H.A. and J.-R.Z. contributed experimental data for bacteria, neutrophils and macrophages. H.Z., S.K., J.W. and Q.D. wrote the manuscript with input from all authors.

## Competing interests

Q.D., J.W. and H.Z. submitted patent applications related to the CELLECT technology described in this work. The other authors declare no competing interests about the technology described in this work.

## Additional information

**Correspondence and requests for materials** should be addressed to Jiamin Wu or Qionghai Dai.

# Reporting Summary

## Statistics

For all statistical analyses, confirm that the following items are present in the figure legend, table legend, main text, or Methods section.

| n/a | Confirmed | |
|---|---|---|
| ☐ | ☒ | The exact sample size (*n*) for each experimental group/condition, given as a discrete number and unit of measurement |
| ☐ | ☒ | A statement on whether measurements were taken from distinct samples or whether the same sample was measured repeatedly |
| ☒ | ☐ | The statistical test(s) used AND whether they are one- or two-sided *Only common tests should be described solely by name; describe more complex techniques in the Methods section.* |
| ☒ | ☐ | A description of all covariates tested |
| ☒ | ☐ | A description of any assumptions or corrections, such as tests of normality and adjustment for multiple comparisons |
| ☒ | ☐ | A full description of the statistical parameters including central tendency (e.g. means) or other basic estimates (e.g. regression coefficient) AND variation (e.g. standard deviation) or associated estimates of uncertainty (e.g. confidence intervals) |
| ☒ | ☐ | For null hypothesis testing, the test statistic (e.g. *F*, *t*, *r*) with confidence intervals, effect sizes, degrees of freedom and *P* value noted *Give P values as exact values whenever suitable.* |
| ☒ | ☐ | For Bayesian analysis, information on the choice of priors and Markov chain Monte Carlo settings |
| ☒ | ☐ | For hierarchical and complex designs, identification of the appropriate level for tests and full reporting of outcomes |
| ☒ | ☐ | Estimates of effect sizes (e.g. Cohen's *d*, Pearson's *r*), indicating how they were calculated |

*Our web collection on statistics for biologists contains articles on many of the points above.*

## Software and code

Policy information about availability of computer code

| Data collection | 2pSAM and SLFM imaging data were acquired using MATLAB R2021a (Mathworks) and ScanImage (version SI5.7R1. refer to Zhao,Z. et al. Cell. 2023. https://di.org/10.1016/j.cell.2023.04.016 and Jiamin, Wu et al. Cell 2021. https://doi.org/10.1016/j.cell.2021.04.029). |
|---|---|
| Data analysis | Data analysis was performed using customized Python scripts (version 3.9). Imaris (version 9.0.1) and TrackMate 7 within ImageJ were used for cell tracking and segmentation analysis. The custom code developed in this study is available at GitHub: https://github.com/zzz333za/CELLECT. |

For manuscripts utilizing custom algorithms or software that are central to the research but not yet described in published literature, software must be made available to editors and reviewers. We strongly encourage code deposition in a community repository (e.g. GitHub). See the Nature Portfolio guidelines for submitting code & software for further information.

## Data

Policy information about availability of data

All manuscripts must include a data availability statement. This statement should provide the following information, where applicable:

- Accession codes, unique identifiers, or web links for publicly available datasets
- A description of any restrictions on data availability
- For clinical datasets or third party data, please ensure that the statement adheres to our policy

The C. elegans embryo datasets used in this manuscript were taken from 2 public datasets, mskcc-confocal (data: https://doi.org/10.5281/zenodo.6460303) and

nih-ls (data: https://doi.org/10.5281/zenodo.6460375). All other relevant data that support the findings of this work are available from the corresponding authors upon request.

# Research involving human participants, their data, or biological material

Policy information about studies with human participants or human data. See also policy information about sex, gender (identity/presentation), and sexual orientation and race, ethnicity and racism.

| | |
|---|---|
| Reporting on sex and gender | *Use the terms sex (biological attribute) and gender (shaped by social and cultural circumstances) carefully in order to avoid confusing both terms. Indicate if findings apply to only one sex or gender; describe whether sex and gender were considered in study design; whether sex and/or gender was determined based on self-reporting or assigned and methods used.*<br>*Provide in the source data disaggregated sex and gender data, where this information has been collected, and if consent has been obtained for sharing of individual-level data; provide overall numbers in this Reporting Summary. Please state if this information has not been collected.*<br>*Report sex- and gender-based analyses where performed, justify reasons for lack of sex- and gender-based analysis.* |
| Reporting on race, ethnicity, or other socially relevant groupings | *Please specify the socially constructed or socially relevant categorization variable(s) used in your manuscript and explain why they were used. Please note that such variables should not be used as proxies for other socially constructed/relevant variables (for example, race or ethnicity should not be used as a proxy for socioeconomic status).*<br>*Provide clear definitions of the relevant terms used, how they were provided (by the participants/respondents, the researchers, or third parties), and the method(s) used to classify people into the different categories (e.g. self-report, census or administrative data, social media data, etc.)*<br>*Please provide details about how you controlled for confounding variables in your analyses.* |
| Population characteristics | *Describe the covariate-relevant population characteristics of the human research participants (e.g. age, genotypic information, past and current diagnosis and treatment categories). If you filled out the behavioural & social sciences study design questions and have nothing to add here, write "See above."* |
| Recruitment | *Describe how participants were recruited. Outline any potential self-selection bias or other biases that may be present and how these are likely to impact results.* |
| Ethics oversight | *Identify the organization(s) that approved the study protocol.* |

Note that full information on the approval of the study protocol must also be provided in the manuscript.

# Field-specific reporting

Please select the one below that is the best fit for your research. If you are not sure, read the appropriate sections before making your selection.

☒ Life sciences ☐ Behavioural & social sciences ☐ Ecological, evolutionary & environmental sciences

For a reference copy of the document with all sections, see nature.com/documents/nr-reporting-summary-flat.pdf

# Life sciences study design

All studies must disclose on these points even when the disclosure is negative.

| | |
|---|---|
| Sample size | Sample sizes, in our case the number of mice and recordings, were chosen to ensure that animal-to-animal and recording-to-recording variability was reflected in the captured data. Only animals were included in the study for which all animal procedures (as described in Methods) worked successfully to allow for signal detection (see next item). Provided that animal procedures (surgeries and viral injections/ indicator expression) were successful as verified using a conventional two-photon microscope. For T cell experiment, we used a standard two-photon microscope but other experiments including B cell imaging, tripartite interaction imaging, and brain tissue imaging were performed using a 2pSAM system. We used single photon SLFM system only for neutrophil tracking dataset in SVideo1. We confirmed that imaging results and data quality to be reliably reproducible and consistent, both across imaging sessions with the same animal, and across animals. Since the subject of our manuscript is to establish a neural recording and signal extraction method rather than any biological findings, we consider this sample size sufficient to verify the performance of our method. |
| Data exclusions | Only animals were included in the study for which all animal procedures (as described in Online Methods) worked successfully to allow for signal detection (i.e., GECI expression observable), as verified using a conventional two-photon microscope. Of these animals, none were excluded. |
| Replication | For all animals in which animal procedures (surgeries and viral injections/GECI expression, see Online Methods) were successful (as verified using a conventional two-photon microscope), imaging and data analysis results were reliably reproduced, both across imaging sessions with the same animal, and across animals. |
| Randomization | Randomization was not relevant to this study, because no experimental groups were formed. |
| Blinding | Blinding was not relevant to this study, because no group allocation was performed. |

# Reporting for specific materials, systems and methods

We require information from authors about some types of materials, experimental systems and methods used in many studies. Here, indicate whether each material, system or method listed is relevant to your study. If you are not sure if a list item applies to your research, read the appropriate section before selecting a response.

## Materials & experimental systems

| n/a | Involved in the study |
|-----|----------------------|
| ☒ ☐ | Antibodies |
| ☒ ☐ | Eukaryotic cell lines |
| ☒ ☐ | Palaeontology and archaeology |
| ☐ ☒ | Animals and other organisms |
| ☒ ☐ | Clinical data |
| ☒ ☐ | Dual use research of concern |
| ☒ ☐ | Plants |

## Methods

| n/a | Involved in the study |
|-----|----------------------|
| ☒ ☐ | ChIP-seq |
| ☒ ☐ | Flow cytometry |
| ☒ ☐ | MRI-based neuroimaging |

## Animals and other research organisms

Policy information about studies involving animals; ARRIVE guidelines recommended for reporting animal research, and Sex and Gender in Research

| | |
|---|---|
| Laboratory animals | C57BL/6J mice (Mus musculus, inbred strain, substrain J), male and female, 6–8 weeks old, obtained from The Jackson Laboratory. Mice were housed in standard cages with a maximum of 5 mice per cage. Cages were housed in an environment with a 12/12h reverse dark/light cycle, and ambient temperature of 72°F and an ambient humidity of ~30%. Mice were provided food and water. |
| Wild animals | The study did not involve wild animals. |
| Reporting on sex | Sex was not considered in the study design and no sex-based analysis was performed. |
| Field-collected samples | The study did not involve field-collected samples. |
| Ethics oversight | All animal procedures were approved by the Institutional Animal Care and Use Committee (IACUC) of Tsinghua University. |

Note that full information on the approval of the study protocol must also be provided in the manuscript.

## Plants

| | |
|---|---|
| Seed stocks | *Report on the source of all seed stocks or other plant material used. If applicable, state the seed stock centre and catalogue number. If plant specimens were collected from the field, describe the collection location, date and sampling procedures.* |
| Novel plant genotypes | *Describe the methods by which all novel plant genotypes were produced. This includes those generated by transgenic approaches, gene editing, chemical/radiation-based mutagenesis and hybridization. For transgenic lines, describe the transformation method, the number of independent lines analyzed and the generation upon which experiments were performed. For gene-edited lines, describe the editor used, the endogenous sequence targeted for editing, the targeting guide RNA sequence (if applicable) and how the editor was applied.* |
| Authentication | *Describe any authentication procedures for each seed stock used or novel genotype generated. Describe any experiments used to assess the effect of a mutation and, where applicable, how potential secondary effects (e.g. second site T-DNA insertions, mosiacism, off-target gene editing) were examined.* |

