## [Peer Review File · Nature Methods]

CELLECT: Contrastive Embedding Learning for Large-scale Efficient Cell Tracking

Corresponding Author: Professor Qionghai Dai

Version 0:

Decision Letter:

31st Mar 2025

Dear Qionghai,

Your Article, "Contrastive Embedding Learning for Large-scale Efficient Cell Tracking: CELLECT", has now been seen by four reviewers (refs 2 and 3 co-reviewed). As you will see from their comments below, although the reviewers find your work of considerable potential interest, they have raised a number of concerns. We are interested in the possibility of publishing your paper in Nature Methods, but would like to consider your response to these concerns before we reach a final decision on publication. We therefore invite you to revise your manuscript to address these concerns.

We ask that you focus on adding the requested benchmarking, improving software usability, add the requested assessment of segmentation accuracy and address/clarify the other technical points. Please note, because the referees were very concerned about the benchmarking results compared to the Cell Tracking Challenge, we cannot promise to send the paper back to the reviewers until we've seen the revised manuscript.

Link Redacted

We hope to receive your revised paper within three months. If you cannot send it within this time, please let us know. In this

event, we will still be happy to reconsider your paper at a later date so long as nothing similar has been accepted for publication at Nature Methods or published elsewhere.

OPEN SCIENCE REQUIREMENTS

REPORTING SUMMARY AND EDITORIAL POLICY CHECKLISTS

EXTENDED DATA FIGURES

DATA AVAILABILITY

All novel DNA and RNA sequencing data, protein sequences, genetic polymorphisms, linked genotype and phenotype data, gene expression data, macromolecular structures, and proteomics data must be deposited in a publicly accessible database, and accession codes and associated hyperlinks must be provided in the "Data Availability" section.

CODE AVAILABILITY

Please include a "Code Availability" subsection in the Online Methods which details how your custom code is made available. Only in rare cases (where code is not central to the main conclusions of the paper) is the statement "available upon request" allowed (and reasons should be specified).

For more information on our code sharing policy and requirements, please see:
<https://www.nature.com/nature-research/editorial-policies/reporting-standards#availability-of-computer-code>

MATERIALS AVAILABILITY

ORCID

Sincerely,
Rita

Rita Strack, Ph.D.
Senior Editor
Nature Methods

Reviewers' Comments:

Reviewer #1 (Remarks to the Author):

A. Summary of the key results

The paper "Contrastive Embedding Learning for Large-scale Efficient Cell Tracking: CELLECT" by Zhou et al. proposes contrastive learning using sparse labeling with different fluorescent markers to train a 3D cell tracking. Besides contrastive learning, different customized subnets significantly improve segmentation and tracking results, shown by ablation studies. The superiority of the approach against leading methods of the Cell Tracking Challenge (Maska 2023 Nature Methods) is claimed, but a more elaborate evaluation is necessary to rate the progress against existing tracking methods. This includes a formal submission to the Challenge website (<https://celltrackingchallenge.net>) and an extended set of benchmark methods. If these evaluations will show that the proposed approach outperforms the leading algorithms, the paper could be a candidate for a Nature Methods publication. In the given state, this decision is not yet possible. Very encouraging, fast and impressive results are shown for various other complex tasks without retraining, e.g. for an in vivo screen for bacteria, neutrophils and bacteria in mice or tracking of firing neurons in elastically deformed drosophila brains.

B. Originality and significance: if not novel, please include reference

The approach of contrastive learning to 3D cell tracking is novel, but relatively straightforward by combining leading methods from cell tracking and deep learning. The main improvement comes from the combination of sparse fluorescent labeling combining with the mentioned method to avoid a labor-intensive labeling of 3D datasets.

C. Data & methodology: validity of approach, quality of data, quality of presentation

Major point:

* It is not clear from the paper if the results were officially submitted at the Cell Tracking Challenge and are visible in the Leaderboard of the challenge (<https://celltrackingchallenge.net/latest-ctb-results/>). This is for my opinion mandatory because it shows the generalization on an unseen and hidden test dataset. Please add your team name, the places in the leaderboard for a defined date and the official code names of the dataset. It would be also interesting to quantify which

subset of training data was used. It would also be useful to discuss the differences to the other TOP3 algorithms on the same dataset in detail, including a quantitative evaluation in Fig. 2 - especially in case of available code. And explain why you selected the 3 concurrent algorithms in Fig. 2.

* Fig. 1 Why is necessary to work with two separate MLPs (inter-frame vs. inter-frame) instead of integrating these tasks in one network architecture with multiple outputs including track linkage?

* In many applications, cell divisions are not the only event to qualitatively change cell tracks. How do you handle dying/apoptotic cells (especially with visible fragments) and cells leaving or entering the field of view? This could heavily influence the matching of cells with the MLPs in Supp Fig. 5.

* There were other papers in the past that worked with embeddings and real-time approaches, e.g. [Neven 2018] to estimate the position of the corresponding cell center followed by clustering approaches in instance segmentation:

Neven, D., Brabandere, B. D., Proesmans, M., & Gool, L. V. (2019). Instance segmentation by jointly optimizing spatial embeddings and clustering bandwidth. In Proceedings of the IEEE/cvf conference on computer vision and pattern recognition (pp. 8837-8845).

I would recommend to give a short overview about such approaches (including similar ideas for direct clustering) and compare the chosen approach with such papers.

* How does the approach solve missing segmentations (e.g. very low confidence values) at time point t+1 leading to broken linkages? Other clustering approaches are looking for candidate segmentations subsequent time points as t+2 and t+3 and uses these segmentations to repair missing t+1-segmentations.

* How do you guarantee that the features for the sparse annotations are representative for the other cells?

* Please explain in all figure captions which datasets are used.

Minor points:

* I would recommend to state earlier (even in the abstract) that the sparse annotations stem from different fluorescent labels. Otherwise, sparse annotation could be misinterpreted as manual labeling.

* Explain what csc and sSVM (support vector machine?) means in Linaje+csc+sSVM to give readers insights into concurrent algorithms without checking the references.

* The CEN should be explicitly mentioned in Fig. 1, with a reference to Supp Fig. 1.

* 480/481 "The obtained features were subsequently input into an MLP network for further classification." What do you mean with the "obtained features"? The embeddings?

* 496/497 duplicated text: represent points from distinct labeled cells. represent points from distinct labeled cells

* 789 captured by confocal microscopy (CLSM, left) -> captured by confocal laser scanning microscopy (CLSM, left)

* 836 faciliate -> facilitate

* Fig. 4g neutorphil -> neutrophil

* Supp Fig. 1 Add the dimensions of the input figure and move all dimension numbers to the related block. Otherwise, a misunderstanding is possible that 8x256x256x32 belongs to the input dimensions.

* Supp Fig. 1: Explain, why the confidence map in the output channel is 4-dimensional. And why the center point location is 5-dimensional (the four levels + background?). I would recommend to add a supplementary table containing all dimensions of input and output channels per neural network.

* Supp Fig. 1 Is it really a blue + magenta arrow in the decoder path (lowest element) of the U-net?

* Supp Fig. 1 embedding -> embedding

* Supp Fig. 1 division -> division

* Legend Supp Fig. 1 The color of the arrows correspond -> The color of the arrows corresponds

* Add one sentence to the interpretation of marker patterns in the right membrane-labeled cell in Supp Fig. 2 g & i - to gain evidence that this is really one cell.

* Supp Fig. 5 - The three (as shown in the right panel?) or more (?) inputs of inter- and intra-frame MLP remain unclear. I would recommend to add a supplementary table containing all dimensions of input and output channels per neural network.

* Supp Fig 6: Is the right figure correct? I did not had expected a dotted link from blue to gray and a correct link from blue to blue, this looks for me like a division event.

* Legend Supp Fig. 8 with a green box in the each original image -> with a green box in each original image

* Dataset numbers #1 and #2 only occur in supplement and are not used throughout the paper.

D. Appropriate use of statistics and treatment of uncertainties

Yes. Uncertainties are explicitly handled for segmentation masks in form of the confidence map and with retraining results in Supp Fig. 11. However, confidence maps for linking cells between two subsequent time points are missing.

E. Conclusions: robustness, validity, reliability

see suggested improvements

F Suggested improvements: experiments, data for possible revision

see suggested improvements

G. References: appropriate credit to previous work?

Yes, mostly. But please compare your approach with other contrastive-learning approaches for cell tracking, see e.g.

Zyss, D., Sharma, A., Ribeiro, S. A., Repellin, C. E., Lai, O., Ludlam, M. J., ... & Fehri, A. (2024). Contrastive learning for cell division detection and tracking in live cell imaging data. bioRxiv, 2024-08.

H. Clarity and context: lucidity of abstract/summary, appropriateness of abstract, introduction and conclusions

No problems.

Reviewer #2 (Remarks to the Author):

Please see the attached PDF document.

Reviewer #2 (Remarks on code availability):

Code could not be installed, with errors detailed in the review document.

Reviewer #4 (Remarks to the Author):

A. Summary of the key results:

The authors introduce CELLECT, a contrastive learning-based method for learning embeddings at the voxel level in 3D image volumes. CELLECT achieves computational efficiency by avoiding global-in-time optimization, instead leveraging a lightweight 3D model (<3M parameters) and two MLPs (Intra-frame and Inter-frame). The authors report it to be 56 times faster than Linajea (Malin-Mayor et al.). The method demonstrates competitive tracking performance across multiple datasets: two public *C. elegans* datasets, real-time large-scale B cell tracking in mouse lymph nodes, and neural signal extraction in *Drosophila*.

F. Suggested improvements: experiments, data for possible revision

****Ablation Study on CEN, Intra-frame MLP, and Inter-frame MLP Contributions:****

To better understand the individual contributions of the CEN, Intra-frame MLP, and Inter-frame MLP modules, I suggest including a quantitative ablation study. Specifically, on at least one dataset, it would be valuable to report how Tracking Accuracy and/or other relevant metrics change when adding each component incrementally. While Supplementary Figure 7 provides qualitative insight, clear quantitative results would strengthen the argument and clarify the role of each module, particularly the two MLPs.

A table similar to the one below would be informative:

3D-UNet	CEN	Intra-frame MLP	Inter-frame MLP	Tracking Accuracy
---	---	---	---	---
Yes	No	No	No	?
Yes	Yes	No	No	?
Yes	Yes	Yes	No	?
Yes	Yes	Yes	Yes	?

****Clarification on Linajea's Performance at Low Frame Rates:****

In the Introduction, the authors state:

>"However, its global optimization of cell trajectories based on time series information leads to large computational costs and degradation in datasets acquired at low frame rates, restricting its applications in large scale long term 3D imaging."

This conclusion about performance degradation at low frame rates may not be straightforward. It would strengthen the manuscript to include an experiment where time frames are deliberately undersampled, and Linajea is evaluated on such datasets. Similarly, it would be informative to assess CELLECT's robustness under frame rate undersampling conditions. This would provide direct evidence supporting the claim and offer practical insights into both methods' performance in low temporal resolution settings.

****Tiling Artifacts in Supplementary Figure 4:****

In Supplementary Figure 4 (bottom row), noticeable tiling artifacts are present. While overlapping tiles are mentioned, details on how these artifacts are mitigated are lacking. It would be helpful if the authors could clarify whether padding strategies are applied during inference.

Valid (unpadded) convolutions combined with overlapping tiles might eliminate the artifacts. Reducing such artifacts may further improve downstream tracking accuracy.

****Clarification on Cell Tracking Challenge (CTC) Claims:****

The manuscript mentions achieving state-of-the-art tracking accuracy in the Cell Tracking Challenge. To improve transparency and reproducibility:

- Please specify which CTC public datasets were used for training and evaluation.

- It would be helpful to provide a link to the corresponding page on the CTC website showing the CELLECT results and method description.
- Please also include a link to the trained model (if available) in the manuscript.
- Consider adding quantitative results on the public CTC datasets to support the state-of-the-art claim. Currently, results are shown only on two *C. elegans* datasets, which, though publicly available on Zenodo, are not part of CTC.

H. Clarity and context:

Typographical Errors:

- Line 547 ("C. elegans embryo Public"): "CELLEECT" → should be "CELLECT".
- Figure 1: "Division prediction" → should be "Division prediction".
- Figure 2a: "StarrayNite" → should be "StarryNite".
- Supplementary Figure 8: "Singal Intensity" → should be "Signal Intensity".
- Supplementary Figure 7: "Divsion predictor" → should be "Division predictor".

Metric Clarification:

Please explicitly mention in the manuscript that the "Tracking Accuracy" metric originates from Linajea (Malin-Mayor et al.). This metric is not commonly used in other tracking benchmarks, so citing its origin would provide helpful context for readers unfamiliar with it.

Reviewer #4 (Remarks on code availability):

I was able to install the code (CELLECT-ctc.ver) and download the example data (Fluo-N3DH-CE).

However, a few additional steps were required to install the method:

- Adding tqdm, scikit-learn, and ipywidgets to the requirements.txt file.
- Creating a new directory within /src and running the training script from there, as the code relies on relative paths.

The codebase was straightforward to start training, but at epoch 3/199, I ran into the following error:

```
line 781, in __init__
super().__init__(torch._C.PyTorchFileWriter(self.name, _compute_crc32))
^^^^^^^^^^^^^^^^^^^^^^^^^^^^^^^^^^^^^^^^^^^^^^^^^^^^^^^^^^^^^^^^^^^^^^^^^^^^
RuntimeError: Parent directory ./Trained models/ does not exist.
```

This directory 'Trained models' should be created by default, if it doesn't exist already.

In general, the method would benefit from improved documentation and restructuring into smaller, modular files to enhance readability and usability.

Version 1:

Decision Letter:

28th Jul 2025

Dear Qionghai,

Thank you for your letter detailing how you would respond to the reviewer concerns regarding your Article, "Contrastive Embedding Learning for Large-scale Efficient Cell Tracking: CELLECT". We have decided to invite you to revise your manuscript as you have outlined, before we reach a final decision on publication. Because there will be updates to the software, we will have to send the paper quickly back to the reviewer to make sure it's working properly.

- * include a point-by-point response to the reviewers and to any editorial suggestions
- * please underline/highlight any additions to the text or areas with other significant changes to facilitate review of the revised manuscript
- * address the points listed described below to conform to our open science requirements

* ensure it complies with our general format requirements as set out in our guide to authors at www.nature.com/naturemethods

* resubmit all the necessary files by using the link below to access your home page

Link Redacted

Note: This URL links to your confidential account and associated information about manuscripts you may have submitted, or that you are reviewing for us. If you wish to forward this email to co-authors, please delete this link.

We hope to receive your revised paper within one month. If you are substantially delayed, please let us know. In this event, we will still be happy to reconsider your paper at a later date so long as nothing similar has been accepted for publication at Nature Methods or published elsewhere.

OPEN SCIENCE REQUIREMENTS

REPORTING SUMMARY

When revising your manuscript, please update your reporting summary.

For any revision that includes light microscopy data, we ask our authors to please include a completed light microscopy reporting table [https://www.nature.com/documents/Light_microscopy_reporting_table.xlsx] to ensure the methods are described thoroughly. The table will be available to reviewers and ultimately published should the manuscript be accepted at the journal.

EXTENDED DATA FIGURES

DATA AVAILABILITY

For papers containing bioimaging data, we strongly recommend depositing the data to Bioimage Archive (<https://www.ebi.ac.uk/bioimage-archive/>). Associated accession codes and hyperlinks should be provided in the "Data Availability" section.

CODE AVAILABILITY

Please include a "Code Availability" subsection in the Online Methods which details how your custom code is made available. Only in rare cases (where code is not central to the main conclusions of the paper) is the statement "available upon request" allowed (and reasons should be specified).

We request that you deposit code in a DOI-minting repository such as Zenodo, Gigantum or Code Ocean and cite the DOI in

the Reference list. We also request that you use code versioning and provide a license.

MATERIALS AVAILABILITY

ORCID

Nature Methods is committed to improving transparency in authorship. As part of our efforts in this direction, we are now requesting that all authors identified as 'corresponding author' on published papers create and link their Open Researcher and Contributor Identifier (ORCID) with their account on the Manuscript Tracking System (MTS), prior to acceptance. This applies to primary research papers only. ORCID helps the scientific community achieve unambiguous attribution of all scholarly contributions. You can create and link your ORCID from the home page of the MTS by clicking on 'Modify my Springer Nature account'. For more information please visit <http://www.springernature.com/orcid>.

Best regards,
Rita

Rita Strack, Ph.D.
Senior Editor
Nature Methods

Reviewers' Comments:

Reviewer #1 (Remarks to the Author):

A. Summary of the key results

The paper "Contrastive Embedding Learning for Large-scale Efficient Cell Tracking: CELLECT" by Zhou et al. proposes contrastive learning using sparse labeling with different fluorescent markers to train a 3D cell tracking. Besides contrastive learning, different customized subnets significantly improve segmentation and tracking results, shown by ablation studies. The superiority of the approach against leading methods of the Cell Tracking Challenge (Maska 2023 Nature Methods) was shown by first places in the segmentation and tracking challenge in a formal submission to the Challenge website (<https://celltrackingchallenge.net>) for one 3D dataset.

Very encouraging, fast and impressive results are shown for various other complex tasks without retraining, e.g. for an in vivo screen for bacteria, neutrophils and bacteria in mice or tracking of firing neurons in elastically deformed drosophila brains.

In the submitted revised version, the paper is technical sound and I recommend a publication in Nature Methods.

B. Originality and significance: if not novel, please include reference

The approach of contrastive learning to 3D cell tracking is novel, but relatively straightforward by combining leading methods from cell tracking and deep learning. The main improvement comes from the combination of sparse fluorescent labeling combining with the mentioned method to avoid a labor-intensive labeling of 3D datasets.

C. Data & methodology: validity of approach, quality of data, quality of presentation

All recommendations and questions from the previous review were successfully answered.

Minor points:

* Clarify the exact input and output dimensions of intra- and inter-frame-MLPs in figure caption of Supp. Fig. 5 to the exact and well understandable settings in Suppl. Table 3.

* Remove Neven at the end of Reference [52]

D. Appropriate use of statistics and treatment of uncertainties

Yes. Uncertainties are explicitly handled for segmentation masks in form of the confidence map (see e.g. Suppl. Fig. 2).

E. Conclusions: robustness, validity, reliability

Okay.

F Suggested improvements: experiments, data for possible revision

Okay.

G. References: appropriate credit to previous work?

Yes.

H. Clarity and context: lucidity of abstract/summary, appropriateness of abstract, introduction and conclusions

No problems

Reviewer #1 (Remarks on code availability):

No comments

Reviewer #2 (Remarks to the Author):

See the attached document 'Second review of Contrastive Embedding Learning for Large-scale Efficient Cell Tracking CELLECT by Zhou and colleagues.pdf'

Reviewer #2 (Remarks on code availability):

See the attached document 'Second review of Contrastive Embedding Learning for Large-scale Efficient Cell Tracking CELLECT by Zhou and colleagues.pdf'

Reviewer #4 (Remarks to the Author):

A. Summary of Key Results

(a) The authors introduce CELLECT, a deep learning-based tracking framework that improves accuracy and reduces computational cost by learning per-detection embeddings using a contrastive loss. These embeddings are then used to match detections between adjacent frames based on feature similarity, eliminating the need for globally constrained optimization and enabling real-time inference.

(b) The authors demonstrate that a model trained on a single dataset (mskcc-confocal) generalizes well to a other datasets and imaging modalities without requiring retraining or fine-tuning.

(c) The combination of high tracking accuracy, strong cross-domain generalization, and real-time performance makes CELLECT a practical tool for large-scale biological applications.

(d) The paper presents a quite thorough investigation with several experiments, ablations and evaluated datasets.

F. Suggested Improvements

(a) The discussion section could be strengthened by addressing how CELLECT might support human-in-the-loop tracking, e.g., by identifying uncertain cell detections or links that could be prioritized for manual annotation, particularly when the pre-trained model underperforms on a novel dataset.

(b) The authors should make the pre-trained, generalist CELLECT model available for public download and include the URL in the manuscript to facilitate reproducibility and adoption by the community.

(c) Although the generalizability of the pretrained model becomes evident later in the paper, the authors could make it more explicit early on (e.g., in the abstract and introduction) that the model was trained solely on the mskcc dataset and applied across diverse datasets. This broad generalization is a significant contribution on its own.

(d) What happens if a new dataset has much larger cells than the ones in mskcc? How does CELLECT deal with this (for example, does one need to downsample new datasets to have similar cell size as mskcc?) A mention of this in the discussion would be welcome.

(e) Often biologists have this question - how frequently should one image to have the best downstream tracking? Could the authors provide any guidelines or discussion related to this?

H. Clarity and Context

(a) Title:

Consider reordering: "CELLECT: Contrastive Embedding for Large-scale, Efficient Cell Tracking"

(b) Abstract:

(i) "a pretrained CELLECT model" → a pre-trained CELLECT model

(ii) "CELLECT reduces the computational costs by 56 times" →

CELLECT reduces inference-time computational cost by a factor of 56.

(iii) Consider adding:

This efficiency gain arises from replacing globally constrained optimization with greedy, frame-wise matching based on learned embeddings. This insight may help attract broader interest from users seeking real-time solutions.

(iv) "CELLECT, which achieves state of the art tracking accuracy in the Cell Tracking Challenge" → CELLECT achieves state-of-the-art tracking accuracy on the *C. elegans* dataset in the Cell Tracking Challenge.

(c) Introduction:

(i) "Deep-learning-based cell tracking algorithms" → Deep learning-based cell tracking algorithms

(ii) The sentence: "However, its reliance on fixed-length temporal windows for global trajectory optimization introduces high computational costs and strict requirements on temporal sampling, limiting its real time performance and applicability to low-frame rate or irregularly sampled data." could be trimmed to avoid over-claiming: "...introduces high computational costs and strict requirements on temporal sampling, limiting real-time performance." (The claim about low-frame-rate applicability was not tested in the *Linajea* paper.)

(d) Results – Section Title:

(i) Consider revising "Principle of CELLECT" to:

"Core Idea Behind CELLECT" or "Overview of CELLECT" for improved tone.

(e) In general, the language in the paper can be refined a little further by using shorter and cleaner sentences.

Version 2:

Decision Letter:

Our ref: NMETH-A59736B

5th Sep 2025

Dear Dr. Dai,

Thank you for submitting your revised manuscript "CELLECT: Contrastive Embedding Learning for Large-scale Efficient Cell Tracking" (NMETH-A59736B). It has now been seen by the original referees and their comments are below. The reviewers find that the paper has improved in revision, and therefore we'll be happy in principle to publish it in Nature Methods, pending minor revisions to satisfy the referees' final requests and to comply with our editorial and formatting guidelines.

TRANSPARENT PEER REVIEW

Please note: we allow redactions to authors' rebuttal and reviewer comments in the interest of confidentiality. If you are

concerned about the release of confidential data, please let us know specifically what information you would like to have removed. Please note that we cannot incorporate redactions for any other reasons. Reviewer names will be published in the peer review files if the reviewer signed the comments to authors, or if reviewers explicitly agree to release their name. For more information, please refer to our [FAQ page](https://www.nature.com/documents/nr-transparent-peer-review.pdf).

ORCID

Sincerely,
Madhura

Madhura Mukhopadhyay, PhD
Senior Editor
Nature Methods

Reviewer #2 (Remarks to the Author):

Code quality and reusability

We thank the authors for their recent work on the code. Thanks to the added documentation, there is no more installation issues nor confusion about utility and usage of each file. The addition of the inference module will prove useful to users and we could run it on custom data. While the code is still not perfectly clean across all files, we believe the improvements are sufficient to ensure both the use of CELLECT and the reusability of its codebase.

Reviewer #2 (Remarks on code availability):

The recent changes introduced after the previous round of revision have addressed all of our concerns. We are convinced that the code shipped with this article will be of great values to the field and to biologists.

Reviewer #3 (Remarks to the Author):

I co-reviewed this manuscript with one of the reviewers who provided the listed reports. This is part of the Nature Methods initiative to facilitate training in peer review and to provide appropriate recognition for Early Career Researchers who co-review manuscripts.

Version 3:

Decision Letter:

30th Sep 2025

Dear Qionghai,

I am delighted to tell you that your manuscript NMETH-A59736C has been accepted for publication in Nature Methods. The received and accepted dates will be Feb 09, 2025 and Sep 30, 2025. This note is intended to let you know what to expect from us over the next month or so, and to let you know where to address any further questions.

As discussed, we will publish your paper on an accelerated schedule. **Please carefully review the details below and contact us immediately at methods@us.nature.com if you have any travel plans or other conflicts that may make you unable to respond to us for the next 5-7 days.**

In approximately 2 business days you will receive a link to choose the appropriate publishing options for your paper and complete the appropriate grant of rights necessary to publish your work. As it is vital that this process not be delayed, we strongly encourage you to [check your spam filter and whitelist](https://www.simpleminds.com/how-to-check-your-spam-filter-and-whitelist).

emails/">whitelist the email address do-not-reply@springernature.com to ensure that this message is received.

Shortly after this step is completed, you will receive a link to your electronic proof via email with a request to make any necessary corrections as soon as possible. You will find that we have made minor changes to enhance the clarity of the text and to ensure that your paper conforms to the journal's style so we ask that you review these proofs carefully to ensure that we have not inadvertently introduced errors or altered the sense of your text in any way.

Please return your proof within 24 hours of receiving it. If you have any questions about your proofs or anticipate any delays please contact rjsproduction@springernature.com immediately.

Authors may need to take specific actions to achieve compliance with funder and institutional open access mandates. If your research is supported by a funder that requires immediate open access (e.g. according to [Plan S principles](https://www.springernature.com/gp/open-science/plan-s-compliance) or the [NIH public access policy](https://www.springernature.com/gp/open-science/us-federal-agency-compliance)) then you should select the gold OA route, and we will direct you to the compliant route where possible. Because authors warrant under our subscription licensing terms that they haven't committed to licensing any version of their article under a licence inconsistent with the terms of our agreement – including the applicable embargo period – publication under the subscription model isn't suitable for authors whose funders require no embargo.

If you have any questions about our publishing options, costs, Open Access requirements, or our legal forms, please contact ASJournals@springernature.com.

If you are active on Twitter/X, please e-mail me your and your coauthors' handles so that we may tag you when the paper is published.

Sincerely,
Madhura

Madhura Mukhopadhyay, PhD
Senior Editor
Nature Methods

** Visit the Springer Nature Editorial and Publishing website at [www.springernature.com/editorial-and-publishing-jobs](https://group.springernature.com/gp/group/careers/editorial) for more information about our career opportunities. If you have any questions please click [here](mailto:editorial.publishing.jobs@springernature.com).**

Response to Reviewers – NMETH-A59736

We sincerely thank the reviewers and editors for their insightful comments and constructive feedback on our manuscript entitled “Contrastive Embedding Learning for Large-scale Efficient Cell Tracking: CELLECT.” We have thoroughly revised the manuscript in response to all concerns and provide below a detailed, point-by-point response. Additions and major changes in the revised manuscript are indicated by highlighting in red. For ease of communication, the original referee comments are shown in black color, whereas our specific answers are **in blue**. Thank you again for your great efforts.

Reviewer #1

Comment A: Summary of the key results

The paper “Contrastive Embedding Learning for Large-scale Efficient Cell Tracking: CELLECT” by Zhou et al. proposes contrastive learning using sparse labeling with different fluorescent markers to train a 3D cell tracking. Besides contrastive learning, different customized subnets significantly improve segmentation and tracking results, shown by ablation studies. The superiority of the approach against leading methods of the Cell Tracking Challenge (Maska 2023 Nature Methods) is claimed, but a more elaborate evaluation is necessary to rate the progress against existing tracking methods. This includes a formal submission to the Challenge website (<https://celltrackingchallenge.net>) and an extended set of benchmark methods. If these evaluations will show that the proposed approach outperforms the leading algorithms, the paper could be a candidate for a Nature Methods publication. In the given state, this decision is not yet possible. Very encouraging, fast and impressive results are shown for various other complex tasks without retraining, e.g. for an in vivo screen for bacteria, neutrophils and bacteria in mice or tracking of firing neurons in elastically deformed drosophila brains.

Response:

We highly appreciate the reviewer's positive feedback. In response to the constructive comment on evaluation, we have officially submitted CELLECT to the Cell Tracking Challenge, where it achieved 1st place in both segmentation and tracking in the data track of Fluo-N3DH-CE. Additionally, we expanded our benchmark evaluation against a wider set of methods, demonstrating CELLECT's superiority in accuracy and computational efficiency. We summarized the revisions corresponding to the reviewer's all comments as below:

- We officially submitted CELLECT to the Cell Tracking Challenge under the team name "THU-CN (3)", achieving 1st place in both segmentation (CSB) and tracking (CTB) on the Fluo-N3DH-CE test set (**Fig. R1**; <https://celltrackingchallenge.net/>).

- We clarified the rationale for designing 2 separate light-weight MLPs in the whole framework instead of a unified architecture (**Fig. R3**).
- We explained how CELLECT handles cell apoptosis and cells entering/leaving the field of view, and how this impacts matching (**Fig. R3**).
- We added the conceptual differences between CELLECT and Neven et al. (CVPR 2019) in the revised **Methods** section ("*Comparison with embedding-based and contrastive learning methods*").
- We clarified that CELLECT's local peak-based detection minimizes missing segmentations, but for flickering datasets (e.g., *Drosophila* calcium imaging), we added optional post-processing to bridge gaps across ± 200 frames (**Methods**).
- We added that CELLECT prioritizes local feature discriminability under sparse supervision, using triplet-based contrastive learning and auxiliary MLPs, with candidate selection guided by spatial proximity (**Methods; Fig. R4**).
- We revised all figure captions in the main manuscript and supplementary materials to specify the datasets used.
- We performed reproducibility analysis of CELLECT's linking confidence across multiple models, confirming consistent top-scoring matches even when the temporal resolution was reduced by a factor of 3 (i.e., evaluating matches every 3 frames), with slight drops during mitosis (**Methods; Fig. R10**).
- We compared CELLECT with Zyss et al. (2024), emphasizing differences in annotation needs, objectives, and application focus. CELLECT is optimized for sparse-supervised 3D tracking, while Zyss et al. target lineage reconstruction from dense segmentation (**Methods**).

In the following sections, we address each specific concern raised by the reviewer to further enhance the manuscript and clarify key aspects of our work.

Comment B: Originality and significance: if not novel, please include reference

The approach of contrastive learning to 3D cell tracking is novel, but relatively straightforward by combining leading methods from cell tracking and deep learning. The main improvement comes from the combination of sparse fluorescent labeling combining with the mentioned method to avoid a labor intensive labeling of 3D datasets.

Response:

We appreciate the reviewer's positive feedback. As the reviewer pointed out, CELLECT integrates deep learning and cell tracking techniques to achieve accurate 3D cell tracking from sparsely annotated datasets and reduces the end-to-end computational costs by orders of magnitude with a novel design of the whole framework. Notably, benefiting from our center-based contrastive learning and multidimensional feature optimization strategies, CELLECT achieves significant improvements in both accuracy and efficiency compared to existing deep-learning-assisted tracking methods, while substantially reducing the

annotation workload typically required for 3D datasets. Simultaneous fluorescent labeling of sparse and dense samples is also a new method used to evaluate the tracking performance experimentally without the requirement of labor-intensive labeling.

Comment C: Data & methodology: validity of approach, quality of data, quality of presentation

Major point:

Comment 1:

* It is not clear from the paper if the results were officially submitted at the Cell Tracking Challenge and are visible in the Leaderboard of the challenge (<https://celltrackingchallenge.net/latest-ctb-results/>). This is for my opinion mandatory because it shows the generalization on an unseen and hidden test dataset. Please add your team name, the places in the leaderboard for a defined date and the official code names of the dataset. It would be also interesting to quantify which subset of training data was used. It would also be useful to discuss the differences to the other TOP3 algorithms on the same dataset in detail, including a quantitative evaluation in Fig. 2 - especially in case of available code. And explain why you selected the 3 concurrent algorithms in Fig. 2.

Response:

We thank the reviewer for the constructive suggestion. We agree that official submission to the Cell Tracking Challenge (CTC) is essential for assessing generalizability on an unseen test dataset. We have officially submitted CELLECT to the CTC website under the team named as "THU-CN (3)", which was automatically generated by the platform to indicate the third submission from our affiliation (Tsinghua University, China). Please note that the superscript (3) in the score of THU-CN means that there are 2 earlier submissions under the same institution which were actually submitted several years ago by other research teams in Tsinghua University not our group. Our results became available on October 20, 2024, and are now visible on the CTC leaderboard, ranking first for both tasks of cell tracking and segmentation in the Fluo-N3DH-CE dataset in terms of the overall performance measure (OP_{CSB} and OP_{CTB}). We are still awaiting the update of our method description on the CTC website, which is not controlled by us and only accessible by the third-party team of CTC. They told us by email that it would be updated soon. The scores achieved by our method on this dataset have been included in the revised manuscript. The corresponding rankings are available on the official website (celltrackingchallenge.net) and are visualized in the figure below, with our team highlighted for clarity.

We chose the Fluo-N3DH-CE dataset for benchmarking of CELLECT, because the Fluo-N3DH-CE dataset is the only publicly available 3D time-lapse microscopy dataset that provides both manually annotated cell centers and algorithm-generated 3D segmentation masks in the training set, which itself is a good resource to train our pretrained model. In addition, this dataset includes a large number of 3D cell-division events, which are challenging for previous cell tracking algorithms. It's hard to compare all other top algorithms in detail because past CTC competitions did not require participants to release their training code, and we found that many top-performing methods either lack complete training pipelines or have become inaccessible. We believe the scores in the CTC operated by the third party reflect a direct quantitative evaluation of their performance. Both the training and evaluation pipelines of CELLECT used in the CTC submission have been officially validated through the CTC benchmark. To ensure transparency and reproducibility, we have also released the full codebase at: <https://github.com/zzz333za/CELLECT> and <https://github.com/zzz333za/CELLECT-ctc.ver> (This repository contains a dedicated branch that reproduces the exact configuration and results submitted to the CTC leaderboard.)

The reason that we selected the 3 concurrent algorithms in **Fig. 2** are listed below. First, Linajea (Malin-Mayor et al., *Nat. Biotechnol.* 2023) is one of the top 3 algorithms in CTC for the same track we submitted. As the reviewer mentions before, we try to add more detailed comparisons to these state-of-the-art algorithms. Secondly, the paper of Linajea uses 2 open-source datasets with the first 270 frames of each sub-volume annotated with manually verified center points. They also provide performance results of both the Linajea-series methods and StarryNite on this dataset, making it highly suitable for comparing tracking algorithms in similar settings. Thirdly, among the previously reported deep learning methods, only the Linajea series supports training based on sparse center annotations, which is a very good advantage for user-friendly applications, making it directly comparable to our approach. Our CELLECT also has this advantage with only the requirement of sparse annotations during training. Finally, we also want to include non-deep-learning algorithms without the requirement of training in comparisons. Therefore, we also included Imaris, a widely used commercial tool, as a baseline in comparisons.

In addition, the 2 datasets used in **Fig. 2** are among the very few 3D cell-tracking datasets that provide complete sparse trajectory annotations (center point annotations, as shown in **Table R1**). These datasets also contain several challenging properties, including low signal-to-noise ratio, substantial cell shape variation, and frequent cell division events. Therefore, they are particularly suitable for evaluating models designed to operate under sparse annotation in 3D settings.

We further provide a summary of related methods in **Table R2**, which shows that only a very limited number of algorithms support training with sparsely annotated data. Aside from Linajea and CELLECT, most existing approaches can only be applied either through pretrained models or without training at all. These include various detection methods

integrated into TrackMate; among them, only basic 3D-compatible options such as LoG, DoG, and MorphoLibJ are available (TrackMate currently does not support Cellpose for 3D). However, due to the aforementioned challenges in these datasets—low SNR, large morphological variation, and frequent divisions—such traditional methods tend to fail to produce stable tracking results, as illustrated in **Fig. R2a**. Their performance is substantially lower than CELLECT’s, and **Fig. R2b** provides a direct comparison on mskcc-confocal#2, where the error rates of these methods far exceed those of CELLECT. As a result, we did not include them as primary baselines in **Fig. 2**.

Although Cellpose supports 3D segmentation, it does not include a tracking module and thus struggles to handle long-range displacements or cell divisions. In our experiments, even using its most stable mode—ensemble averaging over 4 pretrained models—the segmentation remained inconsistent (**Fig. R2c**), and thus we did not consider it a viable baseline either.

Beyond these, we also considered 3D methods that ranked highly in the Cell Tracking Challenge (CTC). However, most of them either do not support sparse annotation for training or do not provide publicly available training code (**Table R3**), which makes comparison impractical. While the KTH-SE(1) method does not require training, it heavily relies on dataset-specific manual parameter tuning at each step of the pipeline. Therefore, in our benchmarking, we ultimately followed the selection criteria used in the Linajea paper and included methods that are compatible with sparse annotation and commonly used tools such as Imaris for practical comparisons.

We included all the details in the revised manuscript and new subsection entitled *Evaluation on Cell Tracking Challenge benchmark* to the **Methods** section.

T	X	Y	Z	CELLID	PARENTID	RADIUS
0	95.0	215.0	255.0	0	-1	13.5
0	95.0	306.0	365.0	1	-1	16.5
1	105.0	228.0	225.0	2	0	14.5
1	110.0	211.0	272.0	3	0	15.5
1	90.0	300.0	369.0	5	1	13.5

Table R1 | Example of sparse annotation for cell centers and lineage in training dataset. This table illustrates the training annotations used in our sparse-supervision setting.

Method (Publication)	3D Support	Sparse Annotation Support	Pretrained Model Available	3D Pretrained Model	Mitosis Tracking Capability	Usable for Comparison	Remarks
CELLECT (This study)	✓	✓	✓	✓	✓	✓	Fully applicable.
Linajea (Nat. Biotechnol. 2023)	✓	✓	✗	✗	✓	✓	Fully applicable.
Imaris	✓	✗	✓	✓	✓	✓	Usable with manual configuration.
StarryNite (PNAS 2006)	✓	✗	✗	✗	✓	✓	Usable with manual curation.
TrackMate-LOG	✓	✓	✗	✗	✓	✓	Usable with tuned parameters.
TrackMate-DOG	✓	✓	✗	✗	✓	✓	Usable with tuned parameters.
TrackMate-MorphoLibj	✓	✗	✗	✗	✗	✓	Usable with tuned parameters.
TrackMate-Weka	Partial 3D support	✗	✗	✗	✗	✗	Not usable due to dense annotation requirement.
TrackMate-Cellpose	✓	✗	✓	✓	✗	✗	Not usable, TrackMate does not support 3D outputs from Cellpose.
Cellpose (Nat. Methods 2021)	✓	✗	✓	✓	✗	✓	Usable for segmentation-only comparison.
YeaZ (Nat. Commun. 2020)	✗	✓	✓	✗	✗	✗	Not usable due to 2D-only design.
DeepCell Kiosk (Nat. Methods 2021)	Partial 3D support	✗	✓	✗	✓	✗	Not usable due to lack of sparse and full 3D support.

Table R2 | Comparison of representative cell tracking methods. Note: Sparse annotation refers to partial cell trajectory center-point labels, without segmentation masks.

Figure R2 | Additional comparisons with other available methods not included in the main evaluation. **a**, Detection results using TrackMate combined with LOG, DOG, or MorphoLibJ detectors. Despite extensive manual parameter tuning, the outputs are still unsuitable for reliable tracking due to inaccurate or missing detections. **b**, Tracking comparison between TrackMate (with LOG/DOG detection) and CELLECT on the same sequence. **c**, Segmentation results obtained with the Cellpose nuclei model using the best-performing configuration (ensemble of 4 pretrained models). A large number of cells remain undetected, particularly in high-density regions or when multiple cells overlap along the z-axis. **d**, Circular tree diagrams showing representative error types and trajectories for 3 different methods.

Method	Training Code Availability	Segmentation OP	Tracking OP	Applicable for Sparse Annotation	Reason for Exclusion
THU-CN(3) (CELLECT)	Fully open source	0.853 (Highest)	0.850 (Highest)	✓	
CZB-US	Partially open source	0.847	0.844	✗	Not open-source yet
KTH-SE(1)	Open source	0.842	0.829	✗	Requires extensive manual parameter tuning and internal selection; difficult to generalize to new datasets
KIT-GE(2)	Partially open source	0.830	0.808	✗	Does not support sparse annotation; requires full segmentation labels for training

Table R3 | Summary of selected Cell Tracking Challenge (CTC) methods and their applicability for comparison in Fig. 2. This table summarizes CTC methods that are applicable to 3D datasets and for which publicly available implementations or explanatory documentation exist. Segmentation and tracking OP values are reported from the CTC leaderboard on the Fluor-N3DH-CE dataset. ✓ indicates compatibility with sparse annotation.

Comment 2:

* Fig. 1 Why is necessary to work with 2 separate MLPs (inter-frame vs. inter-frame) instead of integrating these tasks in one network architecture with multiple outputs including track linkage?’’

Response:

We thank the reviewer for this insightful question. The intra-frame and inter-frame MLPs in CELLECT are specifically designed to address different problems with minimized

computational costs. Integrating all tasks into a single end-to-end network often leads to limited flexibility when handling varying input sizes or cross-patch inference, and can impose additional computational and memory overhead. To avoid these issues, CELLECT employs 2 individual light-weight MLPs (**Fig. R3, Supplementary Fig. 5** in the revised manuscript) that share the same main-branch feature extraction network as input for 2 different functions, while significantly reducing the overall computational burden.

Specifically, intra-frame MLP operates only the feature within a single frame, aiming to suppress redundant or overlapping cell center predictions. It treats any 2 objects in the same frame—even if they originate from the same dividing cell—as distinct cells, as long as they appear at the same time. In contrast, inter-frame MLP is used for linking cells between adjacent frames. It leverages the output of the division predictor to determine whether one cell in current frame t should be linked to 1 or 2 cells in next frame $t+1$. In this case, daughter cells resulting from division are considered the same cell lineage and are grouped accordingly during linking.

Moreover, we deliberately kept these 2 modules separate, not only due to their different functions and input distributions, but also to enhance flexibility and scalability. In particular, the MLPs are designed to operate outside the core feature extraction pipeline (3D U-Net and CEN modules) of CELLECT. This choice is critical for large-scale datasets, where it is often necessary to process the input in spatial patches. Embedding the MLPs within the 3D U-Net would make it difficult to manage linkages across patch boundaries. By decoupling them, we enable efficient patch-wise processing, multi-threaded parallelism, and independent reuse of the feature extraction module during data reconstruction.

Furthermore, we intentionally avoided designing a monolithic architecture that integrates temporal modeling directly into the core network. This decision is motivated by the diversity in temporal resolutions commonly observed in real-world datasets. Sequence-based tracking models, such as Linajea (which uses 15 frames per input), typically impose a significant computational cost and introduce biases toward the specific time intervals encountered during training. Consequently, such models often struggle with under-sampled or irregularly sampled data. In contrast, our modular and light-weight design is inherently more generalizable and robust across different temporal scales, offering practical advantages for large-scale 3D imaging applications.

To better clarify this design choice, we have added a simplified explanation in the **Main**, a detailed explanation in the revised **Methods** section under *Model architecture* and revised **Supplementary Fig. 5**, outlining the functional differences between the 2 MLPs, their input structures, and the rationale for their modular separation from the main network backbone.

Figure R3 (Supplementary Figure 5 in the revised manuscript) | Illustrations of the intra-frame MLP and inter-frame MLP. a, Illustration of the workflow of 2 MLPs. The 5 nearest candidate cells to the target cell are selected based on the spatial distances between cell centers. For intra-frame MLP, we selected the 5 candidate cells within the same time frame. For inter-frame MLP, we selected the 5 candidate cells in the next frame closest to the target cell in the current frame. For each candidate, four feature distances relative to the target cell are computed: (1) size and division probability difference, (2) size and division probability ratio, (3) Euclidean distance, and (4) feature similarity. These metrics serve as inputs to the MLPs.

The intra-frame MLP identifies redundant cell center point detections of the same cell, while the inter-frame MLP determines whether the target cell matches a candidate in the next frame or undergoes division. Each MLP outputs 6 similarity scores including 5 candidates and one ‘None’ class indicating that the target cell is not matched with any of the 5 candidates. The inter-frame MLP outputs 6 similarity scores and also provides a division prediction. **b**, Representative example of intra-frame MLP. For a given target cell (target 1), its six nearest cell center points (labeled 1-1 to 1-6) are identified within the same frame. We note that point 1-1 is a redundant detection of the same cell as the target. **c**, Intra-frame MLP outputs for different combinations of 5 out of the 6 candidates. Including the redundant detection (1-1) results in high confidence scores for that match, whereas its exclusion leads to the ‘None’ class having the highest score. **d**, Inter-frame MLP example for two target cells: target 2 (not dividing) and target 3 (undergoing division). Each is associated with 7 nearest candidates in the next frame. **e**, Inter-frame MLP outputs using three different candidate selection sets (e.g., 1-5, 2-6, and 3-7) for target cells 2 and 3. Inclusion of the correct corresponding cell (e.g., 2-1 for target 2) or daughter cells (3-1 and 3-2 for target 3) in the candidate set results in high similarity scores from the MLP. Exclusion of true matches causes the ‘None’ score to dominate. The division prediction is mainly driven by the input division probability and remains unaffected by candidate selection. The visualization in **(a)** is based on the Cell Tracking Challenge (CTC) Fluo-N3DH-CE dataset, while the examples in **(b–e)** are from sequence #2 of the mskcc-confocal dataset. All results were generated using a model trained on sequence #3 of the mskcc-confocal dataset.

Comment 3:

* In many applications, cell divisions are not the only event to qualitatively change cell tracks. How do you handle dying/apoptotic cells (especially with visible fragments) and cells leaving or entering the field of view? This could heavily influence the matching of cells with the MLPs in **Supplementary Fig. 5**.

Response:

We thank the reviewer for this valuable and multifaceted question. CELLECT tracks cells based on their extracted features and does not explicitly detect cell viability or death. While it is theoretically possible to infer apoptosis from morphological or dynamic patterns, this aspect lies beyond the current scope of our study. Notably, we did not observe substantial apoptotic events in the datasets used. When such cells retain fluorescence, they typically exhibit passive displacement driven by fluid motion rather than active migration. This behavioral difference may be inferred by comparing background drift patterns with

individual trajectories. However, robust identification of apoptotic or dying cells would require future work specifically focusing on biological interpretation.

For cells leaving the field of view (FOV), CELLECT evaluates candidates in the subsequent frame using an MLP. If none of the candidates surpasses the matching threshold, and all predicted scores fall below the score of the “None” class, the track is terminated. For cells entering the FOV, we apply a conservative decision rule: a newly detected object is only confirmed as a valid cell if it persists for at least 2 consecutive frames. This strategy helps prevent erroneous matches caused by noise or transient particles. We have added this explanation in the revised version.

The MLP-based matching mechanism (**Fig. R3, Supplementary Fig. 5** in the revised manuscript) is designed to flexibly handle such cases. Specifically, the MLP determines whether a target cell in the current frame should be linked to 1 of the nearest 5 candidate cells in the next frame or classified as unmatched. It generates 6 output scores- 5 corresponding to candidate cells and 1 representing the unmatched (“None”) class. This relative scoring framework allows the model to handle complex biological scenarios, including cell disappearance, fragmentation, and passive drifting, without relying on fixed thresholds or rule-based linkage. We have added more examples to show these phenomena (**Fig. R3b-e**).

We have revised the **Methods** section under *Model architecture* and updated **Supplementary Fig. 5** to clarify this decision process.

Figure R3 (Supplementary Figure 5 in the revised manuscript) | Illustrations of the intra-frame MLP and inter-frame MLP. a, Illustration of the workflow of 2 MLPs. The 5 nearest candidate cells to the target cell are selected based on the spatial distances between cell centers. For intra-frame MLP, we selected the 5 candidate cells within the same time frame. For inter-frame MLP, we selected the 5 candidate cells in the next frame closest to the target cell in the current frame. For each candidate, four feature distances relative to the target cell are computed: (1) size and division probability difference, (2) size and division probability ratio, (3) Euclidean distance, and (4) feature similarity. These metrics serve as inputs to the MLPs.

The intra-frame MLP identifies redundant cell center point detections of the same cell, while the inter-frame MLP determines whether the target cell matches a candidate in the next frame or undergoes division. Each MLP outputs 6 similarity scores including 5 candidates and one ‘None’ class indicating that the target cell is not matched with any of the 5 candidates. The inter-frame MLP outputs 6 similarity scores and also provides a division prediction. **b**, Representative example of intra-frame MLP. For a given target cell (target 1), its six nearest cell center points (labeled 1-1 to 1-6) are identified within the same frame. We note that point 1-1 is a redundant detection of the same cell as the target. **c**, Intra-frame MLP outputs for different combinations of 5 out of the 6 candidates. Including the redundant detection (1-1) results in high confidence scores for that match, whereas its exclusion leads to the ‘None’ class having the highest score. **d**, Inter-frame MLP example for two target cells: target 2 (not dividing) and target 3 (undergoing division). Each is associated with 7 nearest candidates in the next frame. **e**, Inter-frame MLP outputs using three different candidate selection sets (e.g., 1-5, 2-6, and 3-7) for target cells 2 and 3. Inclusion of the correct corresponding cell (e.g., 2-1 for target 2) or daughter cells (3-1 and 3-2 for target 3) in the candidate set results in high similarity scores from the MLP. Exclusion of true matches causes the ‘None’ score to dominate. The division prediction is mainly driven by the input division probability and remains unaffected by candidate selection. The visualization in **(a)** is based on the Cell Tracking Challenge (CTC) Fluo-N3DH-CE dataset, while the examples in **(b–e)** are from sequence #2 of the mskcc-confocal dataset. All results were generated using a model trained on sequence #3 of the mskcc-confocal dataset.

Comment 4:

* There were other papers in the past that worked with embeddings and real-time approaches, e.g. [Neven 2018] to estimate the position of the corresponding cell center followed by clustering approaches in instance segmentation:

Neven, D., Brabandere, B. D., Proesmans, M., & Gool, L. V. (2019). Instance segmentation by jointly optimizing spatial embeddings and clustering bandwidth. In Proceedings of the IEEE/cvf conference on computer vision and pattern recognition (pp. 8837-8845).

I would recommend to give a short overview about such approaches (including similar ideas for direct clustering) and compare the chosen approach with such papers.

Response:

We thank the reviewer for highlighting this relevant prior work. We have added a short overview and comparison of these approaches mentioned by the reviewer in the revised version.

The method proposed by Neven et al. (CVPR 2019) and Neven et al. (IEEE IV 2018, Neven, D., De Brabandere, B., Georgoulis, S., Proesmans, M. & Van Gool, L. Towards end-to-end lane detection: an instance segmentation approach. IEEE Intell. Vehicles Symp. (IV) 286–291 (2018)) shares some conceptual similarities with ours: both aim to represent object instances via central points and learned embeddings, thereby avoiding traditional post-processing steps such as object proposals and non-maximum suppression (NMS: a post-processing technique used to eliminate redundant or overlapping detections by keeping only the most confident prediction), and enabling a more efficient, real-time framework. However, there are fundamental differences that make their approach less applicable to the application of 3D cell tracking. First, annotation granularity and learning direction: Neven et al.’s method relies on dense pixel-level supervision. Their model learns per-pixel embeddings and offset vectors based on fully labeled images—leveraging complete object masks to infer center points through clustering. In contrast, our model starts with only sparsely labeled center points. We use contrastive learning to enforce feature dissimilarity between different cell centers, and from this sparse supervision, we infer pixel-level segmentation masks around each center using learned embeddings and estimated radii.

In short: their model learns center points from the whole image; our model learns object-level structure from center points. Second, applicability to 3D microscopy data: Unlike natural images, 3D cell imaging data is characterized by anisotropic resolution, cell movement and division, and extremely limited annotations. These challenges make pixel-supervised clustering approaches less effective or almost infeasible. Our center-based approach better fits the sparse and dynamic nature of the task.

We have added a dedicated subsection titled *Comparison with embedding-based and contrastive learning methods* in the revised **Methods** section, providing a conceptual and practical comparison between CELLECT and representative prior works such as Neven et al. and Zyss et al.

Comment 5:

* How does the approach solve missing segmentations (e.g. very low confidence values) at time point $t+1$ leading to broken linkages? Other clustering approaches are looking for candidate segmentations subsequent time points as $t+2$ and $t+3$ and uses these segmentations to repair missing $t+1$ - segmentations.

Response:

We thank the reviewer for this insightful question. We agree that missing or low-confidence segmentations at intermediate frames (e.g., time $t+1$) can lead to broken tracking links, and handling such cases is critical for real-world applications.

We solve this problem by trying to avoid missing segmentations in the CELLECT framework. Our approach uses local-peak-based confidence evaluation, where center detection is guided by the relative contrast between a point and its surrounding

neighborhood. As a result, the system is generally more prone to redundant detections of center points instead of missing ones. However, in the specific challenging scenarios—such as those involving extremely low signal-to-noise ratios or flickering signals—missing detections can still occur. In such cases, we can perform additional post-processing to reconnect disjointed tracks by considering matches across non-adjacent frames as suggested by the reviewer. Actually, our CELLECT method is compatible with multiple-frame linking. For instance, in the task of neuronal calcium activity tracking in the deforming brain tissue of *Drosophila* (**Fig. 5** in the original manuscript). In general, calcium signal intensity of the neuron is highly flickering and the baseline intensity is low. In this scenario, we implemented a post-hoc strategy that searches within ± 200 frames for re-appearing cells in the same spatial neighborhood. If a consistent match is found, we bridge the trajectory through the missing frame. We applied this temporal linking approach also to Imaris and TrackMate (within $\pm 2,000$ frames) in the same scenario to ensure a fair comparison. Importantly, this additional bridging strategy is only applied in this experiment. All other results in the manuscript rely solely on CELLECT's default frame-by-frame inference, without extended temporal linking.

We have clarified this problem in the revised **Methods** section under the subsection *Temporal linking with larger intervals*.

Comment 6:

* How do you guarantee that the features for the sparse annotations are representative for the other cells?

Response:

We thank the reviewer for this insightful question. We agree that under sparse supervision, it is inherently difficult to guarantee that the extracted features are globally representative of all cell appearances. In our framework, this limitation arises because the annotations are limited to a small number of center points, which constrains the completeness of feature extractor training.

However, CELLECT is intentionally designed not to rely on global representativeness. Instead, our focus is on enhancing feature discriminability between adjacent cells within local spatial and temporal contexts. We formulate this as a local contrastive learning problem, where the goal is to ensure that spatially adjacent but biologically distinct cells exhibit distinct and separable features. We use the sparse annotations to find the embedding space not specific features.

Due to input size constraints, each training iteration operates on a fixed-size patch ($256 \times 256 \times 32$), and thus the contrastive learning primarily focuses on enhancing feature separability within local regions rather than enforcing global feature consistency. Based on this setting, we employ a triplet-based contrastive loss to promote instance-level feature discriminability within the local neighborhood. Specifically, for each annotated or detected center point, we define a local positive region around it and, across 2 consecutive frames,

identify the hardest negative (i.e., the nearest neighboring cell from a different instance) and the least similar positive (i.e., the region within the same instance exhibiting the lowest feature similarity). By optimizing the embedding distances among these samples, the model learns to effectively distinguish between different cell instances within local spatial and temporal contexts. Additionally, CELLECT incorporates classification losses from 2 auxiliary MLP modules, which are trained jointly with the backbone feature extractor to further enhance the expressiveness of center embeddings and reinforce local contrast in feature space. This design is particularly effective in dense or ambiguous regions, where precise feature distinction is crucial for maintaining correct identity associations.

To further address the limitation of sparse training, we adopt a spatial distance-based strategy to exploit the prior knowledge that the cell closer to the target cell has higher probability to be the candidate cell during linkage when selecting candidate cells. As the number of cells increases, insufficient supervision can lead to misleading feature similarities between distant cells, which interfere with correct candidate selection. To address this, we select candidates based on spatial proximity first, and then evaluate them using feature similarity. As shown in **Fig. R4 (Supplementary Fig. 11** in the revised manuscript), feature-based candidate selection performs reliably in low-density settings and is less sensitive to temporal resolution changes. However, in high-density regions, it results in significantly more errors. In contrast, our spatial distance-based strategy, followed by embedding evaluation, consistently achieves more stable performance under diverse challenging conditions.

We have clarified this design principle of CELLECT in the revised manuscript and **Methods** section under subsection *Model architecture* and added a **Fig. R4 (Supplementary Fig. 11** in the revised manuscript).

a Inference across different frame rates

Figure R4 (Supplementary Figure 11) | Influence of imaging frame rate on tracking performance under different candidate selection strategies. **a**, Illustration of 3 different levels of temporal down-sampling were used for evaluation: $\times 1$ (75-s interval), $\times 2$ (150-s interval), and $\times 3$ (225-s interval) down-sampling. **b**, Quantitative tracking performance of CELLECT under 3 conditions of temporal down-sampling. **c**, Schematic illustration of 2 cell selection strategies: left, spatial-distance-based selection of the 5 nearest neighbors (used in CELLECT); right, feature-distance-based selection of the 5 most similar feature vectors. **d**, Error curves (FN + FND + IS) over time for both strategies under different temporal down-sampling. The feature distance-based strategy performs comparably at early frames when cell density is low, even under reduced temporal resolution, but its performance degrades rapidly as cell density increases over time. **e**, Circular tree diagrams displaying tracking errors across all 6 settings. All evaluation data were drawn from mskcc-confocal dataset #2; the model was trained on mskcc-confocal dataset #3 without temporal down-sampling. Scale bar: 10 μm .

Comment 7:

* Please explain in all figure captions which datasets are used.

Response:

We thank the reviewer for the helpful suggestion. We have carefully revised the figure captions throughout the main manuscript and supplementary materials to explicitly indicate the dataset used in each figure.

- **Fig. 1:** “Panels (a–c) use sample data from the Cell Tracking Challenge (CTC) Fluoro-N3DH-CE dataset. Panel (d) uses nuclei-labeled data from the mskcc-confocal dataset and membrane-labeled neutrophil data from mouse imaging (same as in **Fig. 4**). The model used in all examples was trained on sequence #3 of the mskcc-confocal dataset.”
- **Fig. 2:** “All evaluations were conducted using the first 270 frames of each of the 3 sub-volumes for both datasets (mskcc-confocal and nih-light sheet), following a cross-validation protocol. The CELLECT models used in each comparison were trained on one of the 2 other sub-volumes. Further details on the training and evaluation splits are provided in Supplementary Table 1 and 2.”
- **Fig. 3:** “All data were acquired from live imaging of mouse lymph nodes using two-photon scanning adaptive microscopy (2pSAM). Panels (a–h) use dual-labeled T cell datasets, while panels (i–m) use B cell datasets during germinal center formation. All results were obtained using the CELLECT model trained on sequence #3 of the mskcc-confocal dataset.”
- **Fig. 4:** “All data were acquired by live 2pSAM imaging of surgically exposed mouse spleen over a 4-hour timespan. CELLECT results were obtained using the model trained on sequence #3 of the mskcc-confocal dataset.”

- **Fig. 5:** “All data were acquired from live 2pSAM imaging of GCaMP6s-expressing *Drosophila* brains during spontaneous activity. CELLECT results were obtained using the model trained on sequence #3 of the mskcc-confocal dataset.”
- **SFig. 1:** “The network outputs shown here were generated from the Cell Tracking Challenge (CTC) Fluo-N3DH-CE dataset. The model used was trained on sequence #3 of the mskcc-confocal dataset.”
- **SFig. 2:** “Panels (a–f) use data from the Cell Tracking Challenge (CTC) Fluo-N3DH-CE dataset. Panels (g–j) use nuclei-labeled data from the mskcc-confocal dataset and membrane-labeled neutrophil data from mouse imaging, as also shown in **Fig. 4**. All results were generated using a CELLECT model trained on sequence #3 of the mskcc-confocal dataset.”
- **SFig. 3:** “The example shown here uses data from the Cell Tracking Challenge (CTC) Fluo-N3DH-CE dataset. The model used for feature extraction was trained on sequence #3 of the mskcc-confocal dataset.”
- **SFig. 4:** “The visualization shown here is based on the Cell Tracking Challenge (CTC) Fluo-N3DH-CE dataset. The model used was trained on sequence #3 of the mskcc-confocal dataset.”
- **SFig. 5:** “The visualization shown here is based on the Cell Tracking Challenge (CTC) Fluo-N3DH-CE dataset. The model used was trained on sequence #3 of the mskcc-confocal dataset.”
- **SFig. 7-13:** “All evaluation data in this figure were drawn from mskcc-confocal dataset #2. All models were trained using mskcc-confocal dataset #3.”
- **SFig. 14:** “All 5 models were independently trained on mskcc-confocal dataset #3.”
- **SFig. 15:** “All imaging data were acquired using 2pSAM microscopy of mouse lymph nodes, consistent with the dataset used in **Fig. 3**. CELLECT results were generated using the model trained on sequence #3 of the mskcc-confocal dataset.”
- **SFig. 16-17:** “All data were acquired using 2pSAM imaging of B cells in mouse lymph nodes, consistent with the dataset used in **Fig. 3**. CELLECT results were generated using the model trained on sequence #3 of the mskcc-confocal dataset.”
- **SFig. 18:** “All data were acquired using 2pSAM imaging of B-cell dynamics in mouse lymph nodes. These datasets are distinct from the one shown in **Fig. 3**, but were collected under identical imaging conditions and protocols. The total data volume exceeds 4 TB. CELLECT results were generated using the model trained on sequence #3 of the mskcc-confocal dataset.”
- **SFig. 19:** “The data used here were acquired via 2pSAM imaging of neutrophils, macrophages, and bacteria in mouse spleen, consistent with the dataset shown in **Fig. 4**. CELLECT results were generated using the model trained on sequence #3 of the mskcc-confocal dataset.”
- **SFig. 20:** “The data used here were acquired via 2pSAM imaging of neutrophils, macrophages, and bacteria in mouse spleen, consistent with the dataset shown in **Fig.**

4. CELLECT results were generated using the model trained on sequence #3 of the mskcc-confocal dataset.”

- **SFig. 21:** “This data was validated using the mskcc-confocal dataset #2, which explains the observed differences compared to the cross-validation results reported in the main text. CELLECT results were generated using the model trained on sequence #3 of the mskcc-confocal dataset.”

Minor point:

Comment 1:

* I would recommend to state earlier (even in the abstract) that the sparse annotations stem from different fluorescent labels. Otherwise, sparse annotation could be misinterpreted as manual labeling.

Response:

We thank the reviewer for the constructive comment and sincerely apologize for the ambiguity in terminology. In our manuscript, the term "sparse annotation" is used in 2 distinct contexts:

- Manually annotated sparse labelling – used during training, where a small subset of cell centers and trajectories are manually labeled, without full segmentation masks. (**Table. R1**) Only sparse annotation is required to train the network, which makes the algorithm user-friendly without the requirement of full annotations.
- Biological marker expressed with a low density – used in experimental evaluation (**Fig. 3a-f** in the original manuscript), where only a subset of cells is visible due to selective expression of fluorescent reporters. This method is used to generate groundtruth in experimental data for benchmarking.

To better clarify, we have distinguished in the revised manuscript that "sparse annotation" refers to manual sparse labeling during training, whereas "sparse labeling" refers to selective fluorescent labeling during experimental evaluation. An example of manual sparse annotations is provided in **Table R1**.

In **Fig. 3a-f** in the original manuscript, we evaluated CELLECT on a simultaneously acquired two-fluorescence-channel dataset labelled with sparsely and densely expressing fluorescence markers, respectively. One with sparse lineage-specific expression and another with dense nuclear labeling. This setup enabled a direct comparison of the same cells under sparse and dense cell imaging conditions. The cells with sparse labeling can be used as groundtruth based on the assumption that it is much easier to track sparse samples across a large field of view due to less ambiguity of the cell linking. CELLECT showed similarly stable tracking performance in both densely-labeled samples and sparsely-labelled samples, demonstrating its robustness to varying cell densities and marker sparsity.

We have now updated in the main texts and clarified this distinction in the caption of **Fig. R5 (Fig. 3 in the revised manuscript)**.

T	X	Y	Z	CELLID	PARENTID	RADIUS
0	95.0	215.0	255.0	0	-1	13.5
0	95.0	306.0	365.0	1	-1	16.5
1	105.0	228.0	225.0	2	0	14.5
1	110.0	211.0	272.0	3	0	15.5
1	90.0	300.0	369.0	5	1	13.5

Table R1 | Example of sparse annotation for cell centers and lineage in training dataset. This table illustrates the training annotations used in our sparse-supervision setting.

Figure R5 (Figure 3 in the revised manuscript) | 3D tracking of dense immune cells in mouse lymph nodes. **a**, Example orthogonal maximum intensity projections (MIP) of dual-color labeled T cell imaged by 2pSAM in mouse lymph nodes. T cells were labeled with both DsRed and GFP at a 1:5 ratio. Sparsely labeled cells are generally co-labeled with GFP to be used as ground-truth during evaluation. **b**, Tracking result of the dense GFP channel obtained by CELLECT. **c**, Tracking result of the sparse DsRed channel obtained by Imaris using as the ground-truth. **d**, Tracking result of the dense GFP channel obtained by Imaris. **e**, Accurately tracking traces in the dense channel obtained by CELLECT (about 91.5% of all traces). We define the overlap ratio of the total tracking length compared with the ground-truth trace

in the sparse channel larger than 0.8 as the accurately tracked cell. **f**, Accurately tracking traces in the dense channel obtained by Imaris (about 70.7% of all traces). **g**, Comparison of the traces of the same cell in dense datasets obtained by CELLECT and Imaris. While Imaris cannot track the same cell consistently, CELLECT can track in a continuous way. **h**, Bar chart showing the comparisons of the tracking accuracy under different overlap thresholds used to define accurately tracked cells. **i**, Example orthogonal MIPs of B cells in mouse lymph nodes during the formation process of a GC. **j**, 3D tracking traces obtained by CELLECT with different colors corresponding to different time stamps. **k**, Spatiotemporal distribution of the events of cell division (corresponding to each dot) identified by CELLECT, indicating that a large amounts of cell division happen during GC formation. **l**, Comparison of the traces of the same cell obtained by CELLECT and Imaris, indicating a cell-division event. **m**, Comparison of cell proliferation and accumulated cell division events over time in both GC and non-GC regions. Note: The term ‘sparse’ in this figure refers to biological sparsity due to selective DsRed expression in a subset of cells, not to sparsity in manual annotations used during model training. All data were acquired from live imaging of mouse lymph nodes using two-photon scanning adaptive microscopy (2pSAM). Panels (**a–h**) use dual-labeled T cell datasets, while panels (**i–m**) use B cell datasets during germinal center formation. All results were obtained using the CELLECT model trained on sequence #3 of the mskcc-confocal dataset. Scale bars: 100 μm .

Comment 2:

Explain what csc and sSVM (support vector machine?) means in Linajea+csc+sSVM to give readers insights into concurrent algorithms without checking the references.

Response:

We thank the reviewer for the thoughtful comment. ‘Linajea + csc + sSVM’ is an enhanced version of the original Linajea framework, designed to improve tracking performance by adding 2 modules. ‘csc’ stands for cell state classifier, which predicts biological states (e.g., parent, daughter, continuation, polar body) to refine the modeling of cell-division event. ‘sSVM’ stands for structured support vector machine, which learns the cost function weights for tracking using structured output loss, instead of manual tuning.

We have added brief explanation of both csc and sSVM in the main texts and in the caption of **Fig. R6 (Fig. 2)**.

Figure R6 (Figure 2) | Benchmarking of CELLECT with state-of-the-art algorithms in Cell Tracking Challenge. a-d, Performance evaluation on the dataset of mskcc-confocal. **e-h**, Performance evaluation on the dataset of nih-light sheet. **a** and **e**, The average number of errors per 1,000 ground-truth edges for each error type. The errors were categorized into 5 conditions: false-positive edges, false-negative edges, identity switches, false positive divisions, and false negative divisions. **b** and **f**, The proportion of error-free tracks (marked as tracking accuracy) versus a tracking duration in terms of frames. **c** and **g**, Comparisons of average computational time cost for each frame among different algorithms. **d** and **h**, The hierarchical circular diagram of lineage tracing with the grey lines indicating the cell division. The bold black lines and circles indicate the false negative identity switch and division, respectively. The scores for the 3 models other than CELLECT and the accuracy curves of Linajea are derived from the previous report³⁵. ‘Linajea + csc + sSVM’ refers to an enhanced version of Linajea that incorporates a cell state classifier (csc) to model biological states (e.g., division, continuation, polar body) and a structured Support Vector Machine (sSVM) to learn tracking cost functions from data. All evaluations were conducted using the

first 270 frames of each of the 3 sub-volumes for both datasets (mskcc-confocal and nih-light sheet), following a cross-validation protocol. The CELLECT models used in each comparison were trained on one of the 2 other sub-volumes. Further details on the training and evaluation splits are provided in **Supplementary Table 1** and **2**. Scale bars: 10 μm .

Comment 3:

* The CEN should be explicitly mentioned in Fig. 1, with a reference to Supp Fig. 1.

Response:

We thank the reviewer for the constructive comment. We have revised accordingly by mentioning the CEN module explicitly in **Fig. R7 (Fig. 1** in the revised manuscript), and added reference in the caption for **Supplementary Fig. 1** where the CEN is explicitly described.

Figure R7 (Figure 1 in the revised manuscript) | Schematic of CELLECT.

a, We trained the feature extraction model using confidence maps generated from 2 adjacent frames with the sparse annotations of center points. For each voxel in the first frame, the model outputs an embedding vector, division prediction probabilities, and confidence map predictions. Using the confidence map, the model further derives cell segmentation, size predictions, and an enhanced confidence map for center point regions. Peaks in the enhanced confidence map are selected as masks for cell center points. The features are extracted by masking the feature map with the center-point mask to reduce the computational costs. **b**, Schematic of the training procedure. The model is designed to extract center point coordinates and feature map from the input data using the confidence maps generated from sparse

annotations as ground-truth. By combining the embedding vectors of different cell center points with contrastive learning, the model minimizes feature distances for the same cell while maximizing feature distances between different cells. **c**, Examples of center point and segmentation obtained via Imaris and CELLECT on nuclei-labeled cells captured by confocal laser scanning microscopy (CLSM, left) and membrane-labeled cells captured by scanning light-field microscopy (sLFM, right). **d**, Schematic of the inference procedure for cell tracking. The model extracts center points together with paired feature maps from the input data. For each cell, the model first removes redundant cell centers by merging the cells corresponding to the same cell. The merging process is conducted within 1 frame to identify the 5 nearest candidate cells for each target cell and calculates their feature distances, which are then input into an intra-frame multi-layer perceptron (MLP) to determine whether they belong to the same cell. Then we conduct cell linking in the inter-frame process by identifying 5 nearest cells across 2 frames and uses an inter-frame MLP to determine whether it correspond to the same cell in the first frame or correspond to a cell-division event, or another cell. The Center Enhancement Network (CEN), which refines the confidence map for more precise center localization, is included in the pipeline and detailed in **Supplementary Fig. 1**. Panels (**a–c**) use sample data from the Cell Tracking Challenge (CTC) Fluo-N3DH-CE dataset. Panel (**d**) uses nuclei-labeled data from the mskcc-confocal dataset and membrane-labeled neutrophil data from mouse imaging (same as in **Fig. 4**). The model used in all examples was trained on sequence #3 of the mskcc-confocal dataset. Scale bars: 1 μm for CLSM images and 5 μm for sLFM images (**c**).

Comment 4:

* 480/481 "The obtained features were subsequently input into an MLP network for further classification." What do you mean with the "obtained features"? The embeddings?

Response:

We thank the reviewer for pointing out this ambiguity. We apologize for any confusion. In this context, the “obtained features” refer to the output of the main branch of CELLECT, a composite feature vector constructed for each detected center, which includes—but is not limited to—the following components:

- The embedding vector extracted from the local neighborhood,
- The estimated probability of division,
- Size information inferred from the surrounding voxel region, and
- Geometric relationships between cells (e.g., relative distances and directions).

These combined features are then passed into the intra-frame MLP and the inter-frame MLP as described before.

To clarify, we have revised the sentence (originally line 480–481) to:

“The obtained features, including embedding vector offsets, spatial distances, estimated size differences, and changes in predicted division probabilities, were subsequently input into the intra-frame and inter-frame MLP networks for further classification.”

Comment 5-8:

* 496/497 duplicated text: represent points from distinct labeled cells. represent points from distinct labeled cells

* 789 captured by confocal microscopy (CLSM, left) -> captured by confocal laser scanning microscopy (CLSM, left)

* 836 faciliate -> facilitate

* Fig. 4g neutrophil -> neutrophil

Response:

We thank the reviewer for carefully pointing out these typos. We have revised accordingly.

Comment 9

Supp Fig. 1 Add the dimensions of the input figure and move all dimension numbers to the related block. Otherwise, a misunderstanding is possible that 8x256x256x32 belongs to the input dimensions.

Response:

We appreciate your valuable suggestions regarding **Supplementary Fig. 1**.

We clarified the dimension of the input. We confirm that the input size is 2×256×256×32, corresponding to 2 consecutive 3D image frames. To avoid confusion, we have updated the figure by placing all dimension annotations directly next to the respective blocks.

Comment 10

Supp Fig. 1: Explain, why the confidence map in the output channel is 4-dimensional. And why the center point location is 5-dimensional (the four levels + background?). I would recommend to add a supplementary table containing all dimensions of input and output channels per neural network.

Response:

We appreciate your valuable suggestions regarding **Supplementary Fig. 1**.

The output of the CEN module contains 5 channels, representing 4 center probability levels plus 1 background channel, as you assumed. We have clarified this in the figure caption and added a **Table R4 (Supplementary Table 3)** to explicitly describe all input/output dimensions for each module. The 4-channel confidence map output from the 3D U-Net backbone corresponds to multi-scale segmentation confidence maps, and is used

for region segmentation and cell-size estimation across 4 spatial resolution levels. We also sincerely appreciate your recommendation for the dimension table.

As the reviewer recommended, we have added a **Table R4 (Supplementary Table 3** in the revised manuscript), which summarizes the input and output dimensions and feature types across modules, including MLP stages and embedding operations. Also, we updated **Fig. R8 (Supplementary Fig. 1** in the revised manuscript) and the subsection titled *Model architecture* in **Methods**.

Figure R8 (Supplementary Figure 1 in the revised manuscript) | Detailed network framework of CELLECT. The overall framework of the CELLECT model is illustrated; The upper-left section depicts the main network structure, a 3D U-Net, with data dimensions and arrows representing the data flow and connection between modules. The color of the arrows corresponds to the network modules shown in the at the left bottom panel. The main model generates 3 outputs: cell division predictions, a confidence map, and a 64-dimensional embedding map, all maintaining same spatial resolution as the input, derived from 2 frames of input images. The confidence map serves as the basis for cell segmentation and size estimation. The lower-right section introduces the Center Enhancement Network (CEN), a light-weight 3D U-Net model. This network takes the confidence map and the 2 original image frames as inputs, refining the probability distribution by focusing on high-probability regions around cell center points while suppressing probabilities in other areas. The peaks in the confidence map generated by the CEN correspond directly to precise center point locations. This framework enables high-accuracy cell division prediction, segmentation, and size estimation while significantly improving the precision of center point localization. The network outputs shown here were generated from the Cell Tracking Challenge (CTC) Fluo-N3DH-CE dataset. The model used was trained on sequence #3 of the mskcc-confocal dataset.

Module	Input Size	Description	Output Size	Description
3D U-Net	$2 \times 256 \times 256 \times 32$	Two consecutive 3D image frames	$4 \times 256 \times 256 \times 32$	Confidence map (background/embryo /polar body/size)
			$2 \times 256 \times 256 \times 32$	Division probability map (2-classes)
			$64 \times 256 \times 256 \times 32$	Embedding map
CEN	$6 \times 256 \times 256 \times 32$	Two frames + confidence map	$5 \times 256 \times 256 \times 32$	Confidence map (center-enhanced, 4 levels + background)
Intra-frame MLP	64×1	Current cell embedding	6×1	5 candidates + 'None' scores
	64×5	Embeddings of 5 candidates		
	3×5	Spatial distance to candidates		
	6×5	Difference and ratio of size and spatial distance		
	3×5	Difference and ratio of division probability		
Inter-frame MLP	64×1	Current cell embedding	6×1	5 candidates + 'None' scores
	64×5	Embeddings of 5 candidates		
	3×5	Spatial distance to candidates		
	6×5	Difference and ratio of size and spatial distance	2×1	Division probability
	3×5	Difference and ratio of division probability		

Table R4 (Supplementary Table 3) | Input and output tensor dimensions for each module.

Comment 11

Supp Fig. 1 Is it really a blue + magenta arrow in the decoder path (lowest element) of the U-net?

Response:

We thank the reviewer's suggestions regarding **Supplementary Fig. 1** and apologize for any confusion. As the reviewer correctly pointed out, there was an omission in the depiction of the decoder path in **Supplementary Fig. 1** in the original manuscript. Specifically, after the upsampling step (blue arrow) and the convolution operation (magenta block), the decoder output is merged with features from the corresponding encoder level via a horizontal skip connection. In the original version, the black arrow indicating this skip connection was mistakenly omitted. We have corrected this in the **Fig. R8 (Supplementary Fig. 1)** in the revised manuscript) and revised the figure by restoring the missing black arrow, which now clearly illustrates the merging operation between encoder and decoder branches.

Figure R8 (Supplementary Figure 1 in the revised manuscript) | Detailed network framework of CELLECT. The overall framework of the CELLECT model is illustrated; The upper-left section depicts the main network structure, a 3D U-Net, with data dimensions and arrows representing the data flow and connection between modules. The color of the arrows corresponds to the network modules shown in the at the left bottom panel. The main model generates 3 outputs: cell division predictions, a confidence map, and a 64-dimensional embedding map, all maintaining same spatial resolution as the input, derived from 2 frames of input images. The confidence map serves as the basis for cell segmentation and size estimation. The lower-right section introduces the Center Enhancement Network (CEN), a light-weight 3D U-Net model. This network takes the confidence map and the 2 original image frames as inputs, refining the probability distribution by focusing on high-probability regions around cell center points while suppressing probabilities in other areas. The peaks in the confidence map generated by the CEN correspond directly to precise center point locations. This framework enables high-accuracy cell division prediction, segmentation, and size estimation while significantly improving the precision of center point localization. The network outputs shown here were generated from the Cell Tracking Challenge (CTC) Fluo-N3DH-CE dataset. The model used was trained on sequence #3 of the mskcc-confocal dataset.

Comment 12-18:

- * Supp Fig. 1 embedding -> embedding
- * Supp Fig. 1 division -> division
- * Legend Supp Fig. 1 The color of the arrows correspond -> The color of the arrows corresponds
- * Add one sentence to the interpretation of marker patterns in the right membrane-labeled cell in Supp Fig. 2 g & i - to gain evidence that this is really one cell.
- * Supp Fig. 5 - The three (as shown in the right panel?) or more (?) inputs of inter- and intra-frame MLP remain unclear. I would recommend to add a supplementary table containing all dimensions of input and output channels per neural network.
- * Supp Fig 6: Is the right figure correct? I did not had expected a dotted link from blue to gray and a correct link from blue to blue, this looks for me like a division event.
- * Legend Supp Fig. 8 with a green box in the each original image -> with a green box in each original image

Response:

We thank the reviewer for the thorough and helpful set of comments. We have carefully addressed all points as follows:

- In **Fig. R8 (Supplementary Fig. 1** in the revised manuscript), we corrected the spelling of “embedding” to “embedding” and “division” to “division.” The legend sentence was also revised for grammar: “The color of the arrows corresponds to the information type.”

- In **Supplementary Fig. 2g & i**, we added a sentence in the caption to clarify that the membrane-labeled structure in the right panel corresponds to a single cell. This interpretation is supported by the continuous and closed membrane signal without interruption, providing visual evidence that it is not a juxtaposition of multiple cells.
- In **Fig. R3 (Supplementary Fig. 5** in the revised manuscript), to clarify the architecture of the intra- and inter-frame MLPs, we added **Table R4 (Supplementary Table 3** in the revised manuscript), which summarizes all input and output tensor dimensions for each module in the pipeline, including embeddings, spatial features, and division indicators.
- In **Fig. R9 (Supplementary Fig. 6** in the revised manuscript), the fourth panel (“False positive division”) originally included a real division in the ground truth node and an additional dashed link from the model prediction, which visually resembled a 3-way division. We have revised the figure to focus on the erroneous dashed link while preserving the ground truth structure. The caption has been updated to clarify that the model prediction is incorrect, and the ground truth division remains valid.

Figure R9 (Supplementary Figure 6 in the revised manuscript) | **Illustration of different types of errors during cell tracking used for evaluation.** False positive (FP), false negative (FN), identity switch (IS), false positive division (FP-D) and false negative division (FN-D). Red and blue circles indicate ground truth tracks and reconstructed tracks, respectively. Blue lines represent edges correctly matched between the ground truth and reconstructed track, while dashed lines highlight redundant error tracks.

- In **Supplementary Fig. 8** in the revised manuscript, we corrected a grammatical error in the legend, which now reads: “with a green box in each original image.” All relevant figures, captions, and supplementary sections have been updated accordingly.

Comment 19:

* Dataset numbers #1 and #2 only occur in supplement and are not used throughout the paper.

Response:

We thank the reviewer for the comment. Datasets #1 and #2 were used in the cross-validation experiment presented in **Fig. R6 (Fig. 2 in the revised manuscript)**. These dataset labels are used consistently in **Tables R5-R6 (Supplementary Tables 1-2 in the revised manuscript)** to provide detailed performance descriptions. We have clarified this in the **Methods** section under the *C. elegans embryo public dataset* subsection and in the **Fig. R6 (Fig. 2 in the revised manuscript)** caption.

	FP	FN	IS	FP-D	FN-D	Division	Sum
Mskcc-confocal 270 frames							
StarryNite*	7.9	13	0.62	0.58	1.2	1.8	24
Linajea*	3.6	5.5	0.062	0.89	0.26	1.2	10.3
Linajea+csc+sSVM*	3.7	5.6	0.046	0.053	0.4	0.46	9.6
CELLECT	1.26	3.56	0.013	0.076	0.41	0.49	5.32
Nih-ls 270 frames							
StarryNite*	22	18	2.4	0.66	1.6	2.2	45
Linajea*	12	6.5	0.46	1.5	0.4	1.86	21
Linajea+csc+sSVM*	13	5.3	0.59	0.20	0.49	0.69	20
CELLECT	6.37	3.68	0.070	0.777	1.14	1.92	12.04

Table R5 (Supplementary Table 1 in the revised manuscript) | Quantitative results on mskcc-confocal and nih-ls dataset. For descriptions of the error metrics, see **Supplementary Fig. 6**. All values represent absolute errors normalized per 1,000 ground-truth (GT) edges; best value bold. * The values are obtained from the previous report (Hirsch, P. *et al. Medical Image Computing and Computer Assisted Intervention 2022*, 25–35 (2022)).

Train-id	Test-id	FP	FN	IS	FP-D	FN-D	Division	Sum
Mskcc-confocal 270 frames								
1	2	1.38	2.13	0.039	0.099	0.48	0.58	4.13
	3	0.9	3.62	0.039	0.039	0.26	0.30	4.86
2	1	1.32	5.33	0.0	0.097	0.50	0.60	7.25
	3	1.89	5.20	0.0	0.14	0.33	0.47	7.56
3	1	1.12	2.75	0.0	0.058	0.39	0.40	4.32
	2	0.95	2.34	0.0	0.02	0.50	0.52	3.81
Mean		1.26	3.56	0.013	0.076	0.41	0.49	5.32
Nih-ls 270 frames								
1	2	4.71	4.36	0.018	0.712	1.57	2.28	11.37
	3	6.14	6.38	0.049	0.5	0.77	1.27	13.84

2	1	2.44	5.21	0.064	0.614	1.67	2.28	9.99
	3	15.41	1.37	0.129	1.565	0.364	1.93	18.84
3	1	3.43	3.71	0.085	0.678	1.01	1.69	8.91
	2	6.09	1.05	0.074	0.591	1.46	2.05	9.27
Mean		6.37	3.68	0.070	0.777	1.14	1.92	12.04

Table R6 (Supplementary Table 2 in the revised manuscript) | Quantitative results for each fold in the cross-validation experiments on mskcc-confocal and nih-ls data. For descriptions of the error metrics, see **Supplementary Fig. 6**. All values represent absolute errors normalized per 1,000 ground-truth (GT) edges.

Comment D:

Appropriate use of statistics and treatment of uncertainties

Yes. Uncertainties are explicitly handled for segmentation masks in form of the confidence map and with retraining results in Supp Fig. 11. However, confidence maps for linking cells between two subsequent time points are missing.

Response:

We thank the reviewer for the comment. We agree that uncertainty in the temporal linking step is an important consideration for robust cell tracking. In response, we conducted an additional analysis to assess the stability and reproducibility of CELLECT's linking confidence across different temporal intervals.

Specifically, we selected 3 representative target cells from the mskcc-confocal dataset #2 and analyzed their 5 nearest spatial neighbors at frames $t+1$, $t+2$, and $t+3$. Using 5 independently trained CELLECT models with identical configurations, we evaluated the predicted similarity scores assigned to these candidates. As shown in **Fig. R10 (Supplementary Fig. 14 in the revised manuscript)**, the correct candidate consistently received the highest score across multiple models, even under extended frame gaps. While performance remained stable for adjacent-frame linking, we observed a moderate decline in accuracy at $t+3$, especially when mitotic events occurred. This indicates that CELLECT's linking module is capable of estimating reliable confidence even under temporally sparse or ambiguous conditions, but may be challenged for data with complex division scenarios and low imaging speeds.

This reproducibility experiment complements the retraining analysis presented in revised manuscript **Fig. R11 (Supplementary Fig. 13 in the revised manuscript)**, which focused on segmentation confidence. Together, they demonstrate that both segmentation and linking modules produce consistent results across independent runs, while also providing insight into where performance variability arises.

We have added this analysis and its interpretation to the **Methods** section under the subsection *Temporal linking with larger intervals*.

Figure R10 (Supplementary Figure 14 in the revised manuscript) | Evaluation of the reproducibility of cell linking during different imaging frame rate. To evaluate the reproducibility of CELLECT’s cell linking under different imaging frame rate, we selected 3 representative target cells (Target #1–3) from mskcc-confocal dataset #2 and selected 5 nearest spatial neighbors at frames $t+1$, $t+2$, and $t+3$ for testing. **a**, Location of the target cells at time t (frame 1) and the locations of candidate cells across the 3 subsequent frames. **b**, Similarity scores predicted by 5 independently trained CELLECT models for the 5 candidates. To ensure consistent input across models, redundant point removal was not applied intentionally, resulting in occasional duplicate redundant cell points (e.g., Target #1 and #2 at frame 1 linked to frame 2). This configuration increases linking difficulty by expanding the temporal gap, thereby simulating more complex multi-frame association scenarios. While correct links often correspond to the nearest candidate in adjacent-frame conditions, longer intervals such as frame 1 to frame 3 or frame 4 require the model to rely more on embedding features for

accurate decisions. The results demonstrate that CELLECT produces consistent linking performance across different frame intervals. However, under low imaging frame rate and in the presence of cell divisions (e.g., frame 1 linked to frame 4), a slight drop in accuracy is observed. All 5 models were independently trained on mskcc-confocal dataset #3. Scale bar: 10 μm

Figure R11 (Supplementary Figure 13 in the revised manuscript) | Evaluation of the reproducibility of the model. To assess the reproducibility of the model performance, we trained the model 5 times using identical parameters but with non-deterministic random sampling during training. The results revealed that, despite minor fluctuations, the model's

overall performance remains highly consistent and reproducible. Error connections are highlighted in pink, while correct connections are depicted in grey. All evaluation data in this figure were drawn from mskcc-confocal dataset #2. All models were trained using mskcc-confocal dataset #3.

Comment E: Conclusions: robustness, validity, reliability

see suggested improvements

Response: We thank the reviewer for the suggestions and we have revised accordingly.

Comment F: Suggested improvements: experiments, data for possible revision

see suggested improvements

Response: We thank the reviewer for the suggestions and we have revised accordingly.

Comment G: References: appropriate credit to previous work?

Yes, mostly. But please compare your approach with other contrastive-learning approaches for cell tracking, see e.g. Zyss, D., Sharma, A., Ribeiro, S. A., Repellin, C. E., Lai, O., Ludlam, M. J., ... & Fehri, A. (2024). Contrastive learning for cell division detection and tracking in live cell imaging data. bioRxiv, 2024-08.

Response:

We thank the reviewer for highlighting the recent work by Zyss et al. (2024), which also applies contrastive learning to cell tracking. We have cited these papers and added comparisons in the revised version. While both Neven et al. and Zyss et al.'s approaches leverage contrastive learning, their approaches and our approaches differ significantly in terms of training objectives, annotation requirements, and application scenarios.

Zyss et al. rely on fully segmented cell masks to learn dense, instance-level features and focus on reconstructing cell lineages over long timescales. Their method is particularly well-suited for low temporal resolution datasets, where manual trajectories are unavailable and division events must be inferred from cell shape and proximity. In contrast, our method is tailored for 3D microscopy data, where full segmentation is impractical and annotations are limited to sparse center points. Our contrastive learning objective is designed to promote local feature separability between nearby cells, enabling efficient inter-frame association in dynamic, high-resolution time-lapse sequences.

In summary, while Zyss et al. aim to recover lineage structure from dense appearance features, our focus is on tracking individual cell identities under sparse supervision in 3D. We view the 2 approaches as complementary—each addressing distinct challenges in cell tracking depending on annotation availability and imaging modality.

We have added a dedicated subsection titled *Comparison with embedding-based and contrastive learning methods* in the revised **Methods** section, providing a conceptual and

practical comparison between CELLECT and representative prior works such as Neven et al. and Zyss et al.

Comment H: Clarity and context: lucidity of abstract/summary, appropriateness of abstract, introduction and conclusions

No problems.

Response: We thank the reviewer for confirming the clarity and appropriateness of the abstract, introduction, and conclusions.

Reviewer #2

Summary:

Zhou and colleagues developed a novel framework, based on AI, for efficient cell tracking in 3D large samples, that they describe in this article. The framework, called CELLECT, relies on a deep-learning architecture based on a U-Net architecture that operates with two consecutive frames as input for inference. The U-Net outputs the probability for a voxel to be a cell center, and a final output for the cell size. A second output is the probability map of cell division. The confidence map of the cell center is generated by summing the voxel-wise probabilities. To detect cell center disappearance, the authors integrate an estimate of the cell radius and their segmentation. The sparse cell centers obtained in these steps are used to read the 5d-channel feature map, which are then used in the tracking step. In the linking step, for each cell center, the 5 nearest neighbors are sought in the same frame and in the next frame. The nearest neighbors in the same frame are used to remove redundant cell centers, acting in effect as a non-maxima suppression. The nearest neighbors in the next frame are used for linking, detecting cell disappearance, cell division or cell movement. In the training phase, the model was trained with contrastive learning to generate an embedding with features that maximizes the difference between different cells while minimizing the difference between same cells in different time-points.

The authors trained a first model on the data available in the Cell Tracking Challenge. They then run a benchmark for CELLECT, against datasets taken from other publications. The benchmark shows that CELLECT supersedes Linajea, Imaris and StarryNite, both in error rate and in computation time. Ablation studies revealed the importance of the CEN step for accuracy and of patch size for performance.

CELLECT is then used in 3 applications to demonstrate its effectiveness for biological imaging. First it is used in mouse germinal center images to track dense T cells and B cells. The authors actually used sparse labeling to generate a ground truth on which to measure the tracking performance. This application demonstrates the ability of CELLECT to perform well (superseding by memory and energy consumption) on a large dataset with

excellent accuracy, compared to Imaris and Linajea. The second application follows the phagocytosis of bacteria by immune cells. For this, the author used the same pretrained model described above, and noted that obtaining segmentation and tracking does not require adjusting parameters, as for non-AI approaches like Imaris. In the last application, the authors track neurons in a deforming tissue, and use the tracks to obtain robust calcium recording for single cells over a long time. This application exemplifies the ability of CELLECT to yield long, uninterrupted tracks, compared e.g. to TrackMate.

This article clearly describes an important work, in particular because:

- The tracking requires only sparse annotation of cell centers, which considerably lowers the annotation time for the training phase.
- The linking step operates with only the features measured at the few cell centers, which significantly lowers the computational cost for the inference phase.

These two achievements, concomitant with good accuracy, make this work very promising to become widely used in the biological imaging field, and make the analysis of difficult images (long movies, large images, dense cells) feasible by a wide range of labs with standard computing equipment. Also, we can welcome an AI tracking framework that offers a lower energy consumption, even compared to Imaris! And the promises of training on sparse annotations make it realistic to have even pretrained models for CELLECT widely available. There are however several important points missing in this work that should be addressed to fulfill these promises.

Response:

We greatly appreciate the reviewer's detailed illustration and positive assessment of CELLECT, especially regarding its efficiency, accuracy, and potential for widespread application in biological imaging. The review accurately highlights the key innovations of CELLECT, particularly the use of sparse annotations for cell tracking, the high accuracy with good generalization, and the computational efficiency in the linking step. We are pleased to know that the reviewer finds the energy efficiency and the potential for pretrained models to be noteworthy. In the following sections, we address the specific concerns raised by the reviewer to further enhance the manuscript and clarify key aspects of our work.

- We revised the installation instructions, resolved key dependency issues, and verified successful execution across macOS, Ubuntu, and Windows. We also added a Jupyter Notebook demo for reproducibility and fixed the broken data link in the README.
- We added details of CELLECT's official submission to the Cell Tracking Challenge (<https://celltrackingchallenge.net/>), where it ranked first in cell tracking and segmentation for the Fluo-N3DH-CE dataset.
- We clarified that tracking is essential for capturing immune cell-bacteria interactions, as shown in **Fig. R16 (revised Fig. 4)**. We emphasized that the tracking and segmentation are both essential to be conducted when we needed to

quantify the number of cell-bacteria interactions for each specific cell across a long term. In this case, this application can well demonstrate the advantage of CELLECT to achieve high-accuracy segmentation and tracking simultaneously in the same framework.

- We modified the codebase to support both GPU (CUDA) and CPU execution modes, allowing seamless operation on various devices by auto-detecting the hardware environment.
- We clarified that CELLECT uses spatial distance for candidate selection, balancing coverage and efficiency with adjustable settings for different scenarios.
- We improved the clarity of **Figs. R16 (revised Fig. 4) and R21 (revised Fig. 3)** by adding enlarged insets and **Supplementary Figs. 16-17**, which compare CELLECT and Imaris, highlighting where CELLECT succeeds in challenging regions.
- We clarified that CELLECT estimates coarse cell regions using sparse cell-center annotations and contrastive learning, not precise cell contours. We added a subsection in **Methods** to explain this.
- We explained that CELLECT was trained solely on mskcc-confocal and nih-ls datasets using a 3-fold cross-validation protocol. We clarified that the CTC leaderboard results were used only to demonstrate generalizability and were not part of training or evaluation.
- We added specific tool versions, parameters, and citations for Imaris, TrackMate, and StarryNite in the revised **Methods** section.
- We expanded the Discussion to cover limitations, including:
 - Tracking Discontinuity: We added a post-hoc linking strategy to handle missing cells.
 - Temporal Optimizations: We discuss the potential to integrate trajectory priors and motion predictors in the future work.
 - Data Processing Size: We clarified that the large-scale preprocessing files (hundreds of GB) are solely introduced to accelerate data loading during training. These files are not essential and are not part of the CELLECT algorithm itself. As shown in **Fig. R12**, the tracking algorithm can be executed directly without this preprocessing step.
 - Tracking Errors: We identified common errors and directions for future improvement.
- We clarified that CELLECT is distinct in its focus on cell tracking in microscopic imaging. We will define the acronym more prominently in the manuscript and make adjustments in the documentation to reduce confusion. We remain open to a name change if strongly recommended.

Major point:

Comment 1:

Could not install the code & lacking documentation

We could not get the code to install, following the procedure listed on the linked repository (<https://github.com/zzz333za/CELLECT>). Here are the instructions we have tried and the error we got: On a Apple Mac M2, Windows 11 and Ubuntu 18.04:

```
>> git clone git@github.com:zzz333za/CELLECT.git
>> cd CELLECT
>> conda create -n collect python=3.11.7
>> conda activate collect
>> pip install -r requirements.txt
```

The error:

```
Collecting backcall==0.2.0 (from -r requirements.txt (line 1))
  Downloading backcall-0.2.0-py2.py3-none-any.whl.metadata (2.0 kB)
Collecting cyclер==0.10.0 (from -r requirements.txt (line 2))
  Downloading cyclер-0.10.0-py2.py3-none-any.whl.metadata (722 bytes)
ERROR: Could not find a version that satisfies the requirement dataclasses==0.8 (from versions: 0.1, 0.2, 0.3, 0.4, 0.5, 0.6)
```

After removal of the dataclasses requirement (included in the standard library since Python 3.7), we get the following error on imagecodecs:

```
Collecting imagecodecs==2020.5.30 (from -r requirements.txt (line 5))
  Downloading imagecodecs-2020.5.30.tar.gz (9.0 MB)
  9.0/9.0 MB 90.8 MB/s eta 0:00:00
  Preparing metadata (setup.py) ... error
  error: subprocess-exited-with-error

  × python setup.py egg_info did not run successfully.
  exit code: 1
  ↳ [7 lines of output]
  Traceback (most recent call last):
    File "<string>", line 2, in <module>
    File "<pip-setuptools-caller>", line 34, in <module>
    File "/tmp/pip-install-vkhw1zs9/imagecodecs_104b55ae0ab4407a9072c6ad0bb0775d/setup.py", line 320, in <module>
      import Cython # noqa
      ^^^^^^^^^^^^^
  ModuleNotFoundError: No module named 'Cython'
  [end of output]

  note: This error originates from a subprocess, and is likely not a problem with pip.
  error: metadata-generation-failed

  × Encountered error while generating package metadata.
  ↳ See above for output.

  note: This is an issue with the package mentioned above, not pip.
  hint: See above for details.
```

Installing Cython does not solve the issue (Cython compilation error in imagecodecs). We would really need to install and run the code to perform the review in full.

Additionally we think that the readership of the article would greatly benefit from detailed installation instructions that would be widely applicable and reproducible. In the same line, adding one or two examples in the shape of Jupyter notebooks for the inference and training phase would greatly improve the usability of the code and its adoption by the scientific community. As example, we suggest adding data and notebooks to reproduce the analysis of one of the use-cases in the article. Small issue but important nonetheless: the link to the data in the README is incorrect (<https://github.com/zzz333za/CELLECT/blob/main/zenodo.org/record/6460303>). We could still access the data by taking the zenodo part of the link.

Response:

We sincerely apologize for the installation issues and highly appreciate the reviewer's detailed feedback. We have carefully revised the installation instructions, resolved package dependency order issues (especially between numpy and scikit-image), and verified the updated setup across multiple environments, including **macOS** (version 15.1), **Ubuntu 20.04**, and **Windows 11**. The current simplified setup is now fully functional. We also removed the dependency on fixed package versions (including PyTorch and tiffle) to reduce system-specific package requirements. This change improves compatibility with macOS (which has limited support for alternative PyTorch builds), while requiring Windows users to manually install the GPU version of PyTorch if needed.:

- **Install:**

```
cd CELLECT/  
conda create -n collect python=3.11.7 numpy scipy scikit-image -y  
conda activate collect  
pip install -r requirements.txt  
# (Only required on Windows for GPU support)  
pip install torch torchvision torchaudio --index-url  
https://download.pytorch.org/whl/cu126
```

Once installed, the full workflow can now be executed using the following steps:

- **Model evaluation:**

```
python s-test.py --data_dir ../extradata/mskcc-confocal --out_dir ./ \  
--model1_dir ./model/U-ext+-x3rdstr0-149.0-3.4599.pth \  
--model2_dir ./model/EX+-x3rdstr0-149.0-3.4599.pth \  
--model3_dir ./model/EN+-x3rdstr0-149.0-3.4599.pth
```

- **Preprocessing the dataset:**

(Optional) Data preprocessing – only needed when training your own model, to speed up loading

```
python con-label-input.py --data_dir ../extradata/mskcc-confocal --out_dir ./ --num 2
```

- **Model training:**

```
python s-train.py --data_dir ../extradata/mskcc-confocal --out_dir ./ \  
--train 2 --val 2 --model_dir ./model/
```

Figure R12 | Cross-platform installation and training verification of CELLECT. We validated the CELLECT installation and execution process across 3 major platforms: macOS 15.1, Ubuntu 20.04, and Windows 11. All systems successfully completed installation and inference using default commands. On GPU-supported Windows and Ubuntu systems, additional training and preprocessing routines were executed and confirmed to run smoothly. Notably, on Linux, PyTorch's data loader acceleration using multiple workers provided faster training performance.

We have also added a **Jupyter Notebook demo (Fig. R13)** to the repository to support interactive exploration and improve usability for new users. The notebook walks through the full pipeline, including cell center extraction, feature visualization, and candidate cell scoring for association across frames.

<https://nbviewer.org/github/zzz333za/CELLECT/blob/main/example-CELLECT.ipynb>

Figure R13 | Screenshot of Jupyter Notebook output.

The figure shows the maximum intensity projection (MIP) of the raw image and the visualization of the detected center points. In addition, the broken Zenodo download link in the README has been corrected and verified. All installation steps and data download instructions are now updated and reproducible. If there are any further issues with platform compatibility or runtime behavior, we would be happy to assist. We thank the reviewer again for helping us improve the usability and accessibility of the CELLECT codebase and are very sorry about the previous implementation issues.

Comment 2:

The Cell-Tracking-Challenge metrics and benchmark

In the lines 177-178 the authors claim that they secured the top ranking in the Cell Tracking Challenge (CTC), but the data in the Supplemental Tables 1 and 2 do not correspond to the whole CTC metrics, even if they show better accuracy than contenders. We lack a table that shows how CELLECT fares against the CTC benchmarks. It is understood that CELLECT might not achieve top accuracy for all CTC datasets, but this table would help future work in positioning CELLECT within the tracking field among other tracking algorithms. In parallel, CELLECT could be submitted to the CTC as a participant.

Response:

We thank the reviewer for the constructive suggestion. We agree that official submission to the Cell Tracking Challenge (CTC) is essential for assessing generalizability on an unseen test dataset. We have officially submitted CELLECT to the CTC website under the team name "THU-CN (3)", which was automatically generated by the platform to indicate the third submission from our affiliation (Tsinghua University, China). Please note that superscript (3) in the scores of THU-CN means that there are 2 earlier submissions under the same institution which were submitted several years ago by other research teams in Tsinghua University not our group. Our results became available on October 20, 2024, and are now visible on the CTC leaderboard, ranking 1st for both tasks of cell tracking and segmentation in the Fluo-N3DH-CE dataset in terms of the overall performance measure (OP_{CSB} and OP_{CTB}). In the table of the website, only methods that have consistently ranked within the top 3 over the years are included in the rankings by the organizers of the challenge. We are still awaiting the update of our method description on the CTC website, which is not controlled by us and only accessible by the team of CTC. They told us by email that it would be updated soon. It is worth noting that both our code and documentation have been validated by the CTC organizers, which was a prerequisite for leaderboard publication. The scores achieved by our method on this dataset have been included in the revised manuscript. The corresponding rankings are available on the official website (celltrackingchallenge.net) and are visualized in the figure below, with our team highlighted for clarity.

Cell Segmentation Benchmark

	Generalizability	Bf-C2DL-MSC*	Bf-C2DL-MuSC*	DK-C2DH-Hela*	Fluo-C2DL-Huh7	Fluo-C2DL-MSC*	Fluo-C3DH-A549*	Fluo-C3DH-H157*	Fluo-C3DL-MDA231*	Fluo-N2DH-GOWT1*	Fluo-N2DL-Hela*	Fluo-N3DH-CE*	Fluo-N3DH-CHO*	Fluo-N3DL-DRO	Fluo-N3DL-TRIC	PhC-C2DH-TRIF	PhC-C2DL-U573*	Fluo-C3DH-A549-SIM	Fluo-N2DH-SIM*	Fluo-N3DH-SIM*	Rank
OP _{CSB}	0.849 ⁽³⁾ 0.842 ⁽³⁾ 0.799 ⁽³⁾	0.926 ⁽³⁾ 0.910 ⁽³⁾ 0.908 ⁽³⁾	0.884 ⁽³⁾ 0.883 ⁽³⁾ 0.876 ⁽³⁾	0.926 ⁽³⁾ 0.912 ⁽³⁾ 0.912 ⁽³⁾	0.885 ⁽³⁾ 0.879 ⁽³⁾ 0.859 ⁽³⁾	0.761 ⁽³⁾ 0.760 ⁽³⁾ 0.746 ⁽³⁾	0.954 0.952 0.938 ⁽³⁾	0.938 ⁽³⁾ 0.930 ⁽³⁾ 0.927 ⁽³⁾	0.810 ⁽³⁾ 0.807 ⁽³⁾ 0.771 ⁽³⁾	0.952 ⁽³⁾ 0.948 0.948	0.957 ⁽³⁾ 0.853 ⁽³⁾ 0.842 ⁽³⁾	0.926 ⁽³⁾ 0.873 0.863 ⁽³⁾	0.873 0.891 ⁽³⁾ 0.913	0.906 ⁽³⁾ 0.895 ⁽³⁾ 0.885	0.895 ⁽³⁾ 0.961 ⁽³⁾ 0.956 ⁽³⁾	0.859 ⁽³⁾ 0.859 ⁽³⁾ 0.857 ⁽³⁾	0.977 0.975 ⁽³⁾ 0.943 ⁽³⁾	0.905 ⁽³⁾ 0.905 0.905 ⁽³⁾	0.949 0.899 ⁽³⁾ 0.866 ⁽³⁾	1 st 2 nd 3 rd	
SEG	0.791 ⁽³⁾ 0.761 ⁽³⁾ 0.731 ⁽³⁾	0.855 ⁽³⁾ 0.828 ⁽³⁾ 0.826 ⁽³⁾	0.784 ⁽³⁾ 0.782 ⁽³⁾ 0.774 ⁽³⁾	0.877 ⁽³⁾ 0.873 0.871 ⁽³⁾	0.811 ⁽³⁾ 0.791 ⁽³⁾ 0.782	0.687 0.657 0.655 ⁽³⁾	0.908 0.903 0.888 ⁽³⁾	0.908 0.890 ⁽³⁾ 0.888 ⁽³⁾	0.710 ⁽³⁾ 0.710 ⁽³⁾ 0.704 ⁽³⁾	0.938 0.935 ⁽³⁾ 0.933	0.923 ⁽³⁾ 0.729 ⁽³⁾ 0.922 ⁽³⁾	0.759 ⁽³⁾ 0.917 ⁽³⁾ 0.914	0.925 ⁽³⁾ 0.760 0.738 ⁽³⁾	0.760 0.821 ⁽³⁾ 0.713 ⁽³⁾	0.793 ⁽³⁾ 0.776 ⁽³⁾ 0.782	0.931 ⁽³⁾ 0.929 ⁽³⁾ 0.927	0.756 ⁽³⁾ 0.743 ⁽³⁾ 0.741 ⁽³⁾	0.955 0.951 ⁽³⁾ 0.885 ⁽³⁾	0.832 0.830 ⁽³⁾ 0.830 ⁽³⁾	0.906 0.820 ⁽³⁾ 0.759 ⁽³⁾	1 st 2 nd 3 rd
DET	0.922 ⁽³⁾ 0.906 ⁽³⁾ 0.871 ⁽³⁾	0.997 ⁽³⁾ 0.995 0.994 ⁽³⁾	0.984 ⁽³⁾ 0.983 0.982 ⁽³⁾	0.975 ⁽³⁾ 0.961 ⁽³⁾ 0.960	0.968 ⁽³⁾ 0.959 ⁽³⁾ 0.956 ⁽³⁾	0.876 ⁽³⁾ 0.851 ⁽³⁾ 0.832	1.000 1.000 1.000	0.988 ⁽³⁾ 0.982 0.978	0.911 0.907 ⁽³⁾ 0.904 ⁽³⁾	0.980 0.976 ⁽³⁾ 0.970	0.994 0.992 0.992 ⁽³⁾	0.995 0.990 ⁽³⁾ 0.982 ⁽³⁾	0.954 ⁽³⁾ 0.945 ⁽³⁾ 0.934	0.988 ⁽³⁾ 0.985 0.970	0.994 ⁽³⁾ 0.994 ⁽³⁾ 0.986	0.997 ⁽³⁾ 0.997 ⁽³⁾ 0.994	0.990 ⁽³⁾ 0.990 ⁽³⁾ 0.988 ⁽³⁾	0.975 ⁽³⁾ 0.975 ⁽³⁾ 0.973 ⁽³⁾	1.000 1.000 1.000	0.981 ⁽³⁾ 0.978 ⁽³⁾ 0.974 ⁽³⁾	0.992 2 nd 3 rd

AC

FR-GE

MU-US

UFRGS-BR

AMOLF-NL

HD-GE (BMCV)

ND-US

UNSW-AU

BFR-GE

KIT-GE

NUDT-CN

WARW-UK

BGU-IL

KTH-SE

OX-UK

CALT-US

LEID-NL

PURD-US

CSU-CN

MON-AU

RWTH-GE

CUNI-CZ

MPI-GE (CBG)

THU-CN

TUG-AT

CZB-US

MSU-RU

DKFZ-GE

MU-CZ

UCH-CL

Created on 2024-12-18. www.celltrackingchallenge.net

Legend: * (dataset included in or result transferred from the generalizability study)

Cell Tracking Benchmark

	Generalizability	Bf-C2DL-MSC*	Bf-C2DL-MuSC*	DK-C2DH-Hela*	Fluo-C2DL-Huh7	Fluo-C2DL-MSC*	Fluo-C3DH-A549*	Fluo-C3DH-H157*	Fluo-C3DL-MDA231*	Fluo-N2DH-GOWT1*	Fluo-N2DL-Hela*	Fluo-N3DH-CE*	Fluo-N3DH-CHO*	Fluo-N3DL-DRO	Fluo-N3DL-TRIC	PhC-C2DH-TRIF	PhC-C2DL-U573*	Fluo-C3DH-A549-SIM	Fluo-N2DH-SIM*	Fluo-N3DH-SIM*	Rank	
OP _{CTB}	0.791 ⁽³⁾ 0.732 ⁽³⁾ 0.637 ⁽³⁾	0.908 ⁽³⁾ 0.906 ⁽³⁾ 0.901 ⁽³⁾	0.880 ⁽³⁾ 0.878 ⁽³⁾ 0.870 ⁽³⁾	0.912 0.909 ⁽³⁾ 0.904	0.875 ⁽³⁾ 0.843 ⁽³⁾ 0.772	0.759 ⁽³⁾ 0.740 ⁽³⁾ 0.710 ⁽³⁾	0.938 ⁽³⁾ 0.931 ⁽³⁾ 0.925 ⁽³⁾	0.938 ⁽³⁾ 0.929 ⁽³⁾ 0.925 ⁽³⁾	0.797 ⁽³⁾ 0.761 0.757 ⁽³⁾	0.951 ⁽³⁾ 0.945 ⁽³⁾ 0.953 ⁽³⁾	0.956 ⁽³⁾ 0.844 0.829 ⁽³⁾	0.850 ⁽³⁾ 0.912 ⁽³⁾ 0.911	0.926 ⁽³⁾ 0.617 ⁽³⁾ 0.591	0.708 0.816 ⁽³⁾ 0.787 ⁽³⁾	0.867 ⁽³⁾ 0.804 ⁽³⁾ 0.785 ⁽³⁾	0.841 0.954 ⁽³⁾ 0.951 ⁽³⁾	0.855 ⁽³⁾ 0.854 ⁽³⁾ 0.854 ⁽³⁾	0.920 ⁽³⁾ 0.913 0.865 ⁽³⁾	0.905 ⁽³⁾ 0.904 ⁽³⁾ 0.896	0.897 ⁽³⁾ 0.865 ⁽³⁾ 0.848 ⁽³⁾	1 st 2 nd 3 rd	
SEG	0.728 ⁽³⁾ 0.649 ⁽³⁾ 0.530 ⁽³⁾	0.828 ⁽³⁾ 0.826 ⁽³⁾ 0.818 ⁽³⁾	0.784 ⁽³⁾ 0.782 ⁽³⁾ 0.774 ⁽³⁾	0.873 0.863 ⁽³⁾ 0.853 ⁽³⁾	0.791 ⁽³⁾ 0.751 ⁽³⁾ 0.690	0.657 0.655 ⁽³⁾ 0.650	0.876 ⁽³⁾ 0.863 ⁽³⁾ 0.849	0.889 ⁽³⁾ 0.888 ⁽³⁾ 0.878 ⁽³⁾	0.710 ⁽³⁾ 0.642 0.632 ⁽³⁾	0.935 ⁽³⁾ 0.931 0.929	0.923 ⁽³⁾ 0.922 0.919 ⁽³⁾	0.759 ⁽³⁾ 0.728 ⁽³⁾ 0.725 ⁽³⁾	0.917 ⁽³⁾ 0.905 ⁽³⁾ 0.902	0.613 0.567 ⁽³⁾ 0.397	0.791 ⁽³⁾ 0.766 0.680 ⁽³⁾	0.746 0.684 0.654 ⁽³⁾	0.924 ⁽³⁾ 0.923 ⁽³⁾ 0.922 ⁽³⁾	0.743 0.741 ⁽³⁾ 0.740 ⁽³⁾	0.840 ⁽³⁾ 0.827 0.730 ⁽³⁾	0.830 ⁽³⁾ 0.830 ⁽³⁾ 0.825 ⁽³⁾	0.820 ⁽³⁾ 0.759 ⁽³⁾ 0.746 ⁽³⁾	1 st 2 nd 3 rd
TRA	0.855 ⁽³⁾ 0.815 ⁽³⁾ 0.768 ⁽³⁾	0.989 ⁽³⁾ 0.987 ⁽³⁾ 0.985 ⁽³⁾	0.976 ⁽³⁾ 0.974 ⁽³⁾ 0.971 ⁽³⁾	0.955 0.954 ⁽³⁾ 0.954	0.960 ⁽³⁾ 0.934 ⁽³⁾ 0.865 ⁽³⁾	0.873 ⁽³⁾ 0.839 ⁽³⁾ 0.788	1.000 1.000 1.000	0.987 ⁽³⁾ 0.980 ⁽³⁾ 0.976	0.884 ⁽³⁾ 0.882 ⁽³⁾ 0.880	0.993 0.976 ⁽³⁾ 0.880	0.994 0.992 0.991 ⁽³⁾	0.994 0.987 ⁽³⁾ 0.979	0.953 ⁽³⁾ 0.955 ⁽³⁾ 0.935	0.802 0.952 ⁽³⁾ 0.785	0.952 ⁽³⁾ 0.936 0.854	0.985 ⁽³⁾ 0.982 ⁽³⁾ 0.886 ⁽³⁾	0.968 ⁽³⁾ 0.967 ⁽³⁾ 0.962 ⁽³⁾	1.000 1.000 1.000	0.968 ⁽³⁾ 0.978 ⁽³⁾ 0.967 ⁽³⁾	0.974 ⁽³⁾ 0.972 ⁽³⁾ 0.967	1 st 2 nd 3 rd	

AC

HIT-CN

ND-US

AMOLF-NL

IGFL-FR

NUDT-CN

BGU-IL

JAN-US

PURD-US

CAS-CN

KIT-GE

RWTH-GE

CUNI-CZ

KTH-SE

THU-CN

TUG-AT

CZB-US

LEID-NL

UVA-NL

DESU-US

MPI-GE (CBG)

DREX-US

MU-CZ

FR-GE

MU-US

Created on 2024-12-18. www.celltrackingchallenge.net

Legend: * (dataset included in or result transferred from the generalizability study)

<https://celltrackingchallenge.net/latest-csb-results/>
<https://celltrackingchallenge.net/latest-ctb-results/>

Figure R14 | Benchmarking of CELLECT in Cell Tracking Challenge (CTC). **a**, Results for the cell segmentation benchmark (CSB). **b**, Results for the cell tracking benchmark (CTB). We submit CELLECT (named as “THU-CN” based on our affiliation) for the benchmarking of CTC in the track of the Fluo-N3DH-CE dataset, which is the most challenging 3D time-lapse dataset with many cell divisions. Both the score and rank of CELLECT are shown in the 2 tables. Please note that superscript (3) of the score means that there are 2 earlier submissions under the same institution which were submitted several years ago by other research teams in Tsinghua University not our group. Only methods that have consistently ranked within the top 3 over the years are included in the rankings by the organizers of the challenge. CELLECT ranked 1st in terms of overall performance (OP) for both segmentation competition (OP_{CSB}) and tracking competition (OP_{CTB}). OP: Overall performance measure. SEG/TRA/DET are the sub-scores set by competition organizers. The competition test datasets are unannotated and unseen dataset for our model, and evaluations are conducted by third-party organizers without us involved to ensure reproducible results. We used purple color to depict the result of the CELLECT score ranking 1st.

Our official evaluation results on the CTC leaderboard for the Fluo-N3DH-CE dataset are shown below:

- <https://github.com/zzz333za/CELLECT>
- <https://github.com/zzz333za/CELLECT-ctc.ver>

The second repository contains a dedicated branch that reproduces the exact configuration and results submitted to the CTC leaderboard.

In **Fig. R15 (Fig. 2** in the revised manuscript), we benchmarked CELLECT using a 3D dataset described in the Linajea paper, which includes 2 datasets acquired by different imaging modalities (mskcc-confocal and nih-ls), each with 3 sub-volumes. According to the original publication, the first 270 frames of each sub-volume are annotated with manually verified center points. The paper also provides performance results of both the Linajea-series methods and StarryNite on this dataset, making it highly suitable for comparing tracking algorithms in similar settings.

Among the previously reported deep learning methods, only the Linajea series supports training based on sparse center annotations, making it directly comparable to our approach. In addition, we included Imaris, a widely used commercial tool, as a baseline. We also tested TrackMate, but its performance was significantly lower than Imaris in this setting, so it was not included in the main figure.

In summary, we have added corresponding results in the revised manuscript and a new subsection titled *Evaluation on Cell Tracking Challenge benchmark* in the **Methods** section.

Figure R15 (Figure 2) | Benchmarking of CELLECT with state-of-the-art algorithms in Cell Tracking Challenge. **a-d**, Performance evaluation on the dataset of mskcc-confocal. **e-h**, Performance evaluation on the dataset of nih-light sheet. **a** and **e**, The average number of errors per 1,000 ground-truth edges for each error type. The errors were categorized into 5 conditions: false-positive edges, false-negative edges, identity switches, false positive divisions, and false negative divisions. **b** and **f**, The proportion of error-free tracks (marked as tracking accuracy) versus a tracking duration in terms of frames. **c** and **g**, Comparisons of average computational time cost for each frame among different algorithms. **d** and **h**, The hierarchical circular diagram of lineage tracing with the grey lines indicating the cell division. The bold black lines and circles indicate the false negative identity switch and division, respectively. The scores for the 3 models other than CELLECT and the accuracy curves of Linajea are derived from the previous report³⁵. ‘Linajea + csc + sSVM’ refers to an enhanced version of Linajea that incorporates a cell state classifier (csc) to model biological states (e.g., division, continuation, polar body) and a structured Support Vector Machine (sSVM) to learn tracking cost functions from data. All evaluations were conducted using the

first 270 frames of each of the 3 sub-volumes for both datasets (mskcc-confocal and nih-light sheet), following a cross-validation protocol. The CELLECT models used in each comparison were trained on one of the 2 other sub-volumes. Further details on the training and evaluation splits are provided in **Supplementary Table 1** and **2**. Scale bars: 10 μm .

Comment 3:

The second application focuses on segmentation, not tracking

In the second use-case described in the article, the authors follow the phagocytosis of bacteria by immune cells, and in figure 4 they report the dynamics of the quantity of bacteria that are phagocytosed. But this use-case can work with only cell segmentation and bacteria detection, and does not require tracking. It is enough to segment immune cells at each frame, detect bacteria location, and for each bacteria classify it as phagocytosed if it colocalizes with a voxel of an immune cell. Putting this use case forward requires a slightly longer discussion on the segmentation accuracy of CELLECT. The CTC metrics mentioned above skip such metrics, that can be used to evaluate the tool performance for this task. The authors might consider that including this would dilute the focus of the article and shift the message away from cell tracking. In that case, another use case might be more appropriate.

Response:

We appreciate the reviewer's thoughtful and constructive feedback. While we agree that the phagocytosis itself does not involve highly complex cell tracking, we believe that the distinct cellular behaviors exhibited by bacteria, neutrophils and macrophages still provide a representative and meaningful demonstration of CELLECT's robustness and adaptability in challenging, real-world biological contexts. In addition, this experiment is designed to show the multi-function advantage of CELLECT that it can achieve both tracking and segmentation in the same framework without the requirement of 2 individual networks.

This dataset presents a highly dynamic case for 3D intravital environment in mammals, characterized by membrane-labeled neutrophils undergoing active migration, macrophages exhibiting complex interactions, and bacterial particles passively drifting within the tissue microenvironment. These components co-exist within the same 3D volume and collectively pose significant challenges for tracking. Despite the absence of dataset-specific training or manual annotations, CELLECT robustly associates bacterial trajectories with corresponding immune cells and captures phagocytosis events over time.

In this context, tracking was essential not only for localizing bacteria, but for associating each bacterium with the corresponding immune cell and monitoring whether it was fully internalized and disappeared in subsequent frames. In particular, tracking enabled us to determine the trajectories of phagocytosed bacteria and to identify the dynamic movements of neutrophils that actively chasing and engulf bacteria. Otherwise, some

bacteria may be washed away quickly in 1 frame in the blood vessel and then disappear. We have included more representative examples in **Fig. R16** (updated **Fig. 4** in the revised manuscript) i and j, which illustrate the trajectories of both neutrophils and bacteria, highlighting the dynamic interactions that lead to phagocytosis events.

To further highlight the role of tracking, we also included neutrophil motion trajectories and their temporal alignment with engulfed bacteria in the revised **Fig. R16** (**Fig. 4** in the revised manuscript). This joint visualization clearly demonstrates the relationship between bacterial fate and cell movement over time. While segmentation was used to assess whether a bacterium was located within a cell body, it primarily served as a supporting tool for determining cell-bacteria interaction and was not the sole purpose of this analysis.

Finally, we can further quantify the number of bacteria phagocytosed by each specific cell, which cannot be achieved without the tracking capability and segmentation capability simultaneously (**Fig. R17** (**Supplementary Fig. 20** in the revised manuscript)). We have added all the new results in the revised manuscript.

Figure R16 (Figure 4 in the revised manuscript) | Simultaneous tracking and segmentation of CELLECT facilitate accurate identification of immune interactions among bacteria, neutrophils, and macrophages. **a**, Experimental and processing schematics: Bacteria were injected into the surgically exposed spleen of a mouse, and activities of bacteria, neutrophils and macrophages were monitored by 2pSAM at 30 volumes per second over the subsequent 4 hours. The events of bacterial phagocytosis by 2 different immune cell types were identified by the tracking and segmentation of CELLECT. **b**, Example orthogonal MIP of 3-color imaged by 2pSAM in mouse spleen. **c**, Visualization of bacterial and neutrophil trajectories extracted by CELLECT. The combined display highlights dynamic immune–bacteria interactions. **d**, Segmentation results of neutrophils obtained by CELLECT and Imaris. **e**, Segmentation results of macrophages obtained by CELLECT and Imaris. The results of Imaris in **d** and **e** are sensitive to different intensity thresholds, so we adjusted manually for optimum accuracy and demonstrated the influence in **Supplementary Fig. 19**. The inverted triangles indicate distinct structures that were inaccurately segmented as neutrophils but resembled macrophages. **f**, Quantification of bacteria in different states: extracellular, intracellular within neutrophils, and within macrophages. **g**, Spatiotemporal distribution of neutrophil-phagocytosis events indicated by dots with different colors labeling the time points. **h**, Spatiotemporal distribution of macrophage-phagocytosis events. **i**, MIPs of bacteria engulfed by a neutrophil at different time points. **j**, Corresponding traces indicating the dynamic interaction process with the relative distance changes over time shown below. **k**, MIPs of bacteria engulfed by a macrophage at different time points. **l**, Corresponding traces indicating the dynamic interaction process with the relative distance changes over time shown below. All data were acquired by live 2pSAM imaging of surgically exposed mouse spleen over a 4-hour timespan. CELLECT results were obtained using the model trained on sequence #3 of the mskcc-confocal dataset. Scale bars: 50 μm (**b-h**) and 5 μm (**i-k**).

Figure R17 (Supplementary Fig. 20 in the revised manuscript) | Continuous tracking and segmentation of single neutrophil chasing and engulfing bacteria. **a**, The upper row shows a 2-channel sequential image dataset of bacteria and neutrophils over a 70-minute time window, while the lower row shows representative tracking and segmentation of a single neutrophil pursuing bacteria. The segmented neutrophil is outlined with a solid cyan line, and trajectory is shown as a purple gradient line. Bacterial center positions are marked yellow. Engulfment and digestion events are indicated by red-filled and red-outline triangles, respectively. **b**, Histogram of the number of bacterial engulfment events per neutrophil. The data used here were acquired via 2pSAM imaging of neutrophils, macrophages, and bacteria in mouse spleen, consistent with the dataset shown in **Fig. 4**. CELLECT results were generated using the model trained on sequence #3 of the mskcc-confocal dataset. Scale bars: 20 μ m

Minor point:

Comment 1:

Running the software on Mac platforms

The github repository states that Cuda is required to run the code, which de facto discards using it on Apple computers. It would be good if the code could run on this platform, given

how widely used it is in Biology. It is likely that the training phase will take much longer without a Cuda card, but users could run the inference with decent performance.

Response:

We thank the reviewer for highlighting the importance of supporting macOS platforms. The original implementation indeed required CUDA by default, which limited usability on Apple systems that do not support GPU acceleration.

To address this, we have revised the codebase to support CPU-only execution. Users can now toggle between GPU and CPU modes via a command-line flag, and the system will automatically fall back to CPU if no compatible GPU is detected. We also removed strict version dependencies (e.g., for PyTorch), which previously prevented installation on macOS. These changes have been tested on macOS 15.1(**Fig. R12**), and CELLECT now runs the full inference pipeline smoothly on Apple devices.

Tracking

```
cd CELLECT/
conda create -n collect python=3.11.7 mumpy scipy scikit-image -y
conda activate collect

pip install -r requirements.txt

# (Only required on Windows for GPU support)
pip install torch torchvision torchaudio --index-url https://download.pytorch.org/whl/cu126

# Model evaluation
python s-test.py --data_dir ../extradata/mskcc-confocal \
  --out_dir ./ \
  --model1_dir ./model/E+X3rdstr0-149.0-3.4599.pth \
  --model2_dir ./model/E+X3rdstr0-149.0-3.4599.pth \
  --model3_dir ./model/E+X3rdstr0-149.0-3.4599.pth
```

Win11+Nvidia 3090

```
Successfully installed sympy-1.13.1 torch-2.0.0+cu126

[collect] X:\test\CELECT>
[collect] X:\test\CELECT\python s-test.py --data_dir ../extradata/mskcc-confocal --out_dir ./ --model1_dir ./model/E+X3rdstr0-149.0-3.4599.pth --model2_dir ./model/E+X3rdstr0-149.0-3.4599.pth --model3_dir ./model/E+X3rdstr0-149.0-3.4599.pth
100% Total params: 2.54M | 12/268 [01:02:17:54, 4.20s/it]]
Iteration: 4% |
```

OSX 15.1 (Mac mini M4, 16 GB, 2024)

```
Successfully installed MarkupSafe-3.0.2 filelock-3.18.0 fsspec-2025.3.2 Jinja2-3.1.6 mmh3-1.3.0 opencv-python-4.11.0.86 pandas-2.2.3 python-dateutil-2.9.0.post0 pytz-2025.2 six-1.17.0 sympy-1.14.0 torch-2.7.0 tqdm-4.67.1 typing-extensions-4.13.2 tzdata-2025.2
100% Total params: 2.54M | 6/268 [06:43:41:34:47, 62.69s/it]]
Iteration: 2% |
```

Ubuntu20.4+Nvidia 3090

```
Successfully installed MarkupSafe-3.0.2 filelock-3.18.0 fsspec-2025.3.2 Jinja2-3.1.6 mmh3-1.3.0 nvidia-cublas-cu12-12.1.3.1 nvidia-cuda-cupti-cu12-12.1.105 nvidia-cuda-nvrtc-cu12-12.1.105 nvidia-cuda-runtimes-cu12-12.1.105 nvidia-cuda-toolkit-cu12-12.1.105 nvidia-cufft-cu12-12.0.2.54 nvidia-curand-cu12-10.3.2.106 nvidia-cusolver-cu12-11.4.5.107 nvidia-cusparselt-cu12-2.20.5 nvidia-ml-cu12-12.0.41 nvidia-nccl-cu12-2.16.1 nvidia-nvjitlink-cu12-12.0.41 nvidia-nvtx-cu12-12.1.105 opencv-python-4.11.0.86 pandas-2.2.3 python-dateutil-2.9.0.post0 pytz-2025.2 six-1.17.0 sympy-1.14.0 tzdata-2025.2 typing-extensions-4.13.2 tzdata-2025.2
[collect] inspur@client23:~/CELECT$ python s-test.py --data_dir ../extradata/mskcc-confocal --out_dir ./ \
> --model1_dir ./model/E+X3rdstr0-149.0-3.4599.pth \
* --model2_dir ./model/E+X3rdstr0-149.0-3.4599.pth \
* --model3_dir ./model/E+X3rdstr0-149.0-3.4599.pth
100% Total params: 2.54M | 400/400 [00:00:00:00, 394294.15it/s]
Iteration: 7% |
```

Training

```
# (Optional) Data preprocessing - only needed when training your own model, to speed up loading
python con-label-input.py --data_dir ../extradata/mskcc-confocal --out_dir ./ --num 1

# Model training
python s-train.py --data_dir ../extradata/mskcc-confocal --out_dir ./ --train 2 --val 2 --model_dir ./model/
```

Win11+Nvidia 3090

```
[collect] X:\test\CELECT\python con-label-input.py --data_dir ../extradata/mskcc-confocal --out_dir ./ --num 2
100% Total params: 2.54M | 275/275 [00:00:00, 409925.09it/s]
Iteration: 0% |
[collect] X:\test\CELECT\python s-train.py --data_dir ../extradata/mskcc-confocal --out_dir ./ --train 2 --val 2 --model_dir ./model/
100% Total params: 2.54M | 400/400 [00:00:00, 412823.201it/s]
Epoch 0/149
Iteration: 0% | 8/372 [00:00:07, 71s/it]
Iteration: 0% | 8/372 [00:00:24, 71s/it]
Iteration: 2% | 8/372 [02:19:41:38:51, 18.30s/it]]
```

Ubuntu20.4+Nvidia 3090, worker=0 (default)

```
[collect] inspur@client23:~/CELECT$ python con-label-input.py --data_dir ../extradata/mskcc-confocal --out_dir ./ --num 2
100% Total params: 2.54M | 275/275 [09:39:00:00, 2.11s/it]
[collect] inspur@client23:~/CELECT$ python s-train.py --data_dir ../extradata/mskcc-confocal --out_dir ./ --train 2 --val 2 --model_dir ./model/
100% Total params: 2.54M | 400/400 [00:00:00:00, 111520.00it/s]
Epoch 0/149
Iteration: 0% | 8/372 [00:00:07, 71s/it]
Iteration: 0% | 8/372 [00:00:07, 71s/it]
Iteration: 2% | 8/372 [01:18:58:54, 9.71s/it]]
```

Ubuntu20.4+Nvidia 3090, worker=16

```
[collect] inspur@client23:~/CELECT$ python s-train-16.py --data_dir ../extradata/mskcc-confocal --out_dir ./ --train 2 --val 2 --model_dir ./model/
100% Total params: 2.54M | 400/400 [00:00:00:00, 362046.09it/s]
Epoch 0/149
Iteration: 0% | 8/372 [00:00:07, 71s/it]
Iteration: 0% | 8/372 [00:25:07, 71s/it]
Iteration: 3% | 10/372 [00:37:00:21, 1.39s/it]
Iteration: 4% | 10/372 [00:37:08:21, 1.39s/it]
Iteration: 4% | 16/372 [00:43:06:15, 1.05s/it]]
```

Figure R12 | Cross-platform installation and training verification of CELLECT. We validated the CELLECT installation and execution process across 3 major platforms: macOS 15.1, Ubuntu 20.04, and Windows 11. All systems successfully completed installation and inference using default commands. On GPU-supported Windows and Ubuntu systems, additional training and preprocessing routines were executed and confirmed to run smoothly. Notably, on Linux, PyTorch's data loader acceleration using multiple workers provided faster training performance.

Comment 2:

Choice of nearest candidate cell centers

As we understand, removal of redundant cell centers and frame to frame linking is done by examining the 5 nearest cells, features wise and not distance wise. Could you confirm that we understood correctly? Maybe this could be made clearer in the main text (line 156) but especially in figure 1 legend (line 794). Moreover, could you justify this choice of 5 candidates instead of say 4 or 6?

Response:

We appreciate the reviewer for the constructive comments and apologize for any confusion. Contrary to the reviewer's understanding, candidate cell-center selection in CELLECT is based on spatial-based distance (i.e., Euclidean distance in the image domain), not feature-based distance. Specifically, for each target cell, we first identify the 5 spatially nearest candidate cell centers and then extract their embedding features for evaluation by a intra-frame multi-layer perceptron (MLP) classifier. Yes, it's 5. This classifier compares the target and candidate features and outputs 6 confidence scores: 5 corresponding to the selected candidates and 1 for the "None" class which indicates no valid match. While the learned embedding space provides meaningful cell-level features as shown in **Fig. R18 (Supplementary Fig. 3)** in the revised manuscript), the sparsity of annotations leads to limited supervision and prevents the model from learning globally consistent representations. This limitation is particularly pronounced in densely populated regions, where cells with similar appearances and large spatial distances may produce misleading similarities in the feature space. This is largely caused by insufficient training from sparse annotations and the patch-based learning strategy, which prevents the model from learning globally consistent embeddings.

Therefore, to reduce such ambiguity, CELLECT adopts a spatial-based distance strategy to pre-select candidate cells within the local neighborhood, after which their embeddings are compared for refined matching. This approach ensures greater robustness in both low- and high-density regions. We have added a new **Fig. R19 (Supplementary Fig. 11)** to demonstrate this problem. Feature-based distance selection performs comparably in early frames with sparse cells and is less sensitive to temporal subsampling. However, under increased density or division events, its performance degrades significantly. In contrast, spatial distance-based candidate selection maintains greater stability and accuracy.

The default setting of using 5 candidate cells reflect a balance between coverage and computational efficiency. It ensures the inclusion of potential division events while minimizing the risk of overfitting or increased noise. **Fig. R20 (Supplementary Fig. 12)** shows that this setting performs effectively under standard temporal resolution. For low

frame rate data, the number of candidates can be expanded by the user (e.g., to 10) to improve match coverage. Additionally, users can re-invoke the model multiple times with varying candidate pools if needed.

We have clarified this mechanism in the revised manuscript and **Methods** section under the subsection *Model architecture*. We have also updated the main texts and **Fig. 1** legend according to the reviewer's suggestion.

Figure R18 (Supplementary Figure 3 in the revised manuscript) | Clustering of feature vectors obtained by CELLECT via PCA across the adjacent frames. From 2 adjacent frames of input images, we extracted an embedding map for each frame and applied Principal Component Analysis (PCA) to reduce the dimensionality of feature vectors. In the resulting 3D PCA space, feature vectors corresponding to the same cell identity across the adjacent frames formed distinct clusters. Even dynamical changes during the division process, the feature distances of the daughter cells remained relatively close to that of the parent cell. By combining with Euclidean distances between cell centers, we inferred the potential division relationships between cells. The example shown here uses data from the Cell Tracking Challenge (CTC) Fluo-N3DH-CE dataset. The model used for feature extraction was trained on sequence #3 of the mskcc-confocal dataset.

a Inference across different frame rates

Figure R19 (Supplementary Figure 11) | Influence of imaging frame rate on tracking performance under different candidate selection strategies.

a, Illustration of 3 different levels of temporal down-sampling were used for evaluation: $\times 1$ (75-s interval), $\times 2$ (150-s interval), and $\times 3$ (225-s interval) down-sampling. **b**, Quantitative tracking performance of CELLECT under 3 conditions of temporal down-sampling. **c**, Schematic illustration of 2 cell selection strategies: left, spatial-distance-based selection of the 5 nearest neighbors (used in CELLECT); right, feature-distance-based selection of the 5 most similar feature vectors. **d**, Error curves (FN + FND + IS) over time for both strategies under different temporal down-sampling. The feature distance-based strategy performs comparably at early frames when cell density is low, even under reduced temporal resolution, but its performance degrades rapidly as cell density increases over time. **e**, Circular tree diagrams displaying tracking errors across all 6 settings. All evaluation data were drawn from mskcc-confocal dataset #2; the model was trained on mskcc-confocal dataset #3 without temporal down-sampling. Scale bar: 10 μm .

Figure R20 (Supplementary Figure 12 in the revised manuscript) | Influence of imaging frame rate on tracking performance under different training dataset and candidate numbers used. a, Illustration of the 2 temporal resolutions used in the inference process: $\times 1$ (75 s) and $\times 2$ (150 s) down-sampling. b, Tracking performance of 3 different configurations for different temporal down-sampling in inference: the default

CELLECT configuration, model using 10 candidate cells, and model using 5 candidate cells but trained with different temporal resolutions (randomly incorporating dataset with $\times 1$, $\times 2$ and $\times 3$ temporal down-sampling). All configurations show comparable performance under $\times 1$ temporal down-sampling. However, under $\times 2$ temporal down-sampling, models trained with an expanded candidate pool or mixed-temporal-resolution data exhibit reduced false negative (FN) rates, indicating improved robustness to lower temporal resolution. **c**, Circular lineage diagrams corresponding to the 6 experimental settings. All results were drawn from mskcc-confocal dataset #2, with models trained on dataset #3 with and without temporal down-sampling. Scale bar: 10 μm .

Comment 3:

Figures clarity

The figures are very clear and illustrate well the efficiency of the tool. However the figure 3 and 4 could be improved for readability. For instance the figure 3 includes snapshots of images with track overlaid, but at a very low zoom level. It makes it hard to see more than a colored cloud. Maybe image insets could be added that would show at a fine resolution level what the image data look like at the pixel level, and what a single cell track looks like. It would also be informative and illustrate what events Imaris misses and CELLECT gets.

Response:

We thank the reviewer for the constructive suggestion. We agree that the low zoom level in **Fig. R21 (Fig. 3 in the revised manuscript)** and **Fig. R16 (Fig. 4 in the revised manuscript)** may limit readability and make it difficult to appreciate individual tracks or clearly distinguish differences between tracking methods. Following the reviewer's suggestion, we have incorporated more enlarged insets as possible into both **Fig. R21 (Fig. 3 in the revised manuscript)** and **Fig. R16 (Fig. 4 in the revised manuscript)**.

Furthermore, **Supplementary Figs. 16–17**, previously included in the original manuscript, were intended to provide direct comparative examples between CELLECT and Imaris in challenging regions with an extended field of view. These examples showcase specific cases where Imaris either fails to detect a cell or produces incorrect links, while CELLECT successfully reconstructs the correct trajectory. We believe these visual enhancements can improve figure readability and further demonstrate CELLECT's advantages in resolving complex tracking scenarios.

Figure R21 (Figure 3 in the revised manuscript) | 3D tracking of dense immune cells in mouse lymph nodes. **a**, Example orthogonal maximum intensity projections (MIP) of dual-color labeled T cell imaged by 2pSAM in mouse lymph nodes. T cells were labeled with both DsRed and GFP at a 1:5 ratio. Sparsely labeled cells are generally co-labeled with GFP to be used as ground-truth during evaluation. **b**, Tracking result of the dense GFP channel obtained by CELLECT. **c**, Tracking result of the sparse DsRed channel obtained by Imaris using as the ground-truth. **d**, Tracking result of the dense GFP channel obtained by Imaris. **e**, Accurately tracking traces in the dense channel obtained by CELLECT (about 91.5% of all traces). We define the overlap ratio of the total tracking length compared with the ground-truth trace

in the sparse channel larger than 0.8 as the accurately tracked cell. **f**, Accurately tracking traces in the dense channel obtained by Imaris (about 70.7% of all traces). **g**, Comparison of the traces of the same cell in dense datasets obtained by CELLECT and Imaris. While Imaris cannot track the same cell consistently, CELLECT can track in a continuous way. **h**, Bar chart showing the comparisons of the tracking accuracy under different overlap thresholds used to define accurately tracked cells. **i**, Example orthogonal MIPs of B cells in mouse lymph nodes during the formation process of a GC. **j**, 3D tracking traces obtained by CELLECT with different colors corresponding to different time stamps. **k**, Spatiotemporal distribution of the events of cell division (corresponding to each dot) identified by CELLECT, indicating that a large amounts of cell division happen during GC formation. **l**, Comparison of the traces of the same cell obtained by CELLECT and Imaris, indicating a cell-division event. **m**, Comparison of cell proliferation and accumulated cell division events over time in both GC and non-GC regions. Note: The term ‘sparse’ in this figure refers to biological sparsity due to selective DsRed expression in a subset of cells, not to sparsity in manual annotations used during model training. All data were acquired from live imaging of mouse lymph nodes using two-photon scanning adaptive microscopy (2pSAM). Panels (**a–h**) use dual-labeled T cell datasets, while panels (**i–m**) use B cell datasets during germinal center formation. All results were obtained using the CELLECT model trained on sequence #3 of the mskcc-confocal dataset. Scale bars: 100 μm .

Figure R16 (Figure 4 in the revised manuscript) | Simultaneous tracking and segmentation of CELLECT facilitate accurate identification of immune interactions among bacteria, neutrophils, and macrophages. **a**, Experimental and processing schematics: Bacteria were injected into the surgically exposed spleen of a mouse, and activities of bacteria, neutrophils and macrophages were monitored by 2pSAM at 30 volumes per second over the subsequent 4 hours. The events of bacterial phagocytosis by 2 different immune cell types were identified by the tracking and segmentation of CELLECT. **b**, Example orthogonal MIP of 3-color imaged by 2pSAM in mouse spleen. **c**, Visualization of bacterial and neutrophil trajectories extracted by CELLECT. The combined display highlights dynamic immune–bacteria interactions. **d**, Segmentation results of neutrophils obtained by CELLECT and Imaris. **e**, Segmentation results of macrophages obtained by CELLECT and Imaris. The results of Imaris in **d** and **e** are sensitive to different intensity thresholds, so we adjusted manually for optimum accuracy and demonstrated the influence in **Supplementary Fig. 19**. The inverted triangles indicate distinct structures that were inaccurately segmented as neutrophils but resembled macrophages. **f**, Quantification of bacteria in different states: extracellular, intracellular within neutrophils, and within macrophages. **g**, Spatiotemporal distribution of neutrophil-phagocytosis events indicated by dots with different colors labeling the time points. **h**, Spatiotemporal distribution of macrophage-phagocytosis events. **i**, MIPs of bacteria engulfed by a neutrophil at different time points. **j**, Corresponding traces indicating the dynamic interaction process with the relative distance changes over time shown below. **k**, MIPs of bacteria engulfed by a macrophage at different time points. **l**, Corresponding traces indicating the dynamic interaction process with the relative distance changes over time shown below. All data were acquired by live 2pSAM imaging of surgically exposed mouse spleen over a 4-hour timespan. CELLECT results were obtained using the model trained on sequence #3 of the mskcc-confocal dataset. Scale bars: 50 μm (**b-h**) and 5 μm (**i-k**).

Comment 4:

Methods section, cell segmentation and radius estimate from sparse annotations

The method section could do with more details on how the cell segmentation and radius estimate are computed. Particularly if the article keeps a focus on cell segmentation, as showcased in the use-case ‘quantitative analysis of intercellular interactions in vivo’. For instance, how can CELLECT achieve a good performance on the contour of a cell if the sparse annotations used for training only include the position of cell centers?

Response:

We thank the reviewer for the insightful question. We would like to clarify that CELLECT is not designed to predict precise cell contours, but rather to estimate coarse

cell regions based on a center probability map and a learned radius, using only sparse center annotations. During training on the mskcc-confocal dataset, we utilized the provided radius values to assign representative cell sizes around each annotated center. In datasets such as the CTC, which provides weak or algorithm-generated segmentation masks (i.e., silver standards), our method can leverage these regions to estimate typical cell radii and spatial structure for training. This enables CELLECT to achieve segmentation scores on the CTC leaderboard. However, we emphasize that all models used in this paper were trained without any segmentation mask supervision. The model learns soft spatial regions through contrastive learning. Feature vectors within the radius are encouraged to become similar, while those outside are pushed apart. This results in localized, discriminative embeddings that help distinguish between the center cell, nearby neighbors, and the background. **Fig. R22 (Supplementary Fig. 4 in the revised manuscript)** illustrates this spatial behavior. This may be the reason that CELLECT achieves high scores also in the CTC challenge for segmentation. Nevertheless, as shown in **Fig. R22 (Supplementary Fig. 4 in the revised manuscript)** and **Fig. R16 (Fig. 4 in the revised manuscript)**, CELLECT frequently recovers spatially coherent and biologically meaningful cell regions across various datasets. These coarse region estimates are highly useful for downstream analyses such as measuring cell-to-cell proximity or overlapping, even in the absence of precise contour information.

As the reviewer suggested, we have added a dedicated subsection titled *Segmentation and radius estimation* in the **Methods** section.

Figure R22 (Supplementary Figure 4 in the revised manuscript) | Visualization of the embedding map obtained by CELLECT. To analyze the spatial patterns captured by the model, we extracted embedding map corresponding to each voxel and computed their differences with surrounding voxels. This was achieved by convolving the squared values of the embedding map with a dilated convolution kernel, simulating neighborhood interaction. A dilation rate of 1 represents the average difference with adjacent voxels, while a dilation rate of 5 represents the difference with voxels at a distance of 5 voxels in all directions. The resulting feature distance map highlights the magnitude and positive or negative of feature differences between distinct cell regions in the image. At the boundaries, the opposing patterns between the central and surrounding areas particularly emphasize the differences in feature values between cellular and non-cellular regions. These differences can also assist in optimizing cell segmentation during the inference phase, especially under sparse annotation conditions. The artifacts observed at patch boundaries arise from the independent inference of each patch, which do not affect the accuracy of segmentation or tracking, as only the embedding vectors at predicted cell centers are used for downstream

analysis, and the boundary variations are negligible. The visualization shown here is based on the Cell Tracking Challenge (CTC) Fluo-N3DH-CE dataset. The model used was trained on sequence #3 of the mskcc-confocal dataset.

Comment 5:

Methods section, benchmark and training datasets

Could the methods section be clarified about what data was used to train the main model used in this paper and the benchmarks? The main text states that it is trained on the CTC datasets (lines 127, 128). In the methods section, it is stated that it is trained on the mskcc-confocal dataset (lines 509-510). Maybe the datasets used for contrastive-learning and the center detection learning are not the same? Additionally, the benchmarks are said to be performed on the mskcc-confocal and nis-ls datasets (lines 180, 541-541). But this would mean that the benchmark was done on the same dataset used to train the model.

Response:

We thank the reviewer for pointing out this important inconsistency. We apologize for the confusion caused by inconsistent references to multiple datasets in different sections. To clarify, no part of our model was trained using any Cell Tracking Challenge (CTC) dataset. All training and evaluation presented in this work were conducted solely on the mskcc-confocal and nih-ls datasets. These datasets each contain 3 independent 3D time-lapse sequences with manually annotated sparse center-point trajectories.

For all experiments reported in **Fig. R15 (Fig. 2 in the revised manuscript)** and **Tables R7-R8 (Supplementary Tables 1-2 in the revised manuscript)**, we adopted a 3-fold cross-validation protocol, following the design used in the Linajea series of works. Specifically, for each run, we trained on 1 sequence, validated on another, and tested on the third. This process was repeated for all combinations, and the final performance was averaged across the 3 folds to ensure strict separation between training and evaluation. For visualizations and downstream applications in **Figs. 3–5 in the revised manuscript**, we used models trained on mskcc-confocal dataset #3, unless otherwise stated. We have now explicitly stated these dataset choices in the updated all figure legends and Methods section to improve clarity. Additionally, we clarify that the CTC leaderboard results cited in the main text were obtained via external submission to the challenge website and were not involved in any training or evaluation steps described in this manuscript. These scores serve as supplementary evidence of CELLECT's performance on a standardized test set and are clearly separated from our main experimental design.

To ensure better transparency and reproducibility, we have annotated the training dataset and the source of inference data in every figure caption throughout the manuscript. A consolidated explanation of dataset usage has also been included in the revised **Methods**

section, under the subsection *C. elegans embryo public dataset*, which provides a unified account of training and evaluation protocols.

Figure R15 (Figure 2) | Benchmarking of CELLECT with state-of-the-art algorithms in Cell Tracking Challenge. a-d, Performance evaluation on the dataset of mskcc-confocal. **e-h,** Performance evaluation on the dataset of nih-light sheet. **a** and **e**, The average number of errors per 1,000 ground-truth edges for each error type. The errors were categorized into 5 conditions: false-positive edges, false-negative edges, identity switches, false positive divisions, and false negative divisions. **b** and **f**, The proportion of error-free tracks (marked as tracking accuracy) versus a tracking duration in terms of frames. **c** and **g**, Comparisons of average computational time cost for each frame among different algorithms. **d** and **h**, The hierarchical circular diagram of lineage tracing with the grey lines indicating the cell division. The bold black lines and circles indicate the false negative identity switch and division, respectively. The scores for the 3 models other than CELLECT and the accuracy curves of Linajea are derived from the previous report³⁵. ‘Linajea + csc + sSVM’ refers to an enhanced version of Linajea that incorporates a cell state classifier (csc) to model biological states (e.g., division, continuation, polar body) and a structured Support Vector Machine (sSVM) to learn tracking cost functions from data. All evaluations were

conducted using the first 270 frames of each of the 3 sub-volumes for both datasets (mskcc-confocal and nih-light sheet), following a cross-validation protocol. The CELLECT models used in each comparison were trained on one of the 2 other sub-volumes. Further details on the training and evaluation splits are provided in **Supplementary Table 1** and **2**. Scale bars: 10 μm .

	FP	FN	IS	FP-D	FN-D	Division	Sum
Mskcc-confocal 270 frames							
StarryNite*	7.9	13	0.62	0.58	1.2	1.8	24
Linajea*	3.6	5.5	0.062	0.89	0.26	1.2	10.3
Linajea+csc+sSVM*	3.7	5.6	0.046	0.053	0.4	0.46	9.6
CELLECT	1.26	3.56	0.013	0.076	0.41	0.49	5.32
Nih-ls 270 frames							
StarryNite*	22	18	2.4	0.66	1.6	2.2	45
Linajea*	12	6.5	0.46	1.5	0.4	1.86	21
Linajea+csc+sSVM*	13	5.3	0.59	0.20	0.49	0.69	20
CELLECT	6.37	3.68	0.070	0.777	1.14	1.92	12.04

Table R7 (Supplementary Table 1 in the revised manuscript) | Quantitative results on mskcc-confocal and nih-ls dataset. For descriptions of the error metrics, see **Supplementary Fig. 6**. All values represent absolute errors normalized per 1,000 ground-truth (GT) edges; best value bold. * The values are obtained from the previous report (Hirsch, P. *et al. Medical Image Computing and Computer Assisted Intervention 2022*, 25–35 (2022)).

Train-id	Test-id	FP	FN	IS	FP-D	FN-D	Division	Sum
Mskcc-confocal 270 frames								
1	2	1.38	2.13	0.039	0.099	0.48	0.58	4.13
	3	0.9	3.62	0.039	0.039	0.26	0.30	4.86
2	1	1.32	5.33	0.0	0.097	0.50	0.60	7.25
	3	1.89	5.20	0.0	0.14	0.33	0.47	7.56
3	1	1.12	2.75	0.0	0.058	0.39	0.40	4.32
	2	0.95	2.34	0.0	0.02	0.50	0.52	3.81
Mean		1.26	3.56	0.013	0.076	0.41	0.49	5.32
Nih-ls 270 frames								
1	2	4.71	4.36	0.018	0.712	1.57	2.28	11.37
	3	6.14	6.38	0.049	0.5	0.77	1.27	13.84
2	1	2.44	5.21	0.064	0.614	1.67	2.28	9.99
	3	15.41	1.37	0.129	1.565	0.364	1.93	18.84
3	1	3.43	3.71	0.085	0.678	1.01	1.69	8.91
	2	6.09	1.05	0.074	0.591	1.46	2.05	9.27
Mean		6.37	3.68	0.070	0.777	1.14	1.92	12.04

Table R8 (Supplementary Table 2 in the revised manuscript) | Quantitative results for each fold in the cross-validation experiments on mskcc-confocal and nih-ls data. For descriptions of the error metrics, see

Supplementary Fig. 6. All values represent absolute errors normalized per 1,000 ground-truth (GT) edges.

Comment 6:

Methods section, details on the parameters used for comparison

The methods section should list what are the versions of the tools used for comparison (ilastik, Imaris StarryNite, TrackMate) and with what parameters. The references for the open-source tools should be cited.

Response:

We thank the reviewer for highlighting this important point. In the revised manuscript, we have added the tool versions, key parameter settings, and citations for all comparative tracking methods used in this study. The following tools were used:

- Imaris (Bitplane): version 9.0.1, using the default “Spots” detection and “Tracking” modules. The estimated spot diameter was manually adjusted for each dataset based on voxel size. Gap-closing was enabled with a maximum of 2–3 frames for most datasets, and up to 2000 frames for neuronal data to accommodate frequent signal dropout.
- TrackMate (via Fiji): version 7.10.1, using the provided pre-trained Weka model detector applied in a slice-by-slice manner. The gap-closing interval was set to 2,000 frames for neuronal datasets, consistent with the Imaris configuration.
- StarryNite (2020 version): We adopted the default parameter configuration distributed with the 2020 version of StarryNite and modified the cell radius range settings based on the ground-truth annotation files.
- ilastik: This tool was not used in our experiments.

The corresponding tool citations and configuration details have been added to the updated **Methods** section titled *Comparative methods and parameter settings*.

Comment 7:

Discussion

While interesting, the discussion is a bit short and would benefit from a section on the limitations of the framework. For instance:

Since its input is limited to 2 adjacent frames, CELLECT does not seem to be able to deal with missing cell detections, when a cell is detected at a frame t , but disappears at $t+1$ only to be redetected at $t+2$. Does it happen in the datasets you used? How would you remove this limitation?

You talk about "temporal data-driven optimizations". Do you plan to extend CELLECT with some of those? Which ones? What do you think would be the impact in terms of processing time and memory?

For the data processing module available on the repository, the documentation says that the output folder could be hundreds of GB, which is a noteworthy limitation when the rest of CELLECT is quite efficient. Why is it this heavy? Any ideas on how to make it less so? What are the events missed by CELLECT? The benchmark reveals that CELLECT achieves a tracking accuracy of 46% (a huge improvement from the 22% of Linajea that takes more computer power). The discussion could be augmented with a description of the events that are missed by CELLECT (that could also be illustrated in figure 2). This would prompt the field as to where to put efforts next.

Response:

We thank the reviewer for these thoughtful and constructive suggestions. In the revised Discussion section, we have added following contents that outlines the current limitations of the CELLECT framework and future directions for improvement based on all the suggestions.

CELLECT currently matches cells between 2 adjacent frames to ensure efficient inference. However, when cells temporarily disappear (e.g., due to occlusion or weak signal) and reappear in subsequent frames, broken trajectories can occur. Such cases are observed in challenging datasets such as the neuronal recordings shown in **Fig. 5** in the original manuscript). To mitigate this, we implemented an optional reconnection strategy that allows CELLECT to link terminal and emerging tracks within a 200-frame window. This post-hoc linking mechanism helps restore continuity without significantly increasing computational burden. Additionally, we evaluated CELLECT under lower temporal resolution by performing matching between non-adjacent frames (e.g., t and $t+2$, t and $t+3$). The results, presented in **Figs. R19, R20, R23, R24 (Supplementary Figs. 11-14** in the revised manuscript), demonstrate that CELLECT maintains robust tracking performance even under sparse temporal sampling.

The current version does not explicitly model long-term motion patterns. In future work, we plan to integrate temporal modules such as trajectory priors, recurrent networks, or motion predictors to enhance tracking continuity under signal dropout or ambiguous conditions. Light-weight refinements, such as track smoothing or short-gap interpolation, introduce negligible computational overhead, while more complex architectures may entail higher resource costs depending on their design.

The large preprocessing size (up to several hundred GB) mentioned in the repository arises from the conversion of sparse annotations into dense matrices for faster training. These files are not needed during inference. We clarified that this step is optional and can be bypassed by using frame range settings, compact data types (e.g., uint8), or on-the-fly conversion during training. This module is a utility for training efficiency and not part of the core framework.

As shown in **Fig. R15 (Fig. 2** in the revised manuscript), most errors in CELLECT occur in 3 common conditions: (1) signal dropout in low-SNR regions, (2) limited axial resolution that causes ambiguity in depth, and (3) high-density cell clusters, especially in

late-stage embryos, where complex divisions and overlapping signals make identity distinguishing difficult. These lead to errors such as fragmentation, identity switches, or missed divisions.

We have added these challenges and directions to the revised **Discussion** section.

a Inference across different frame rates

Figure R19 (Supplementary Figure 11) | Influence of imaging frame rate on tracking performance under different candidate selection strategies.

a, Illustration of 3 different levels of temporal down-sampling were used for evaluation: $\times 1$ (75-s interval), $\times 2$ (150-s interval), and $\times 3$ (225-s interval) down-sampling. **b**, Quantitative tracking performance of CELLECT under 3 conditions of temporal down-sampling. **c**, Schematic illustration of 2 cell selection strategies: left, spatial-distance-based selection of the 5 nearest neighbors (used in CELLECT); right, feature-distance-based selection of the 5 most similar feature vectors. **d**, Error curves (FN + FND + IS) over time for both strategies under different temporal down-sampling. The feature distance-based strategy performs comparably at early frames when cell density is low, even under reduced temporal resolution, but its performance degrades rapidly as cell density increases over time. **e**, Circular tree diagrams displaying tracking errors across all 6 settings. All evaluation data were drawn from mskcc-confocal dataset #2; the model was trained on mskcc-confocal dataset #3 without temporal down-sampling. Scale bar: 10 μm .

Figure R20 (Supplementary Figure 12 in the revised manuscript) | Influence of imaging frame rate on tracking performance under different training dataset and candidate numbers used. a, Illustration of the 2 temporal resolutions used in the inference process: $\times 1$ (75 s) and $\times 2$ (150 s) down-sampling. b, Tracking performance of 3 different configurations for different temporal down-sampling in inference: the default

CELLECT configuration, model using 10 candidate cells, and model using 5 candidate cells but trained with different temporal resolutions (randomly incorporating dataset with $\times 1$, $\times 2$ and $\times 3$ temporal down-sampling). All configurations show comparable performance under $\times 1$ temporal down-sampling. However, under $\times 2$ temporal down-sampling, models trained with an expanded candidate pool or mixed-temporal-resolution data exhibit reduced false negative (FN) rates, indicating improved robustness to lower temporal resolution. **c**, Circular lineage diagrams corresponding to the 6 experimental settings. All results were drawn from mskcc-confocal dataset #2, with models trained on dataset #3 with and without temporal down-sampling. Scale bar: 10 μm .

Figure R23 (Supplementary Figure 13 in the revised manuscript) | Evaluation of the reproducibility of the model. To assess the reproducibility of the model performance, we trained the model 5 times using identical parameters but with non-deterministic random sampling during training. The results revealed that, despite minor fluctuations, the model's overall performance remains highly consistent and reproducible. Error connections are highlighted in pink, while correct connections are depicted in grey. All evaluation data in this figure were drawn from mskcc-confocal dataset #2. All models were trained using mskcc-confocal dataset #3.

Figure R24 (Supplementary Figure 14 in the revised manuscript) | Evaluation of the reproducibility of cell linking during different imaging frame rate. To evaluate the reproducibility of CELLECT’s cell linking under different imaging frame rate, we selected 3 representative target cells (Target #1–3) from mskcc-confocal dataset #2 and selected 5 nearest spatial neighbors at frames $t+1$, $t+2$, and $t+3$ for testing. **a**, Location of the target cells at time t (frame 1) and the locations of candidate cells across the 3 subsequent frames. **b**, Similarity scores predicted by 5 independently trained CELLECT models for the 5 candidates. To ensure consistent input across models, redundant point removal was not applied intentionally, resulting in occasional duplicate redundant cell points (e.g., Target #1 and #2 at frame 1 linked to frame 2). This configuration increases linking difficulty by expanding the temporal gap, thereby simulating more complex multi-frame association scenarios. While correct links often correspond to the nearest candidate in adjacent-frame conditions, longer intervals such as frame 1 to frame 3 or frame 4 require the model to rely more on embedding features for

accurate decisions. The results demonstrate that CELLECT produces consistent linking performance across different frame intervals. However, under low imaging frame rate and in the presence of cell divisions (e.g., frame 1 linked to frame 4), a slight drop in accuracy is observed. All 5 models were independently trained on mskcc-confocal dataset #3. Scale bar: 10 μm .

Figure R15 (Figure 2) | Benchmarking of CELLECT with state-of-the-art algorithms in Cell Tracking Challenge. a-d, Performance evaluation on the dataset of mskcc-confocal. **e-h**, Performance evaluation on the dataset of nih-light sheet. **a** and **e**, The average number of errors per 1,000 ground-truth edges for each error type. The errors were categorized into 5 conditions: false-positive edges, false-negative edges, identity switches, false positive divisions, and false negative divisions. **b** and **f**, The proportion of error-free tracks (marked as tracking accuracy) versus a tracking duration in terms of frames. **c** and **g**, Comparisons of average computational time cost for each frame among different algorithms. **d** and **h**, The hierarchical circular diagram of lineage tracing with the grey lines indicating the cell division. The bold black lines and circles indicate the false negative identity switch and division,

respectively. The scores for the 3 models other than CELLECT and the accuracy curves of Linajea are derived from the previous report³⁵. ‘Linajea + csc + sSVM’ refers to an enhanced version of Linajea that incorporates a cell state classifier (csc) to model biological states (e.g., division, continuation, polar body) and a structured Support Vector Machine (sSVM) to learn tracking cost functions from data. All evaluations were conducted using the first 270 frames of each of the 3 sub-volumes for both datasets (mskcc-confocal and nih-light sheet), following a cross-validation protocol. The CELLECT models used in each comparison were trained on one of the 2 other sub-volumes. Further details on the training and evaluation splits are provided in **Supplementary Table 1** and **2**. Scale bars: 10 μm .

Comment 8:

CELLECT name

CELLECT is a meaningful name with a nice ring to it. However, this name – or a close variation – is already used by several other tools and softwares as shown when doing a web search on the name. For example, there are at least 2 GitHub repositories with this name for a biology related tool in Python. See also <https://bmcbioinformatics.biomedcentral.com/articles/10.1186/s12859-016-0927-7>. We fear that this choice of name will reduce the visibility of CELLECT.

Response:

We appreciate the reviewer careful and thoughtful suggestion. We fully agree that name uniqueness plays a crucial role in ensuring the visibility and long-term recognizability of computational tools. The name CELLECT was chosen as an acronym for Contrastive Embedding Learning for Large-scale Efficient Cell Tracking, which directly reflects the core design and motivation of our framework. While we are aware that similar names have been used in other biological software projects—such as tools for expression prioritization or genomic analysis—our method addresses a fundamentally different research area, focusing on cell tracking in microscopic imaging data. Although some existing tools share similar acronyms with our method and also fall within the broad field of biology, we believe CELLECT is clearly distinguishable due to its specific focus on cell tracking in microscopic imaging data.

To reduce potential confusion, we will prominently include the full acronym definition in the abstract, introduction, and code repository. We will also add clarifying statements to the README and documentation to help distinguish CELLECT from similarly named tools. If the reviewer were to strongly recommend a change in name, we would be fully open to modifying it accordingly.

Comment 9-10:

Duplicated reference

The references 20 and 28 (Weigert et al.) are duplicated.

Some typos

- lines 288-290: "we usually need to label the cell membrane with relatively weaker signals are distributed centrally." => grammar
- lines 363-364: "In this case, the framework be scaled up as a foundation model" => grammar
- lines 496-497: "y , $\sim q(y|x)$ represent points from distinct labeled cells. represent points from distinct labeled cells" => duplicated text
- line 547: typo in CELLECT name

Response:

We thank the reviewer for pointing out these errors.

- The duplicate reference to Weigert et al. (references 28) has been removed.
- The sentence on lines 288–290 has been revised for grammar and clarity.
- The grammatical issue in lines 363–364 has been corrected ("the framework can be scaled up...").
- The duplicated text in lines 496–497 has been deleted.
- The typo in the CELLECT name on line 547 has been fixed.

All identified issues have been corrected in the revised manuscript. We appreciate the reviewer's great efforts.

Reviewer #4

Comment A: Summary of the key results:

The authors introduce CELLECT, a contrastive learning-based method for learning embeddings at the voxel level in 3D image volumes. CELLECT achieves computational efficiency by avoiding global-in-time optimization, instead leveraging a lightweight 3D model (<3M parameters) and two MLPs (Intra-frame and Inter-frame). The authors report it to be 56 times faster than Linajea (Malin-Mayor et al.). The method demonstrates competitive tracking performance across multiple datasets: two public *C. elegans* datasets, real-time large-scale B cell tracking in mouse lymph nodes, and neural signal extraction in *Drosophila*

Response:

We greatly appreciate the reviewer's positive assessment of CELLECT, particularly regarding its efficiency, accuracy, and application potential in biological imaging. The review correctly highlights the key innovations of CELLECT, such as using contrastive learning for voxel-level embeddings and the light-weight design with minimal parameters. We are pleased that the reviewer finds CELLECT's computational efficiency, especially in the linking step, to be notable. In the following sections, we address the specific points

raised by the reviewer to further improve the manuscript and clarify important aspects of our work.

- We conducted a quantitative ablation study (**Table R9**) demonstrating that CELLECT's core modules—CEN, intra-frame MLP, and inter-frame MLP—are highly interdependent, and disabling any of them substantially degrades tracking accuracy and stability, confirming their essential roles in reliable cell tracking.
- We updated the manuscript to clarify Linajea's limitations at low frame rates due to its reliance on dense temporal windows. **CELLECT's robustness was assessed on temporal-downsampled datasets**, demonstrating strong performance even with reduced temporal resolution. The Introduction was revised to better describe Linajea's limitations.
- We explained that tiling artifacts in **Supplementary Fig. 4** arise from background suppression at patch edges, not padding issues, and clarified that these artifacts don't affect CELLECT's tracking performance in the figure caption.
- We specified that CELLECT was officially submitted to the CTC under "**THU-CN (3)**" and ranked first for both tracking and segmentation in the Fluo-N3DH-CE dataset. We added a link to the CTC leaderboard and noted that the method description is pending update. Quantitative results are now included in the manuscript.
- We clarified that the "Tracking Accuracy" metric originates from Linajea (Malin-Mayor et al.) and cited it in the **Main**. We also updated the requirements.txt file with missing dependencies, revised the training script to create the "Trained models" directory, and are improving code modularity and documentation.

Comment F: Suggested improvements: experiments, data for possible revision

Major point:

Comment 1:

****Ablation Study on CEN, Intra-frame MLP, and Inter-frame MLP Contributions:****

To better understand the individual contributions of the CEN, Intra-frame MLP, and Inter-frame MLP modules, I suggest including a quantitative ablation study. Specifically, on at least one dataset, it would be valuable to report how Tracking Accuracy and/or other relevant metrics change when adding each component incrementally. While Supplementary Figure 7 provides qualitative insight, clear quantitative results would strengthen the argument and clarify the role of each module, particularly the two MLPs.

A table similar to the one below would be informative:

3D-UNet CEN Intra-frame MLP Inter-frame MLP Tracking Accuracy

Yes	No	No	No	?
Yes	Yes	No	No	?
Yes	Yes	Yes	No	?
Yes	Yes	Yes	Yes	?

Response:

We thank the reviewer for the constructive suggestion and fully agree that understanding the individual contributions of each module is essential to interpret the effectiveness of the CELLECT framework. In response, we conducted a quantitative ablation study summarized in **Table R9**, where we step by step disabled the Center Enhancement Network (CEN), Intra-frame MLP, and Inter-frame MLP modules, sequentially as required by the reviewer. Tracking Accuracy was computed following the definition used in the Linajea benchmark series (Malin-Mayor et al., 2023, *Nat. Biotechnol.* 41, 44–49; Hirsch et al., 2022, MICCAI, pp. 25–35), that is, the proportion of uninterrupted, error-free tracks (no FN, FND, or IS) sustained over time intervals of 50, 100, 150, and 200 frames. We also report the total number of error links (FN+FND+IS) to reflect the cumulative impact on linking stability. In the complete configuration, CELLECT achieved Tracking Accuracy values of 0.94, 0.84, 0.61, and 0.32 at increasing durations, with 138 total link errors. Disabling all 3 modules reduced the accuracy to 0.76, 0.42, 0.13, and 0.02, with 382 errors. To ensure functional evaluation, we replaced the inter-frame MLP with a 2-nearest-neighbor heuristic for linking (to handle divisions), omitted redundancy filtering when removing the intra-frame MLP (which had little impact due to low redundancy in the dataset), and extracted centers from raw confidence maps when removing the CEN, which introduced considerable noise and degraded matching. For fairness, we focused our error analysis on FN, FND, and IS types, which directly reflect linking reliability. The evaluation was conducted on the mskcc-confocal#2 subset, which features substantial embryonic lineage path variation and is sensitive to architectural differences. Models were trained on the neighboring mskcc-confocal#3 subset.

We further clarify that the modules in CELLECT are highly interdependent, and removing any one leads to rapid degradation in tracking accuracy and stability. The 3D U-Net backbone and CEN together provide center localization functionality. The CEN sharpens the confidence map by suppressing background and enhancing core regions. When merged into the backbone or simplified by reducing resolution levels, the output becomes noisy or redundant, as shown in **Figs. R25 and R26 (Supplementary Figs. 7 and 8)**, which impairs downstream association. The intra-frame and inter-frame MLPs support CELLECT’s matching decisions by evaluating embedding similarity, spatial displacement,

and division likelihood. For each cell, the model selects 5 nearby candidates and outputs confidence scores across them and a “None” class. This framing enables robust probabilistic linking without reliance on hand-tuned thresholds. Without these MLPs, the system relies solely on geometric proximity, which performs poorly in dense or ambiguous conditions. The results in **Table R9** confirm that each module is essential for maintaining CELLECT’s accuracy and robustness.

3D U-Net	CEN	Intra-frame MLP	Inter-frame MLP	Tracking Accuracy ²				Error Links (FN+FND+IS)
				Length (frames)				
				50	100	150	200	
Yes	No	No	No	0.76	0.42	0.13	0.02	382
Yes	Yes	No	No	0.84	0.55	0.21	0.10	219
Yes	Yes	Yes	No	0.85	0.57	0.23	0.10	210
Yes	Yes	Yes	Yes	0.94	0.84	0.61	0.32	138

Table R9 (Supplementary Table 4) | Quantitative ablation study of individual modules in the CELLECT framework. We assessed the contributions of the Center Enhancement Network (CEN), intra-frame MLP, and inter-frame MLP by progressively enabling each. Tracking Accuracy is defined as the proportion of uninterrupted, error-free trajectories over 50 to 200 frames² and Error Links refer to the total tracking errors (false negatives, false negative divisions, and identity switches). In ablated models, nearest-neighbor matching replaced the inter-frame MLP, and redundancy filtering was disabled without the intra-frame MLP. Removing CEN resulted in noisy center proposals due to unenhanced confidence maps. Model was trained on mskcc-confocal #3 and evaluated #2.

Figure R25 (Supplementary Figure 7 in the revised manuscript) | Ablation study of CEN module on tracking performance. **a**, The CELLECT model structure used in this study. **b**, A modified version of the CELLECT model without the CEN subnetwork. While the CEN structure was removed, its training methodology and output format were retained to mitigate performance drop. Without these modifications, simply removing the CEN would result in outcomes similar to the 2-level configuration output scenario, leading to significantly worse performance and making it difficult to compare model structures. **c**, We validated the performance differences of 2 methods on the mskcc-confocal dataset^{1,2} #2. As shown, the model without the CEN module produces more errors, with incorrect links appearing even in the early stages. **d**, A comparison of the confidence maps of the 2 models in the early

stages. We found that the CEN module shows more accurate performance of center extraction especially for larger cells, thus more efficiently reduces incorrect cell linking. All evaluation data in this figure were drawn from mskcc-confocal dataset #2. All models were trained using mskcc-confocal dataset #3. Scale bars: 10 μm .

Figure R26 (Supplementary Figure 8 in the revised manuscript) | Influence of level configuration during training on tracking performance. **a**, 2 frames of signal intensity maps and confidence maps obtained with 3-different level configurations via CELLECT. Horizontal and vertical line profiles are depicted in purple and orange, respectively. **b**, Extended view of ground truth-applied maps. From the ground truth of the center location of each cell, we generated a 3-pixel radius binary mask for the 2 frames and multiplied it with intensity maps and confidence maps, respectively. The location of the extended view is indicated with a green box in each original image in **(a)**. Overlapped areas with ground truth are shown in red and cyan for intensity and confidence maps, respectively. **c-d**, The impact of level configuration on the model's performance was evaluated, demonstrating that the 4-level configuration used in this study achieved the best performance. This configuration enhances center point distinction through multi-level refinement. Errors connections are highlighted in pink, while correct connections are depicted in grey. All evaluation data in this figure were drawn from mskcc-confocal dataset #2. All models were trained using mskcc-confocal dataset #3. Scale bars: 10 μ m.

Comment 2:

****Clarification on Linajea's Performance at Low Frame Rates:****

In the Introduction, the authors state: "However, its global optimization of cell trajectories based on time series information leads to large computational costs and degradation in datasets acquired at low frame rates, restricting its applications in large scale long term 3D imaging." This conclusion about performance degradation at low frame rates may not be straightforward. It would strengthen the manuscript to include an experiment where time frames are deliberately undersampled, and Linajea is evaluated on such datasets. Similarly, it would be informative to assess CELLECT's robustness under frame rate undersampling conditions. This would provide direct evidence supporting the claim and offer practical insights into both methods' performance in low temporal resolution settings.

Response:

We thank the reviewer for this insightful and constructive suggestion. The statement in the Introduction regarding Linajea's performance limitations at low frame rates is based on its structural reliance on densely sampled temporal windows. Specifically, Linajea performs global trajectory optimization over a fixed short-term span (e.g., 15 consecutive frames), which introduces strong dependency on continuous temporal information. This design increases its computational costs and limits its robustness when dealing with data sampled sparsely in the temporal domain.

We fully agree that experimental validation would strengthen this claim. However, Linajea does not provide publicly available pretrained models or training configurations, which prevents re-evaluation on custom temporal downsampling settings. As such, our

comparison is based on the algorithmic characteristics described in the original papers, together with the official benchmark results. To directly assess CELLECT's robustness under reduced temporal resolution, we conducted a series of controlled experiments using temporally downsampled versions of the *C. elegans* embryo dataset. These experiments are shown in **Figs. R27, R28, R29** (corresponding to **Supplementary Figs. 11, 12, 14** in the revised manuscript). **Supplementary Fig. 11** evaluates CELLECT without retraining, using models trained solely on the original temporal resolution to simulate inference-only robustness. **Supplementary Fig. 12** presents results from models trained with specific adjustments, including an increased candidate pool size and the incorporation of multi-timescale sampling, to assess how these training strategies affect tracking performance under reduced frame rates. Building on this, **Supplementary Fig. 14** further evaluates the linking performance of 5 independently trained CELLECT models across different temporal resolutions, using three example target cells to provide a more detailed view of frame-to-frame association stability.

The results show that tracking performance does decline under temporal downsampling, particularly in compressed intervals involving division events, which increases the chance of missed detections for dividing cells. Nevertheless, CELLECT still maintains strong overall tracking performance. Furthermore, incorporating multi-timescale training samples and increasing the number of candidate cells during inference significantly reduces false negatives and broken associations, helping to sustain accurate and stable tracking behavior.

This analysis is documented in the revised **Methods** section under ***Temporal linking with larger intervals***. To avoid potential misunderstanding, we have further revised the original sentence in the **Introduction** to more accurately reflect the structural characteristics of Linajea.

This revised formulation is intended to clarify that our observation concerns the inherent architectural constraints of the method, rather than conclusions drawn from new empirical comparisons.

a Inference across different frame rates

b

c

d

e

Figure R27 (Supplementary Figure 11) | Influence of imaging frame rate on tracking performance under different candidate selection strategies. **a**, Illustration of 3 different levels of temporal down-sampling were used for evaluation: $\times 1$ (75-s interval), $\times 2$ (150-s interval), and $\times 3$ (225-s interval) down-sampling. **b**, Quantitative tracking performance of CELLECT under 3 conditions of temporal down-sampling. **c**, Schematic illustration of 2 cell selection strategies: left, spatial-distance-based selection of the 5 nearest neighbors (used in CELLECT); right, feature-distance-based selection of the 5 most similar feature vectors. **d**, Error curves (FN + FND + IS) over time for both strategies under different temporal down-sampling. The feature distance-based strategy performs comparably at early frames when cell density is low, even under reduced temporal resolution, but its performance degrades rapidly as cell density increases over time. **e**, Circular tree diagrams displaying tracking errors across all 6 settings. All evaluation data were drawn from mskcc-confocal dataset #2; the model was trained on mskcc-confocal dataset #3 without temporal down-sampling. Scale bar: 10 μm .

Figure R28 (Supplementary Figure 12 in the revised manuscript) | Influence of imaging frame rate on tracking performance under different training dataset and candidate numbers used. a, Illustration of the 2 temporal resolutions used in the inference process: $\times 1$ (75 s) and $\times 2$ (150 s) down-sampling. b, Tracking performance of 3 different configurations for different temporal down-sampling in inference: the default

CELLECT configuration, model using 10 candidate cells, and model using 5 candidate cells but trained with different temporal resolutions (randomly incorporating dataset with $\times 1$, $\times 2$ and $\times 3$ temporal down-sampling). All configurations show comparable performance under $\times 1$ temporal down-sampling. However, under $\times 2$ temporal down-sampling, models trained with an expanded candidate pool or mixed-temporal-resolution data exhibit reduced false negative (FN) rates, indicating improved robustness to lower temporal resolution. **c**, Circular lineage diagrams corresponding to the 6 experimental settings. All results were drawn from mskcc-confocal dataset #2, with models trained on dataset #3 with and without temporal down-sampling. Scale bar: 10 μm .

Figure R29 (Supplementary Figure 14 in the revised manuscript) | Evaluation of the reproducibility of cell linking during different imaging frame rate. To evaluate the reproducibility of CELLECT's cell linking under different imaging frame rate, we selected 3 representative target cells (Target

#1–3) from mskcc-confocal dataset #2 and selected 5 nearest spatial neighbors at frames $t+1$, $t+2$, and $t+3$ for testing. **a**, Location of the target cells at time t (frame 1) and the locations of candidate cells across the 3 subsequent frames. **b**, Similarity scores predicted by 5 independently trained CELLECT models for the 5 candidates. To ensure consistent input across models, redundant point removal was not applied intentionally, resulting in occasional duplicate redundant cell points (e.g., Target #1 and #2 at frame 1 linked to frame 2). This configuration increases linking difficulty by expanding the temporal gap, thereby simulating more complex multi-frame association scenarios. While correct links often correspond to the nearest candidate in adjacent-frame conditions, longer intervals such as frame 1 to frame 3 or frame 4 require the model to rely more on embedding features for accurate decisions. The results demonstrate that CELLECT produces consistent linking performance across different frame intervals. However, under low imaging frame rate and in the presence of cell divisions (e.g., frame 1 linked to frame 4), a slight drop in accuracy is observed. All 5 models were independently trained on mskcc-confocal dataset #3. Scale bar: 10 μm .

Comment 3:

****Tiling Artifacts in Supplementary Figure 4:****

In Supplementary Figure 4 (bottom row), noticeable tiling artifacts are present. While overlapping tiles are mentioned, details on how these artifacts are mitigated are lacking. It would be helpful if the authors could clarify whether padding strategies are applied during inference. Valid (unpadded) convolutions combined with overlapping tiles might eliminate the artifacts. Reducing such artifacts may further improve downstream tracking accuracy.

Response:

We thank the reviewer for pointing this problem out. In our implementation, we use a non-padded tiled inference strategy with a 16-pixel overlap along the x and y axes, and a 4-pixel overlap along the z axis. No zero-padding or mirror-padding is applied. The tiling artifacts in **Fig. R30 (Supplementary Fig. 4** in the revised manuscript) are not caused by inconsistencies in convolutional padding. Instead, they result from differences in background suppression near the edges of individual patches. These differences arise because the model processes each 3D patch independently, and predictions near the patch boundaries lack surrounding contextual information, leading to subtle variations in background scores.

It is important to clarify that CELLECT's tracking framework does not rely on the full confidence map or embedding volume. Instead, only the embedding vector at each predicted center point is used. These edge-related inconsistencies therefore have minimal impact on center detection or feature-based identity association. Specifically, **Fig. R30(Supplementary Fig. 4** in the revised manuscript) illustrates a coarse

visualization of embedding-space distance to show local variation in feature similarity. This is intended solely for visualization purposes—neither training nor inference depends on the full embedding volume, but rather on localized features near the predicted center points. Moreover, the artifacts are typically confined to the boundary regions of 3D patches and do not affect the interior predictions, which dominate the accuracy of both segmentation and tracking. These visual inconsistencies do not lead to measurable degradation in model performance in terms of the tracking performance.

We have clarified in the caption of **Fig. R30 (Supplementary Fig. 4** in the revised manuscript) that these artifacts result from differences in background suppression near the edges of individual patches and do not affect the model's performance in segmentation or tracking tasks with the procedure of center-point extraction.

Figure R30 (Supplementary Figure 4 in the revised manuscript) | **Visualization of the embedding map obtained by CELLECT.** To analyze the spatial patterns captured by the model, we extracted embedding map corresponding to each voxel and computed their differences with surrounding voxels. This was achieved by convolving the squared values of the embedding map with a dilated convolution kernel, simulating neighborhood

interaction. A dilation rate of 1 represents the average difference with adjacent voxels, while a dilation rate of 5 represents the difference with voxels at a distance of 5 voxels in all directions. The resulting feature distance map highlights the magnitude and positive or negative of feature differences between distinct cell regions in the image. At the boundaries, the opposing patterns between the central and surrounding areas particularly emphasize the differences in feature values between cellular and non-cellular regions. These differences can also assist in optimizing cell segmentation during the inference phase, especially under sparse annotation conditions. The artifacts observed at patch boundaries arise from the independent inference of each patch, which do not affect the accuracy of segmentation or tracking, as only the embedding vectors at predicted cell centers are used for downstream analysis, and the boundary variations are negligible. The visualization shown here is based on the Cell Tracking Challenge (CTC) Fluo-N3DH-CE dataset. The model used was trained on sequence #3 of the mskcc-confocal dataset.

Comment 4:

****Clarification on Cell Tracking Challenge (CTC) Claims:****

The manuscript mentions achieving state-of-the-art tracking accuracy in the Cell Tracking Challenge. To improve transparency and reproducibility:

- Please specify which CTC public datasets were used for training and evaluation.
- It would be helpful to provide a link to the corresponding page on the CTC website showing the CELLECT results and method description.
- Please also include a link to the trained model (if available) in the manuscript.
- Consider adding quantitative results on the public CTC datasets to support the state-of-the-art claim.

Currently, results are shown only on two *C. elegans* datasets, which, though publicly available on Zenodo, are not part of CTC.

Response:

We thank the reviewer for raising this important point. In the revised manuscript, we have clarified that CELLECT was officially submitted to the Cell Tracking Challenge (CTC, <https://celltrackingchallenge.net/>). Our results can be found under the team name “THU-CN (3)” in the respective subsections on the official CTC website: Cell Segmentation Benchmark (CSB, <https://celltrackingchallenge.net/latest-csb-results/>) and Cell Tracking Benchmark (CTB, <https://celltrackingchallenge.net/latest-ctb-results/>). These results also presented in **Fig. R31**. In detail, our results were evaluated on the Fluo-N3DH-CE dataset (<https://celltrackingchallenge.net/3d-datasets/>), a publicly available 3D *C. elegans* embryo dataset from the CTC, which includes manually annotated 3D cell

organizers without us involved to ensure reproducible results. We used purple color to depict the result of the CELLECT score ranking 1st.

CELLECT ranked 1st overall performance (OP) for both segmentation competition ($OP_{CSB} = 0.853$) and tracking competition ($OP_{CTB} = 0.850$) on the test set of Fluo-N3DH-CE, demonstrating strong performance on challenging 3D datasets.

To further enhance transparency and reproducibility, we have released the full codebase, trained models, and the entire pipeline used for the CTC submission, all of which have been independently verified by the CTC organizers:

- <https://github.com/zzz333za/CELLECT>
- <https://github.com/zzz333za/CELLECT-ctc.ver>

The model was trained using the publicly available training subset of the Fluo-N3DH-CE dataset, which includes manually annotated 3D cell center coordinates and algorithm-generated 3D segmentation masks, fully satisfying the training requirements of CELLECT. In contrast, other 3D datasets in CTC either lack consistent annotation formats or contain incomplete labels, making them difficult to adapt without substantial modifications to the pipeline. We are also awaiting updates and documentation from the CTC organizers to facilitate further evaluation on such datasets. It is worth noting that both our code and documentation have been independently validated by the CTC organizers, which was a prerequisite for leaderboard publication. However, the public update of method descriptions on the CTC website has been delayed and is beyond our control (several other submissions are also pending).

To address this comment, we have added a new subsection titled *Evaluation on Cell Tracking Challenge benchmark* in the **Methods** section.

We are also considering additional comparisons on other CTC datasets using their public training data. However, many 3D datasets in CTC either lack compatible annotation formats or contain incomplete labels, making them difficult to adapt without substantial modifications to the pipeline. We are awaiting documentation updates from the CTC organizers to support further evaluation on such datasets. Moreover, the test set annotations are not publicly accessible, and most other top-ranking CTC methods do not release their training pipelines or support sparse annotation (CELLECT is fully open-source). As a result, we rely primarily on the official CTC evaluation metrics for direct performance comparison. A summary of these methods and their limitations is provided in **Table R10** of the revised manuscript.

Method	Training Code Availability	Segmentation OP	Tracking OP	Applicable for Sparse Annotation	Reason for Exclusion
THU-CN(3) (CELLECT)	Fully open source	0.853 (Highest)	0.850 (Highest)	✓	
CZB-US	Partially open source	0.847	0.844	✗	Not open-source yet
KTH-SE(1)	Open source	0.842	0.829	✗	Requires extensive manual parameter tuning and internal selection; difficult to generalize to new datasets
KIT-GE(2)	Partially open source	0.830	0.808	✗	Does not support sparse annotation; requires full segmentation labels for training

Table R10 | Summary of selected Cell Tracking Challenge (CTC) methods and their applicability for comparison in Fig. 2. This table summarizes CTC methods that are applicable to 3D datasets and for which publicly available implementations or explanatory documentation exist. Segmentation and tracking OP values are reported from the CTC leaderboard on the Fluo-N3DH-CE dataset. ✓ indicates compatibility with sparse annotation.

Comment H: Clarity and context:

Comment 1:

****Typographical Errors:****

- Line 547 ("C. elegans embryo Public"): "CELLEECT" → should be "CELLECT" .

- Figure 1: “Divsion prediction” → should be “Division prediction” .
- Figure 2a: “StarrayNite” → should be “StarryNite”.
- Supplementary Figure 8: “Singal Intensity” → should be “Signal Intensity” .
- Supplementary Figure 7: “Divsion predictor” → should be “Division predictor” .

Response:

We thank the reviewer for carefully pointing out these typographical errors. We have corrected all of them in the revised manuscript and supplementary materials:

- **Line 547:** “CELLEECT” → corrected to “CELLECT”.
- **Fig. 1:** “Divsion prediction” → corrected to “Division prediction” .
- **Fig.2a:** “StarrayNite” → corrected to “StarryNite”.
- **Supplementary Fig. 8:** “Singal Intensity” → corrected to “Signal Intensity”.
- **Supplementary Fig. 7:** “Divsion predictor” → corrected to “Division predictor” .

We also have carefully re-checked the manuscript and figures to ensure consistency and have corrected other minor typos as well.

Comment 2:

****Metric Clarification:****

Please explicitly mention in the manuscript that the "Tracking Accuracy" metric originates from Linajea (Malin-Mayor et al.). This metric is not commonly used in other tracking benchmarks, so citing its origin would provide helpful context for readers unfamiliar with it.

Response:

We thank the reviewer for this important clarification. The “Tracking Accuracy” metric reported in our manuscript is indeed adopted from the Linajea framework, originally introduced by Malin-Mayor et al. in their work on large-scale lineage reconstruction. This metric measures the proportion of cell trajectories that are completely reconstructed over a specified temporal window and is particularly useful for evaluating long-term identity consistency.

To ensure clarity and proper attribution, we have now explicitly cited the Linajea paper at the first mention of this metric in the **Main** text. This addition provides necessary context for readers unfamiliar with the metric and highlights its relevance to our evaluation protocol.

Response to Reviewers – NMETH-A59736A

We sincerely thank the reviewers and editors for the insightful comments and constructive feedback on our manuscript entitled “Contrastive Embedding Learning for Large-scale Efficient Cell Tracking: CELLECT.” We also appreciate the reviewers’ positive feedback and acknowledgment of our work. We have carefully addressed all remaining suggestions in this revision. Here, we have provided a point-by-point response to all concerns. For ease of communication, the original referee comments are shown in black color, whereas our specific answers are in blue. Thank you again for your great efforts.

Reviewer #1

Comment A: Summary of the key results:

The paper “Contrastive Embedding Learning for Large-scale Efficient Cell Tracking: CELLECT” by Zhou et al. proposes contrastive learning using sparse labeling with different fluorescent markers to train a 3D cell tracking. Besides contrastive learning, different customized subnets significantly improve segmentation and tracking results, shown by ablation studies.

The superiority of the approach against leading methods of the Cell Tracking Challenge (Maska 2023 Nature Methods) was shown by first places in the segmentation and tracking challenge in a formal submission to the Challenge website (<https://celltrackingchallenge.net>) for one 3D dataset.

Very encouraging, fast and impressive results are shown for various other complex tasks without retraining, e.g. for an in vivo screen for bacteria, neutrophils and bacteria in mice or tracking of firing neurons in elastically deformed drosophila brains.

In the submitted revised version, the paper is technical sound and I recommend a publication in Nature Methods

Response:

We sincerely appreciate the reviewer’s encouraging assessment and are pleased that the strengths of our approach were recognized. Our goal was to develop a framework that combines efficient learning strategies with robust architectural design to address real-world challenges in 3D biological imaging. We are especially encouraged that the practical utility of the method across a variety of settings was well received. This reinforces our belief that contrastive embedding, when paired with lightweight supervision, can serve as a broadly effective tool for large-scale cell tracking and beyond. We remain committed to supporting further use and development of the framework by the research community.

Comment B: Originality and significance: if not novel, please include reference:

The approach of contrastive learning to 3D cell tracking is novel, but relatively straightforward by combining leading methods from cell tracking and deep learning. The main improvement comes from the combination of sparse fluorescent labeling combining with the mentioned method to avoid a labor-intensive labeling of 3D datasets.

Response:

We appreciate the reviewer's thoughtful comments on the originality and significance of our work. Our goal was to adapt contrastive learning principles to the context of 3D cell tracking in a way that is both practical and accessible. In particular, we sought to reduce reliance on dense annotations by leveraging sparse fluorescent labeling, which is often available in biological experiments. We are encouraged that this strategy may help lower the entry barrier for applying deep learning in real-world microscopy workflows, especially when large-scale manual annotation is not feasible.

Comment C: Data & methodology: validity of approach, quality of data, quality of presentation:

All recommendations and questions from the previous review were successfully answered.

Minor points:

- * Clarify the exact input and output dimensions of intra- and inter-frame-MLPs in figure caption of Supp. Fig. 5 to the exact and well understandable settings in Suppl. Table 3.
- * Remove Neven at the end of Reference [52]

Response:

We thank the reviewer for confirming that all previous comments have been successfully addressed. Regarding the additional minor points:

- Clarification of MLP dimensions: We have updated the caption on **Supplementary Fig. 5 (Fig. R1)** to clearly specify the input and output dimensions of both intra-frame and inter-frame MLPs. The dimension format matches the explicit settings shown in **Supplementary Table 3** to ensure consistency and ease of understanding.
- Reference correction: We have removed the redundant mention of "Neven" at the end of Reference [52] as suggested. We apologize for any inconvenience.

We appreciate the reviewer's careful reading and constructive suggestions, which improved the clarity and presentation quality of the manuscript.

Figure R1 (Supplementary Figure 5 in the revised manuscript) | Illustrations of the intra-frame MLP and inter-frame MLP. a, Illustration of the workflow of 2 MLPs. The 5 nearest candidate cells to the target cell are selected based on the spatial distances between cell centers. For intra-frame MLP, we selected the 5 candidate cells within the same time frame. For inter-frame MLP, we selected the 5 candidate cells in the next frame closest to the target cell in the current frame. For each candidate, four feature distances relative to the target cell are computed: (1) size and division probability difference, (2) size and division probability ratio, (3) Euclidean distance, and (4) feature similarity. These metrics serve as inputs to the MLPs. The intra-frame MLP identifies redundant cell center point detections of the

same cell, while the inter-frame MLP determines whether the target cell matches a candidate in the next frame or undergoes division. Each MLP outputs 6 similarity scores including 5 candidates and one ‘None’ class indicating that the target cell is not matched with any of the 5 candidates. The inter-frame MLP outputs 6 similarity scores and also provides a division prediction. **b**, Representative example of intra-frame MLP. For a given target cell (target 1), its six nearest cell center points (labeled 1-1 to 1-6) are identified within the same frame. We note that point 1-1 is a redundant detection of the same cell as the target. **c**, Intra-frame MLP outputs for different combinations of 5 out of the 6 candidates. Including the redundant detection (1-1) results in high confidence scores for that match, whereas its exclusion leads to the ‘None’ class having the highest score. **d**, Inter-frame MLP example for two target cells: target 2 (not dividing) and target 3 (undergoing division). Each is associated with 7 nearest candidates in the next frame. **e**, Inter-frame MLP outputs using three different candidate selection sets (e.g., 1-5, 2-6, and 3-7) for target cells 2 and 3. Inclusion of the correct corresponding cell (e.g., 2-1 for target 2) or daughter cells (3-1 and 3-2 for target 3) in the candidate set results in high similarity scores from the MLP. Exclusion of true matches causes the ‘None’ score to dominate. The division prediction is mainly driven by the input division probability and remains unaffected by candidate selection. The visualization in **(a)** is based on the Cell Tracking Challenge (CTC) Fluo-N3DH-CE dataset, while the examples in **(b–e)** are from sequence #2 of the mskcc-confocal dataset. All results were generated using a model trained on sequence #3 of the mskcc-confocal dataset.

Comment D: Appropriate use of statistics and treatment of uncertainties

Yes. Uncertainties are explicitly handled for segmentation masks in form of the confidence map (see e.g. Suppl. Fig. 2).

Response:

We thank the reviewer for acknowledging the use of confidence maps to represent segmentation uncertainty. We are glad this aspect was clear.

Comment E: Conclusions: robustness, validity, reliability:

Okay.

Response:

We thank the reviewer for the positive assessment of the conclusions.

Comment F: Suggested improvements:

Okay.

Response:

We appreciate the reviewer's confirmation that no further changes are needed.

Comment G: References: appropriate credit to previous work?:

Yes.

Response:

We thank the reviewer for acknowledging the completeness of the citations.

Comment H: Clarity and context:

No problems.

Response:

We are glad the reviewer found the presentation clear and appropriate.

Reviewer #2

Summary:

We would truly like to congratulate the authors on the revision and particularly on the details given in the letters to the reviewers. They truly address all of our concerns, except two that we list below. We think that the issues we have with the code quality are really important to address for a publication showcasing innovative code.

Response:

We sincerely thank the reviewer for their positive comments and for the careful inspection of our code. We revised the repository following reviewer's suggestions to make more structured and formal codebase, and also provided clearer usage examples to facilitate more convenient adoption for users.

Citation of tools used for comparison:

The corresponding tool citations and configuration details have been added to the updated Methods section titled Comparative methods and parameter settings.

We thank the authors for the added precisions. They are indeed useful. However we could not find the citations of the tools used in comparison in the reference list. Maybe they were forgotten?

Response:

We thank the reviewer for the helpful comment and apologize for any inconvenience. We initially cited the tools only at their first appearance of the content. Following your suggestion, we have now added the corresponding citations directly in the comparative methods section. We revised it as below.

“For TrackMate⁵⁵ (via Fiji, version 7.10.1), we used the provided pre-trained Weka⁵⁶ model detector applied in a slice-by-slice manner. The gap-closing interval was set to 2,000 frames for neuronal recordings, consistent with the Imaris configuration. For StarryNite^{34,36,37} (2020 release), we adopted the default parameter configuration distributed with the 2020 version of StarryNite and modified the cell radius range settings based on the ground-truth annotation files.”

34. Santella, A., Du, Z. & Bao, Z. A semi-local neighborhood-based framework for probabilistic cell lineage tracing. BMC Bioinformatics 15, 217–217 (2014).

36. Bao, Z. et al. Automated cell lineage tracing in Caenorhabditis elegans. Proc. Natl. Acad. Sci. 103, 2707–2712 (2006).

37. Murray, J. I. et al. Automated analysis of embryonic gene expression with cellular resolution in C. elegans. Nature Methods 5, 703–709 (2008).

55. Ershov, D. et al. TrackMate 7: integrating state-of-the-art segmentation algorithms into tracking pipelines. *Nature Methods* 19, 829–832 (2022).

56. Arganda-Carreras, I., Kaynig, V., Rueden, C. et al. Trainable Weka Segmentation: a machine learning tool for microscopy pixel classification. *Bioinformatics* 33, 2424–2426 (2017).

Code quality and reusability:

We acknowledge and appreciate the efforts of the authors regarding CELLECT installation and usage. We note that the software can now be installed on Mac platforms. We could install the code and run the pipeline on the test dataset. However, in its current state, CELLECT does not achieve the quality required for a publication. We are convinced that fixing bugs and implementing better code quality will greatly improve the publication. We list below the issues we have found. Without addressing them, CELLECT won't be usable widely, which will greatly limit the impact of the article.

Response:

We sincerely thank the reviewer for the detailed testing and constructive feedback on the CELLECT software, and we appreciate the confirmation that the installation and pipeline execution were successful. We take these comments very seriously and fully agree that improving usability is critical to the value and impact of this work.

In response to the reviewer's suggestions, we have standardized the formatting and naming conventions throughout the codebase. We have also reduced external library dependencies by implementing commonly used functions internally, which improves portability and reproducibility. In addition, we have updated the documentation and added more accessible usage options, including simplified command-line execution and callable functions with examples.

The updated version of CELLECT is now available on GitHub <https://github.com/zzz333za/CELLECT/>.

Major point:

Comment 1:

Some packages are installed via conda, others via pip through the requirements file. Numpy is installed 2 times, both via conda and pip. This is bad practice and could generate dependency conflicts or break the environment.

Response:

We thank the reviewer for pointing out this issue. We fully agree that installing the same package (such as numpy) using both conda and pip in the same environment can cause dependency conflicts.

To address this, we have substantially upgraded the codebase. The data preprocessing components have been rewritten using our own lightweight utility functions, which greatly reduce external dependencies and simplify environment setup. The installation process now only requires creating a virtual environment and executing the following commands:

▪ **Install:**

```
cd CELLECT/  
conda create -n collect python=3.11 -y  
conda activate collect  
pip install -r requirements.txt
```

The requirements.txt includes only four packages: torch (deep learning framework), tqdm (progress display), pandas (table handling), and tiff file (TIFF file reading). We have verified that the entire pipeline runs successfully on macOS, Linux, and Windows systems without any manual modification. We further tested this simplified installation across macOS, Linux, and Windows, and verified that the pipeline also runs without requiring any manual modification.

We would like to clarify that our original installation was based solely on pip, but at that time, we had not fully accounted for the compatibility issues across different platforms. Traditional image processing libraries such as Pillow, scikit-image, and OpenCV, along with torch, rely on different versions of numpy across systems. This also led to the issue noted in the reviewer's earlier feedback: certain packages failed to install on their environment, preventing the program from running. To address this, we adjusted the installation order and, to meet these platform-specific dependency constraints, redundantly installed numpy. This was done to ensure that the provided installation procedure would allow the pipeline to be deployed and executed smoothly across macOS, Linux, and Windows systems without requiring any manual modification. In the updated version, we have completely removed the traditional image processing libraries that previously caused compatibility issues. This further simplifies installation and mitigates the installation failures noted previously.

Comment 2:

Paths are hardcoded. It is thus impossible to run CELLECT as is, but with different data than the test dataset. For example, see s-test.py, line 125:

```
vl=os.listdir(datap+'mskcc_confocal_s'+op+'/images/')
```

Response:

We sincerely thank the reviewer for this valuable suggestion. We agree that parts of the s-test.py script contain hardcoded paths, which may cause inconvenience when attempting direct modification. However, we would like to clarify that this script was primarily

designed to reproduce the results in the paper based on the mskcc-confocal dataset. It supports testing different sub-datasets through input arguments and is intended for controlled and repeatable experimental evaluation.

While users can also apply this script to their own custom datasets, it requires following specific file naming conventions and directory layouts, as the pipeline relies on both fixed file paths and strict sequential ordering of image frames. As illustrated in **Fig. R2**, simply modifying the file path without preserving the expected structure and order will result in failure during execution, whereas using our provided structure justifies the presence of these hardcoded paths.

To improve flexibility, we have provided full training and testing code compatible with multiple public datasets, including mskcc-confocal, nih-ls, and datasets from the Cell Tracking Challenge, all of which conform to the required folder structure. For users working with custom data, we also offer a Jupyter notebook-based tutorial. We further demonstrate how to call the encapsulated CELLECT modules with a single line of code to (**Fig. R3a**) extract cell information per frame, (**Fig. R3b**) perform linking between frames, and (**Fig. R3c**) visualize both instance segmentation and tracking results, as shown in **Fig. R3**. (<https://nbviewer.org/github/zzz333za/CELLECT/blob/main/example-packaged-CELLECT.ipynb>)

In addition to these enhancements, we have developed a new, user-friendly inference module (`inferenc.py`) that lets you apply CELLECT directly to your own data. You only need to point it to a folder of sequentially named 3D `.tif` or `.tiff` images—our parser will automatically detect a wide variety of naming conventions (e.g. `sample_001.tif`, `img_0005.tiff`, `an_frame6.tif`, etc.) and sort them correctly. A rich set of optional command-line parameters allows you to tailor the pipeline to your specific dataset without any code modifications.

- Inference:

```
python inference.py --data_dir /path/to/image_folder \
--out_dir /path/to/output_folder --model_dir ./model/U_ext_x3rd.pth \
--model2_dir ./model/MLP_intra.pth \ --model3_dir ./model/MLP_inter.pth
```

Figure R2 | Dataset structure illustration. a, Directory structure of the *mskcc-confocal* dataset. **b,** Structure of the *Fluo-N3DH-CE* dataset from the Cell Tracking Challenge

Figure R3 | Screenshot of the packaged CELLECT Jupyter notebook example.

Comment 3:

Code readability and maintainability: there are very few comments, no docstrings, some of the variable and function names are obscure (e.g. kf, ud, uc...), which hinders reusing the

code or having others study it or build on it (see specifically `s-test.py` and `s-train.py`). The code would benefit greatly from following Python good practices (see PEP 81 for example). We think that at least the key functions should be well documented and made readable. As we said in our previous review we are convinced that the algorithms implemented are novel and can be impactful. It is likely that many others will be inspired by this work, and they will hope to find a code quality that matches the ambition of the tool.

Response:

We thank the reviewer for emphasizing the importance of code readability and maintainability. In response, we have substantially revised the codebase to improve structure, documentation and overall usability. Over 100 ambiguous variables and function names have been replaced with descriptive identifiers to enhance clarity. For example:

- `U` → `feature_extract_net`
- `EX` → `inter_frame_matcher`
- `EN` → `intra_frame_matcher`
- `kflb` → `search_mask`
- `ud` → `z_enlarge`
- `Uloss` → `loss_seg_total`
- `rloss` → `contrastive_loss`
- `kloss` → `classification_loss`
- `p1` → `center_mask_1`
- `s1` → `division_map_1`
- `lsize1` → `gt_size_1`

These changes are intended to make the code more intuitive and easier to navigate for users seeking to reuse or extend specific modules.

Additionally, we have added detailed docstrings for all major functions, and added inline comments to clearly annotate the stages of data processing and loss computation. For matrix operations involving variables such as `ud` and `kf`, we have grouped them by functional category and annotated each function with a clear explanation of its purpose. We have also reformed the entire codebase to comply with PEP8 standards and used tools such as `autopep8` to verify compliance.

These improvements substantially enhance the readability, maintainability, and reproducibility of the CELLECT's codebase. We continue to refine sub-function documentation and the overall structure to align with best software development practices.

Minor point:**Comment 1:**

Please reread your code. Currently, some Python files look like personal throwaway code which can be okay in some cases but not for a published tool. For example: unordered, duplicated or unused imports, useless lines (e.g. `j=i#+250#+150#+220#+200`), weird and inconsistent spacing... This mostly concerns the notebook and the `s-test.py` and `s-train.py` files.

Response:

We apologize for the presence of development-stage remnants in the previous version of the code. We have carefully reviewed and cleaned the entire codebase. All unnecessary elements have been removed.

Comment 2:

While having a notebook to demonstrate a tool is always nice, in this case it doesn't really help. The goal of a notebook is not to have chunks of code from the framework copy-pasted in different cells, but to call the functions directly from the framework to explain how and why to use them in an example use case. If CELLECT is only meant to be used via command line and not as a framework, a notebook might not be the best way to explain how to use it. But if CELLECT can be used as a framework, please document and illustrate the various functions in a clearer way in the notebook.

Response:

We thank the reviewer for the valuable suggestion, and we apologize for the earlier misunderstanding. Our initial intention with the notebook was to break down and illustrate the test pipeline, including the steps of feature extraction, redundancy filtering, and cell linking, along with the corresponding intermediate outputs.

We now understand the reviewer's point more clearly and have added a new notebook that demonstrates how to use the encapsulated version of CELLECT as a framework. In this version, CELLECT's core functions for embedding extraction and cell linking can be invoked with a single line of code, and all key parameters are returned in a dictionary format. Based on these parameters, users can easily implement cell localization, extraction, and instance segmentation on new images, as illustrated in **Fig.R3**. (<https://nbviewer.org/github/zzz333za/CELLECT/blob/main/example-packaged-CELLECT.ipynb>)

Reviewer #4

Comment A: Summary of the key results:

(a) The authors introduce CELLECT, a deep learning-based tracking framework that improves accuracy and reduces computational cost by learning per-detection embeddings using a contrastive loss. These embeddings are then used to match detections between adjacent frames based on feature

similarity, eliminating the need for globally constrained optimization and enabling real-time inference.

(b) The authors demonstrate that a model trained on a single dataset (mskcc-confocal) generalizes well to a other datasets and imaging modalities without requiring retraining or fine-tuning.

(c) The combination of high tracking accuracy, strong cross-domain generalization, and real-time performance makes CELLECT a practical tool for large-scale biological applications.

(d) The paper presents a quite thorough investigation with several experiments, ablations and evaluated datasets.

Response:

We greatly appreciate the reviewer's positive assessment of CELLECT, particularly regarding its efficiency, accuracy, and application potential in biological imaging. The review correctly highlights the key innovations of CELLECT. We are pleased that the reviewer finds CELLECT's computational efficiency, especially in the linking step, to be notable. In the following sections, we address the specific points raised by the reviewer to further improve the manuscript and clarify important aspects of our work.

Comment F: Suggested improvements:

Comment a:

The discussion section could be strengthened by addressing how CELLECT might support human-in-the-loop tracking, e.g., by identifying uncertain cell detections or links that could be prioritized for manual annotation, particularly when the pre-trained model underperforms on a novel dataset.

Response:

Thank you to the reviewer for the helpful suggestion. We fully agree with this comment and have incorporated this point into the revised discussion. CELLECT already provides a reliability score for each cell link. For detection, it identifies unstable cells by marking those that are lost before reaching image boundaries or those that appear abruptly but exhibit continuous motion across frames. In such uncertain cases, human-in-the-loop manual correction remains highly valuable for refining unreliable detections and links. We believe this can not only enhance performance on unseen datasets but also facilitate more targeted fine-tuning or retraining. To address this, we have added the relevant sentence in the discussion.

“Building on this, CELLECT also supports human-in-the-loop correction by providing additional reliability scores for cell linkage and identifying unstable detections, such as cells that vanish prematurely or appear abruptly. Even for challenging or unseen datasets, manual refinement through assigning higher reliability scores can make CELLECT focus more on the new dataset, further improving usability and adaptability”

Comment b:

The authors should make the pre-trained, generalist CELLECT model available for public download and include the URL in the manuscript to facilitate reproducibility and adoption by the community.

Response:

We thank the reviewer for your kind recognition. The pre-trained model used in our experiments, including the one submitted to the Cell Tracking Challenge, along with the full training and evaluation code, is already available for public download via the GitHub repository linked in the manuscript. Due to the lightweight nature of the pre-trained model (trained on the mskcc-confocal dataset), it is directly hosted within the repository itself (<https://github.com/z333333/CELLECT/tree/main/model>). In addition, the Cell Tracking Challenge submission version is available at https://github.com/z333333/CELLECT-ctc.ver/tree/main/origin_sub/xmodel. We also cited full URLs in the revised manuscript (Results, Overview of CELLECT section).

We would like to clarify that the currently shared model is not yet a generalist model, but one of the cross-validation models trained on a subset of the mskcc-confocal dataset. To facilitate broader reproducibility and adoption, we are in the process of training and releasing more generalist CELLECT models trained across multiple datasets, including mskcc-confocal, nih-1s, and Cell Tracking Challenge benchmarks. These will be made publicly available in the same repository and clearly documented for community use.

Comment c:

Although the generalizability of the pretrained model becomes evident later in the paper, the authors could make it more explicit early on (e.g., in the abstract and introduction) that the model was trained solely on the mskcc dataset and applied across diverse datasets. This broad generalization is a significant contribution on its own.

Response:

We thank the reviewer for the helpful suggestion. We fully agree that the generalization ability of CELLECT is an important contribution and should be emphasized earlier in the manuscript. To address this, we have revised the relevant sentence in the abstract for clarity and emphasis.

Original sentence:

“By contrastive learning of latent embeddings of diverse cellular structures, a pre-trained CELLECT model can be effectively applied across different imaging modalities and species with broad generalization, validated in open-source datasets and experimental benchmarking.”

Revised sentence:

“By contrastive learning of latent embeddings of diverse cellular structures, a CELLECT model pre-trained solely on a dataset can be effectively applied across different imaging modalities and species with broad generalization, validated in open-source datasets and experimental benchmarking.”

We also clarified in the introduction that the model was trained solely on the mskcc-confocal dataset and achieves strong generalization without further tuning, aligning with the reviewer’s helpful suggestion.

Comment d:

What happens if a new dataset has much larger cells than the ones in mskcc? How does CELLECT deal with this (for example, does one need to downsample new datasets to have similar cell size as mskcc?) A mention of this in the discussion would be welcome.

Response:

We thank the reviewer for the valuable comment. In principle, CELLECT is not limited to a specific target scale. For example, the mskcc-confocal dataset used in our study includes a wide range of cell sizes observed during *C. elegans* embryonic development, with diameters ranging from approximately 8 to 70 voxels.

In our implementation, both training and inference are conducted on image patches. When CELLECT is applied to datasets containing cells significantly larger than those seen during training, the increased spatial coverage of a patch may lead to more redundant detections, thereby increasing the risk of linking errors and potential fragmentation of cell trajectories. In such cases, downsampling the input images to reduce the apparent cell size is an effective way to maintain robust tracking performance.

Additionally, CELLECT includes a module for estimating cell size (i.e., approximate cell radius in voxels). When encountering larger cells, it is possible to design a strategy for automatically adjusting the downsampling factor based on the estimated size, thereby adapting to scale differences. We have updated the discussion section to reflect the current limitations of this mechanism and describe future directions for developing automatic scale-adaptive capabilities.

Comment e:

Often biologists have this question - how frequently should one image to have the best downstream tracking? Could the authors provide any guidelines or discussion related to this?

Response:

We thank the reviewer for raising this important and practical question. We have added brief guidelines in the revised version. In general, a lower imaging frequency increases the difficulty of tracking, particularly when key events such as intercellular motion, collisions, or cell division occur between frames. The more such behaviors are missed, the harder it becomes to correctly associate and track individual cells across time.

Therefore, rather than suggesting a fixed imaging frequency, we recommend that the temporal resolution should at least match the timescale of the cellular behaviors under investigation. Ensuring that imaging captures these events as they occur is essential for maintaining tracking accuracy. We have included more discussion on this point in the revised manuscript.

Comment H: Clarity and context:**Comment a:**

Title:

Consider reordering: “CELLECT: Contrastive Embedding for Large-scale, Efficient Cell Tracking”.

Response:

We thank the reviewer for the suggestion. We reordered the title to improve clarity and focus as reviewer's comment.

Comment b:

Abstract:

(i) "a pretrained CELLECT model" → a pre-trained CELLECT mode

(ii) "CELLECT reduces the computational costs by 56 times"

CELLECT reduces inference-time computational cost by a factor of 56.

(iii) Consider adding: This efficiency gain arises from replacing globally constrained optimization with greedy, frame-wise matching based on learned embeddings. This insight may help attract broader interest from users seeking real-time solutions.

(iv) "CELLECT, which achieves state of the art tracking accuracy in the Cell Tracking Challenge" → CELLECT achieves state-of-the-art tracking accuracy on the C. elegans dataset in the Cell Tracking Challenge

Response:

We have revised the abstract accordingly, including clarifying the model's training dataset, refining the description of computational efficiency, and highlighting the use of learned embeddings for frame-wise matching.

Comment c:

Introduction:

(i) "Deep-learning-based cell tracking algorithms" → Deep learning-based cell tracking algorithms

(ii) The sentence: "However, its reliance on fixed-length temporal windows for global trajectory optimization introduces high computational costs and strict requirements on temporal sampling, limiting its real time performance and applicability to low-frame rate or irregularly sampled data." could be trimmed to avoid over-claiming: "...introduces high computational costs and strict requirements on temporal sampling, limiting real-time performance." (The claim about low-frame-rate applicability was not tested in the Linajea paper.)

Response:

We thank the reviewer for the comment. We have updated the wording to avoid overstatements and ensure consistency in phrasing.

Comment d:

Results – Section Title:

(i) Consider revising "Principle of CELLECT" to:

“Core Idea Behind CELLECT” or “Overview of CELLECT” for improved tone.

Response:

We thank the reviewer for the suggestion. We have revised the subsection title to better reflect the content.

Comment e:

In general, the language in the paper can be refined a little further by using shorter and cleaner sentences.

Response:

We apologize for the lack of clarity in some parts. We have carefully revised long sentences in the abstract, introduction, or discussion to improve the overall clarity of the manuscript.